# On the Stability and Generalization of Meta-Learning: the Impact of Inner-Levels

**Wenjun Ding**[1,2], **Jingling Liu**[1,*], **Lixing Chen**[3,4], **Xiu Su**[5], **Tao Sun**[6], **Fan Wu**[1], **Zhe Qu**[1,2,*]

[1]School of Computer Science and Engineering, Central South University
[2]Xiangjiang Laboratory
[3]School of Electronic Information and Electrical Engineering, Shanghai Jiao Tong University
[4]Shanghai Key Laboratory of Integrated Administration Technologies for Information Security
[5]Big Data Institute, Central South University
[6]College of Computer, National University of Defense Technology
`{234712240,jinglingliu,xiusu1994,zhe_qu}@csu.edu.cn,`
`lxchen@sjtu.edu.cn, suntao.saltfish@outlook.com`

## Abstract

Meta-learning has achieved significant advancements, with generalization emerging as a key metric for evaluating meta-learning algorithms. While recent studies have mainly focused on training strategies, data-split methods, and tightening generalization bounds, they often ignore the impact of inner-levels on generalization. To bridge this gap, this paper focuses on several prominent meta-learning algorithms and establishes two generalization analytical frameworks for them based on their inner-processes: the Gradient Descent Framework (GDF) and the Proximal Descent Framework (PDF). Within these frameworks, we introduce two novel algorithmic stability definitions and derive the corresponding generalization bounds. Our findings reveal a trade-off of inner-levels under GDF, whereas PDF exhibits a beneficial relationship. Moreover, we highlight the critical role of the meta-objective function in minimizing generalization error. Inspired by this, we propose a new, simplified meta-objective function definition to enhance generalization performance. Many real-world experiments support our findings and show the improvement of the new meta-objective function.

## 1 Introduction

Meta-learning has been proven to be a powerful paradigm for extracting well-generalization from previous tasks and quickly learning new tasks [1]. It has received increasing attention in many machine learning applications such as few-shot learning [2], robust learning [3], and natural language processing [4]. The key idea of meta-learning is to improve the learning ability of agents through a learning-to-learn process. In recent years, optimization-based meta-learning algorithms have emerged as a popular approach [5–8]. These studies formulate the problem as a bi-level optimization problem and have demonstrated impressive performance across various domains, drawing significant attention from the research community. In particular, at the outer-level, it trains a meta-learner to extract task-shared knowledge from meta-training tasks. At the inner-level, a basic model, which is initialized using the meta-parameters, adapts to each task by taking $Q$ inner-level gradient updates, where the $Q$ times inner-level update is commonly called an inner-process [5, 9–11].

Despite the remarkable success of meta-learning, its theoretical understanding of generalization remains largely unexplored. Recent studies have primarily focused on analyzing the effect of training

---

*Corresponding authors.

Table 1: The summary of main meta-learning algorithms

| Frame. | Algorithm | Inner-Level Process | Convex | Non-Convex |
|---|---|---|---|---|
| GDF | MAML[5] 
 FO-MAML [5] 
 Meta-SGD [6] | $w_{\mathcal{T}_i} = w - \alpha \sum_{q=0}^{Q-1} \nabla \widehat{\mathcal{L}}_i(w_{\mathcal{T}_i}^q, \mathcal{D}_i)$ 
 $= \arg\min_{w_{\mathcal{T}_i}} \{ \langle \sum_{q=0}^{Q-1} \nabla \mathcal{L}_i(w_{\mathcal{T}_i}^q, \mathcal{D}_i),$ 
 $w_{\mathcal{T}_i} - w \rangle + \frac{1}{2\alpha} \|w_{\mathcal{T}_i} - w\|_2^2 \}$ | $\mathcal{O}\left( \frac{TQ}{mn^{\text{tr}}} + \frac{\sqrt{F(w^0) - \min_{\mathcal{W}} F + T}}{m} \right)$ | $\mathcal{O}\big( \frac{1 + \frac{1}{c_\gamma}}{m}(1 + \frac{Q}{n^{\text{tr}}})\big)^{\frac{1}{c_\gamma}} \big( F(w^0)T \big)^{\frac{c_\gamma}{1+c_\gamma}}$ |
| PDF | iMAML [20] 
 Meta-MinibatchProx [7] 
 FO-MuML [11] | $w_{\mathcal{T}_i} = \operatorname{argmin}_{w_{\mathcal{T}_i}} \widehat{\mathcal{L}}_i(w_{\mathcal{T}_i}, \mathcal{D}_i)$ 
 $+ \frac{\lambda}{2}\|w_{\mathcal{T}_i} - w\|^2$ | $\mathcal{O}\left( \frac{T}{mC^Q} + \frac{\sqrt{F(w^0) - \min_{\mathcal{W}} F + T)}}{m} \right)$ | $\mathcal{O}\big( \frac{1 + \frac{1}{c_\gamma}}{m}(1 + \frac{1}{C^Q})\big)^{\frac{1}{c_\gamma}} \big( F(w^0)T \big)^{\frac{c_\gamma}{1+c_\gamma}}$ |

strategy [12, 13], data splitting methods [14, 15] on generalization error or on advancing tighter generalization bounds [16, 17], while overlooking the impact of inner-levels on generalization, i.e., ***the relationship between the generalization error and the number of inner-levels $Q$.***

In particular, there are two main inner-processes frameworks in current meta-learning algorithms, as shown in Table 1. One is the Gradient Descent-based Framework (GDF) [5, 6, 18], where the key idea is to measure the closeness of the initial prior hypothesis to the target optimal hypothesis by the number of gradient descent steps. However, this approach incurs high computational costs due to the need for second-order derivatives and requires careful tuning of multiple hyper-parameters [19]. To address these limitations, another inner-process framework PDF, which is based on proximal descent, has been developed [20, 7, 11, 21]. It only depends on the solution to the inner optimization and not the path taken by the inner optimization algorithm.

To validate the impact of $Q$ on the two frameworks, we conducted two simple experiments on the Omniglot dataset [22] using MAML [5] and Meta-MinibatchProx [7] as examples. In Figure 1, the generalization error (Test Loss - Training Loss) first decreases but then increases as the number of inner-levels $Q$ grows in MAML, while it always decreases in Meta-MinibatchProx. This different behavior, attributed to the distinct inner processes of the two frameworks, motivates the need for a deeper analysis to help the design of $Q$ for improved generalization performance.

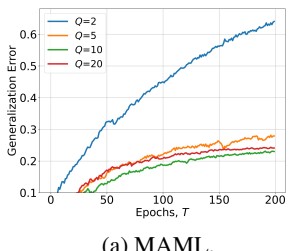

(a) MAML.

To analyze the relationship between inner-levels $Q$ and generalization error under these two frameworks: GDF and PDF, this paper leverages the algorithmic stability to characterize the generalization of algorithm [16, 23], which measures sensitivity to perturbations in the training dataset. To the best of our knowledge, this is the first study to investigate the influence of inner-levels on the generalization of meta-learning under two frameworks. Our findings offer valuable insights for developing efficient meta-learning algorithms. The main contributions can be summarized as follows:

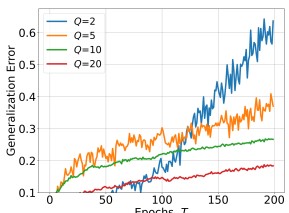

(b) Meta-MinibatchProx.

Figure 1: Effect of $Q$ on Omniglot dataset.

(1) We summarize six mainstream meta-learning algorithms and extract their structural features. Based on their inner-processes, we classify these algorithms into two frameworks: GDF and PDF, and develop two definitions of on-average stability, respectively. Accordingly, we establish a quantitative relationship between inner-levels and the generalization error in convex and non-convex settings.

(2) Our results reveal the influence of the inner-levels $Q$ on generalization error. In particular, we identify a trade-off relationship in GDF, whereas PDF demonstrates a beneficial relationship in its generalization bound. The primary reason for this difference lies in the term introduced by the inner-process. For example, in convex setting, the term for GDF, $\mathcal{O}(\frac{TQ}{mn^{\text{tr}}})$, increases with $Q$, whereas the term for PDF, $\mathcal{O}(\frac{T}{mC^Q})$, decreases with $Q$. These findings help to design a more efficient inner-process of meta-learning.

(3) Based on the generalization results of GDF and PDF, we further derive the generalization bounds for six meta-learning algorithms and analyze their implications. In general, note that the meta-objective $F(w)$ plays a crucial role in reducing the generalization bound. Motivated by this, we propose a new meta-objective $F_{\text{new}}(w)$ and prove $F_{\text{new}}(w) \leq F(w)$, thereby enhancing generalization performance. Extensive experiments confirm the efficiency of the proposed objective.

## 2 Related Work

**Algorithm Stability.** Algorithmic stability is critical for learnability [24]. There are two main approaches to investigating the stability-based generalization bound of meta-learning: (1) The first approach, introduced by [25], focuses on deriving generalization bounds for the transfer error, based on the assumption of independent task environments. This line of research has been further developed in several influential works [12, 26, 13, 17, 27, 28]. In this approach, the generalization error is defined as the transfer error minus the training error. (2) The second approach, by quantifying the test error, successfully eliminates the assumption of independent task environments and provides tighter generalization bounds compared to the first approach [16]. Additionally, [23] re-examined the generalization performance of meta-learning from a bi-level perspective, i.e., the inner-level and outer-level, and concluded that inverted regularization at the inner level helps to reduce the generalization bound. Differently, we make a further development in the second group by re-examining the generalization ability from two main structures at inner-level and reveal the two distinct impacts of inner-levels. Notably, the algorithm stability of prior work is limited in one-step MAML, highlighting the need for a novel algorithm stability definition.

**Meta-Learning.** Recent advances in meta-learning have spurred significant progress. From the optimization perspective, [29, 30] analyzed the convergence rate and computational complexity of ANIL, while [31] investigated the convergence properties with a single adaptation step. Extending this, [32] developed a theoretical framework for MAML with multiple inner-levels. [11] introduced a novel analysis of first-order meta-learning algorithms. To improve the in-context learning performance of pre-trained models, several meta-training-based approaches have been proposed. For example, MetaICT [33] uses BinaryCLFs and LAMA datasets to create tasks, while pre-pending human-generated instructions to each task. In contrast, MetaICL [34] leverages a wide variety of disjoint tasks to meta-train large language models. [35, 36] demonstrates that meta-learning can aid in cross-task generalization for prompt tuning. Moreover, MAML-en-LLM [37] is capable of learning truly generalizable parameters that not only perform well across disjoint tasks but also adapt effectively to unseen tasks.

## 3 Preliminaries

In this paper, each data point $z = (x, y) \in \mathcal{Z}$ consists of an input $x \in \mathcal{X}$ and its corresponding label $y \in \mathcal{Y}$. We assume access to $m$ tasks, denoted by $\mathcal{T}_1, \ldots, \mathcal{T}_m$ with the data for each task $\mathcal{T}_i, i \in [m]$, generated from an unknown distribution $\mathcal{P}_i$. We use the loss function $\ell : \mathbb{R}^d \times \mathcal{Z} \to \mathbb{R}^+$ to evaluate the performance of a model parameterized by $w \in \mathcal{W}$, where $\mathcal{W}$ is a closed subset of $\mathbb{R}^d$. The population loss corresponding for task $\mathcal{T}_i$ is defined as $\mathcal{L}_i(w) := \mathbb{E}_{z \sim \mathcal{P}_i}[\ell(w, z)]$. In addition, we use the notation $\widehat{\mathcal{L}}_i(w)$ to denote the empirical loss for task dataset $\mathcal{D}_i$, i.e., $\widehat{\mathcal{L}}_i(w, \mathcal{D}_i) := \frac{1}{|\mathcal{D}_i|} \sum_{z \in \mathcal{D}_i} \ell(w, z)$, where $|\mathcal{D}_i|$ is the size of $\mathcal{D}_i$. The goal of meta-learning is to learn good meta-parameters $w$ that perform wells across different tasks, which can be written as follows:

$$\min_{w \in \mathcal{W}} F(w) := \frac{1}{m} \sum_{i=1}^{m} F_i(w), \tag{1}$$

where $F_i(w)$ is used to denote the performance of $w$ on task $\mathcal{T}_i$, after undergoing multiple (or one) gradient updates to adapt the parameters to the specific task. At a high level, a.k.a., the outer-level, the learner needs to figure out useful meta-information that can generalize across tasks. At the inner-level, the learner needs to find task-specific parameters $w_{\mathcal{T}_i}$ that perform well on individual tasks after undergoing an inner-process.

### 3.1 The GDF Framework

From the inner-level perspective, two distinct inner-processes have primarily been developed, giving rise to two dominant meta-learning frameworks, as illustrated in Table 1. One prominent meta-learning framework, called the Gradient Descent Framework (GDF), is exemplified by MAML [5], FO-MAML [5], and Meta-SGD [6]. In particular, the inner-process finds the task-specific optimal hypothesis by solving the following optimization problem:

$$\min_{w_{\mathcal{T}_i}} \left\langle \sum_{q=0}^{Q-1} \nabla \widehat{\mathcal{L}}_i(w_{\mathcal{T}_i}^q, \mathcal{D}_i), w_{\mathcal{T}_i} - w \right\rangle + \frac{1}{2\alpha} \|w_{\mathcal{T}_i} - w\|_2^2, \tag{2}$$

**Algorithm 1** GDF and PDF

1: The set of datasets $\mathcal{S} = \{S_i\}_{i=1}^m$, outer iterations $T$, inner-levels $Q$, regulation $\lambda$.
2: Choose arbitrary initial point $w^0 \in W$;
3: **for** $t = 0$ **to** $T - 1$ **do**
4:     Randomly choose the task $i$.
5:     Inner-Level: $w_{\mathcal{T}_i,0}^t = w_t$
6:     **for** $q = 0, 1, ..., Q - 1$ **do**
7:         $w_{\mathcal{T}_i,q+1}^t = w_{\mathcal{T}_i,q}^t - \alpha\nabla\widehat{\mathcal{L}}_i(w_{\mathcal{T}_i,q}^t, S_i^{\mathrm{tr}})$ ;
8:         $w_{\mathcal{T}_i,q+1}^t = w_{\mathcal{T}_i,q}^t - \alpha\nabla\widehat{\mathcal{K}}_i(w_{\mathcal{T}_i,q}^t, S_i)$ ;
9:     **end for**
10:    $w^{t+1} := w^t - \eta_t\nabla_w\widehat{\mathcal{L}}_i(w_{\mathcal{T}_i,Q}^t, S_i^{\mathrm{ts}})$
11:    $w^{t+1} := w^t - \eta_t\lambda(w^t - w_{\mathcal{T}_i,Q}^t)$
12: **end for**
13: $w^T$ and $\overline{w}^T := \frac{1}{T+1}\sum_{t=0}^T w^t$;

where $Q$ is the number of inner-levels, i.e., gradient descent steps in GDF and $\alpha$ is the inner step size. In fact, by taking the derivative on $w_{\mathcal{T}_i}$ and making the gradient equal to 0, (2) can be written as $w_{\mathcal{T}_i} = w - \alpha\sum_{q=0}^{Q-1}\nabla\widehat{\mathcal{L}}_i(w_{\mathcal{T}_i}^q, \mathcal{D}_i)$, which reveals that task-specific parameters are iteratively learned by minimizing the empirical loss based using gradient descent. Specifically, in GDF, without loss of generality, we mainly follow MAML [5], and summarize the training procedure in Algorithm 1. For each task $\mathcal{T}_i$, we group the samples into two distinct sets among $n$ training samples: a support set $S_i^{\mathrm{tr}}$ of size $n^{\mathrm{tr}}$ for meta-train at inner-level and a query set $S_i^{\mathrm{ts}}$ of size $n^{\mathrm{ts}}$ for meta-validation at outer-level. Then, for each task $\mathcal{T}_i$, we have one corresponding training set $S_i := \{S_i^{\mathrm{tr}}, S_i^{\mathrm{ts}}\}$ in GDF. Referring back to (1), the formal definition of the meta-loss of task $\mathcal{T}_i$ is:

$$F_i(w) := \mathbb{E}_{\mathcal{D}_i}\mathbb{E}_{z\in\mathcal{P}_i}[\ell(w_{\mathcal{T}_i}^Q(w, \mathcal{D}_i), z]. \tag{3}$$

Because it is difficult to obtain the (3) in practical applications, we usually use the empirical loss $\widehat{F}_i(w, S_i) := \widehat{\mathcal{L}}(w_{\mathcal{T}_i}^Q(w, S_i^{\mathrm{tr}}), S_i^{\mathrm{ts}}) = \frac{1}{n^{\mathrm{ts}}}\sum_{z\in S_i^{\mathrm{ts}}}\ell(w_{\mathcal{T}_i}^Q(w, S_i^{\mathrm{tr}}), z)$ to approximate it.

### 3.2 The PDF Framework

Another widely used framework of meta-learning, known as the Proximal Descent Framework (PDF), is based on a "proximal" descent, wherein task-specific parameters are iteratively learned by minimizing the empirical loss and an $\ell_2$ regularizer. This framework is exemplified by iMAML [20], Meta-MinibatchProx [7], Fo-MuML [11]. In this framework, the inner-level optimization problem is formulated as follows:

$$\min_{w_{\mathcal{T}_i}}\widehat{\mathcal{L}}_i(w_{\mathcal{T}_i}, \mathcal{D}_i) + \frac{\lambda}{2}\|w_{\mathcal{T}_i} - w\|^2, \tag{4}$$

where $\lambda \geq 0$ is a regularization constant. In PDF, without loss of generality, we mainly follow Meta-MinibatchProx [7]. In particular, the task-specific training set $S_i$ of size $n$ is used directly for meta-train at inner-level, without the data into two distinct sets. Then, we present the following formulation of the meta-loss for task $\mathcal{T}_i$ corresponding to the PDF:

$$F_i(w) := \min_{w_{\mathcal{T}_i}}\left\{\mathbb{E}_{z\sim\mathcal{P}_i}\ell(w_{\mathcal{T}_i}, z) + \frac{\lambda}{2}\|w_{\mathcal{T}_i} - w\|^2\right\} = \mathbb{E}_{z\sim\mathcal{P}_i}\ell_\lambda(w, z). \tag{5}$$

and the empirical loss is $\widehat{F}_i(w, S_i) := \min_{w_{\mathcal{T}_i}}\{\frac{1}{n}\sum_{z\in S_i}\ell(w_{\mathcal{T}_i}, z) + \frac{\lambda}{2}\|w_{\mathcal{T}_i} - w\|^2\} = \widehat{\mathcal{L}}_\lambda(w, S_i)$. Note that we use $\mathbb{E}_{z\sim\mathcal{P}_i}\ell_\lambda(w, z)$ and $\widehat{\mathcal{L}}_\lambda(w, S_i)$ forlicity without impacting our analysis.

For a practical PDF-based algorithm, it's difficult to get an exact solution of $\widehat{\mathcal{L}}_\lambda(w, S_i)$, thereby we usually turn to get the inexact solution, $\widehat{\mathcal{K}}(w, S_i) = \frac{1}{n}\sum_{z\in S_i}\ell(w_{\mathcal{T}_i}, z) + \frac{\lambda}{2}\|w_{\mathcal{T}_i} - w\|^2$. Instead of solving (1), we solve its sample average surrogate problem as $\arg\min_{w\in\mathcal{W}}\widehat{F}(w, \mathcal{S}) := \frac{1}{m}\sum_{i=1}^m\widehat{F}_i(w, S_i)$, in which each $\widehat{F}_i$ is calculated by its corresponding empirical loss.

Specifically, by comparing (2) and (4), we observe that GDF only uses the first-order information of $\widehat{\mathcal{L}}_i(w_{\mathcal{T}_i}, \mathcal{D}_i)$, leading to an inexact solution. In contrast, PDF focuses on optimizing $\widehat{\mathcal{L}}_i(w_{\mathcal{T}_i}, \mathcal{D}_i) = \left\langle \nabla \widehat{\mathcal{L}}_i(w_{\mathcal{T}_i}, \mathcal{D}_i), w_{\mathcal{T}_i} - w \right\rangle + \frac{1}{2} \left\langle \nabla^2 \widehat{\mathcal{L}}_i(w_{\mathcal{T}_i}, \mathcal{D}_i), (w_{\mathcal{T}_i} - w)^{\otimes^2} \right\rangle + \frac{1}{6} \left\langle \nabla^3 \widehat{\mathcal{L}}_i(w_{\mathcal{T}_i}, \mathcal{D}_i), (w_{\mathcal{T}_i} - w)^{\otimes^3} \right\rangle + \cdots$, and thus implicitly leveraging higher-order information. This finding suggests that fewer inner-levels $Q$ may be more suitable for GDF, while more inner-levels $Q$ are better suited for PDF.

### 3.3 Stability and Generalization

Test error is generally considered the most critical metric for evaluating the performance of meta-learning algorithms. To control it, most studies focus on two key perspectives: generalization error and optimization error [38–41]. Specifically, let $\mathcal{S} := \{S_i\}_{i=1}^m$ be the concatenation of all tasks data sets. Given a randomized optimization algorithm $\mathcal{A}$ that acts on the dataset $\mathcal{S}$ and produces an output $\mathcal{A}(\mathcal{S})$, the generalization error is formally defined as $\epsilon_{gen} := \mathbb{E}_{\mathcal{A},\mathcal{S}}[F(\mathcal{A}(\mathcal{S})) - \widehat{F}(\mathcal{A}(\mathcal{S}), \mathcal{S})]$ and the optimization error is $\epsilon_{\text{opt}} := \mathbb{E}_{\mathcal{A},\mathcal{S}}[\widehat{F}(\mathcal{A}(\mathcal{S}), \mathcal{S}) - \min_{\mathcal{W}} \widehat{F}(\cdot, \mathcal{S})]$ [42, 43]. Then the test error of $\mathcal{A}$ can be decomposed into three distinct terms:

$$\mathbb{E}_{\mathcal{A},\mathcal{S}}\left[F(\mathcal{A}(\mathcal{S})) - \min_{\mathcal{W}} F\right] = \epsilon_{\text{gen}} + \epsilon_{\text{opt}} + \underbrace{\mathbb{E}_{\mathcal{S}}\left[\min_{\mathcal{W}} \widehat{F}(\cdot, \mathcal{S})\right] - \min_{\mathcal{W}} F}_{\leq 0}. \tag{6}$$

[32, 31] have shown that $\epsilon_{\text{opt}}$ will converge to 0 as the number of outer iterations $T$ increases, given that the loss function $\ell(w, z)$ satisfies certain assumptions, and the third term is non-positive. As such, analyzing $\epsilon_{\text{gen}}$ to improve the performance of the test error is more crucial. Note that [16, 23] only provide $\epsilon_{\text{gen}}$ of GDF in the strongly-convex setting. To fill the gap, in this paper, we establish a comprehensive theoretical analysis of two meta-learning frameworks in convex and non-convex settings. Before presenting our results, we state the following definition and assumptions widely used in generalization analysis [43, 16, 32].

*Definition* 1. We say a function $\ell(w)$ is $\lambda$-strongly convex if $\forall w_1, w_2, \ell(w_1) \geq \ell(w_2) + \langle \nabla \ell(w_2), w_1 - w_2 \rangle + \frac{\lambda}{2}\|w_1 - w_2\|^2$. If $\lambda = 0$, then we say $\ell(w)$ is convex.

**Assumption 1.** *We assume $\mathcal{Z}$ is a polish space (i.e., complete, separable, and metric) and $\mathcal{F}_{\mathcal{Z}}$ is the Borel $\wp$-algebra over $\mathcal{Z}$. Moreover, for any $i$, $\mathcal{P}_i$ is a non-atomic probability distribution over $(\mathcal{Z}, \mathcal{F}_{\mathcal{Z}})$, i.e., $\mathcal{P}_i(z) = 0$ for every $z \in \mathcal{Z}$.*

**Assumption 2.** *For any $z \in \mathcal{Z}$, the function $\ell(\cdot, z)$ is twice continuously differentiable. Furthermore, we assume it satisfies the following properties for any $w, u \in \mathbb{R}^d$.*

*(i) The loss is G-Lipschitz over $\mathbb{R}^d$, i.e., $\|\ell(w, z) - \ell(u, z)\| \leq G\|w - u\|$;*

*(ii) The loss is L-smooth over $\mathbb{R}^d$, i.e., $\|\nabla \ell(w, z) - \nabla \ell(u, z)\| \leq L\|w - u\|$;*

*(iii) The second derivative is Lipschitz continuous with constant $\rho$ over $\mathbb{R}^d$, i.e., $\|\nabla^2 \ell(w, z) - \nabla^2 \ell(u, z)\| \leq \rho\|w - u\|$;*

*(iv) The third derivative is Lipschitz continuous with constant $\kappa$ over $\mathbb{R}^d$, i.e., $\|\nabla^3 \ell(w, z) - \nabla^3 \ell(u, z)\| \leq \kappa\|w - u\|$.*

**Assumption 3.** *For any $i \in [m]$ and any $w \in \mathbb{R}^d$, the stochastic gradients $\nabla \widehat{F}_i(w, S)$ have bounded variance, i.e., $\mathbb{E}_S\|\nabla \hat{F}_i(w, S) - \nabla F(w)\|^2 \leq \sigma^2$.*

Stability-based generalization error analysis has been widely used to characterize the generalization properties for optimization algorithms such as stochastic gradient [43], adversarial training [44], and federated learning [45, 46]. Next, we will establish two different stability definitions and then leverage them to prove the generalization bound for GDF and PDF, respectively.

## 4 Theoretical Analysis

Under stability analysis, [16] improves the previous generalization error bound in the homogeneous case and provides a total variation-based analysis in the heterogeneous case. In addition, [23],

focusing on the inner-process, demonstrates that introducing inverted regularization at the inner-level can enhance generalization performance. However, their analyses are limited to GDF with one inner-level update ($Q = 1$), which ignores the effect of multiple inner-levels ($Q > 1$). To fully characterize the effect of multiple inner-level updates, a different stability definition is required.

*Definition* 2. (on-average stability for GDF). A randomized algorithm $\mathcal{A}$ with output $w_{\mathcal{S}}$ is called $\epsilon$-on-average stable if the following condition holds for any $i \in [m]$: Take the dataset $\widetilde{\mathcal{S}}$ which is the same as $\mathcal{S}$, except that $\widetilde{S}_{i,k}^{\text{tr}}$ and $\widetilde{S}_{i,j}^{\text{ts}}$ differ from $S_i^{\text{tr}}$ and $S_i^{\text{ts}}$ by replacing the $k$-th and $j$-th data points, respectively, where $k \in [n^{\text{tr}}], j \in [n^{\text{ts}}]$. Then, we have: $\max_{k \in [n^{\text{tr}}], j \in [n^{\text{ts}}]} \mathbb{E}_{\mathcal{S},\mathcal{A},\widetilde{S}_{i,k}^{\text{tr}},\widetilde{z}_{i,j}^{\text{ts}}} [\ell(w_{\mathcal{T}_i}(w_{\mathcal{S}}, \widetilde{S}_{i,k}^{\text{tr}}), \widetilde{z}_{i,j}^{\text{ts}}) - \ell(w_{\mathcal{T}_i}(w_{\widetilde{\mathcal{S}}}, \widetilde{S}_{i,k}^{\text{tr}}), \widetilde{z}_{i,j}^{\text{ts}})] \leq \epsilon$.

Different with the stability definition for stochastic gradient descent in [47, 48] which considers only a single-level structure, our definitions for meta-learning mean any perturbation of samples across levels cannot lead to a big change of the model trained by an algorithm in expectation. The main reason that we are interested in the stability of an algorithm is its connection with generalization error. Next, we formalize this connection for GDF and show that whether an Algorithm $\mathcal{A}$ is $\epsilon$-on-average stable, then its generalization error is bounded above by $\epsilon$.

**Theorem 1.** *Consider the population risk $F(w)$ and empirical risk $\widehat{F}(w)$. The corresponding $F_i(w)$ and $\widehat{F}_i(w)$ are approached by GDF in Algorithm 1. Under Assumption 1 and Definition 2, if $\mathcal{A}$ is a randomized GDF-based algorithm, then $\epsilon_{\text{gen}} \leq \mathbb{E}_{\mathcal{A},\mathcal{S}} \left[ F(w_{\mathcal{S}}) - \widehat{F}(w_{\mathcal{S}}, \mathcal{S}) \right] \leq \epsilon$.*

Building on the GDF analysis, we now extend this concept to the PDF setting, where the stability definition needs to account for differences in training procedures.

*Definition* 3. (on-average stability for PDF). A randomized algorithm $\mathcal{A}$ with output $w_{\mathcal{S}}$ is called $\epsilon$-on-average stable if the following condition holds for any $i \in [m]$: Take the dataset $\widetilde{\mathcal{S}}$ which is the same as $\mathcal{S}$, except that $\widetilde{S}_{i,j}$ differ from $S_i$ by replacing the $j$-th data point. Then, for any $\tilde{z} \in \mathcal{Z}$, we have: $\max_{j \in [n]} \mathbb{E}_{\mathcal{S},\mathcal{A},\widetilde{z}_{i,j}} [\ell_\lambda(w_{\mathcal{S}}, \widetilde{z}_{i,j}) - \ell_\lambda(w_{\widetilde{\mathcal{S}}}, \widetilde{z}_{i,j})] \leq \epsilon$, where $\ell_\lambda$ is defined in (5).

Under this definition, we can establish a connection between generalization and stability. Although similar to Theorem 1, i.e., $\epsilon_{\text{gen}} \leq \mathbb{E}_{\mathcal{A},\mathcal{S}}[F(w_{\mathcal{S}}) - \widehat{F}(w_{\mathcal{S}}, \mathcal{S})] \leq \epsilon$, the proof of this is significantly different from it. In addition, $F(w)$ and $\widehat{F}(w)$ in PDF also differ from those in GDF, as composed of different $F_i(w)$ and $\widehat{F}_i(w)$, respectively.

Based on the above theorems, we will present the generalization bounds under convex and non-convex.

## 4.1 Generalization bounds of GDF

**Theorem 2.** *Let the outer-level step size and inner-level step size be chosen as $\eta_t \leq \frac{1}{L_Q}$ and $\alpha \leq \frac{1}{L}$, respectively. Under Assumptions 1- 3, the generalization error $\epsilon_{\text{gen}}$ of GDF can be bounded by:*

$$\mathcal{O}\left( \sum_{t=0}^{T-1} \eta_t (1+\alpha L)^{Q-1} \frac{(6QG + Q^2\alpha^2 G^2\rho)}{mn^{\text{tr}}} + \frac{1}{m}\sqrt{F(w^0) - \min_{\mathcal{W}} F + \frac{L_Q\sigma^2}{2}\sum_{t=0}^{T-1}\eta_t^2} \right),$$

*where $L_Q = \alpha\rho Q(1+\alpha L)^{Q-1} + (1+\alpha L)^Q L$. If we further let $\alpha \leq \frac{1}{LQ}$ and $\eta_t = \eta$ be fixed, we can obtain a concise result as follows:*

$$\epsilon_{\text{gen}} \leq \mathcal{O}\left( \frac{TQ}{mn^{\text{tr}}} + \frac{1}{m}\sqrt{F(w^0) - \min_{\mathcal{W}} F + T} \right).$$

*Remark* 1. The first observation from the above results is that the generalization bound worsens as $Q$ increases, primarily due to the first term, $\mathcal{O}(\frac{TQ}{mn^{\text{tr}}})$. Similarly, [15] provides a comparable bound, $\mathcal{O}(\sqrt{T + TQ})$, using the information-theoretic analysis. However, by revisiting (1) and (3), we observe that $F(w)$ is influenced by $w_{\mathcal{T}_i}^Q(w, \mathcal{D}_i)$. Consequently, after $Q$ steps of gradient descent, the divergence between $F(w^0)$ and $\min_{\mathcal{W}} F$ decreases, which in turn positively impacts the generalization bound. In summary, our findings indicate a trade-off in selecting the number of inner steps $Q$, which also explains the phenomenon in Figure 1a.

*Remark* 2. In previous work, [9] also provides a similar trade-off relationship, $\mathcal{O}(\frac{(1+\alpha L)^Q}{n} + F(w^T) - \min_{\mathcal{W}} F)$, with $\alpha \leq \frac{1}{L}$. However, under the choice $\alpha \leq \frac{1}{QL}$, the first term simplifies to $\mathcal{O}(\frac{1}{n})$,

effectively mitigating the trade-off relationship. In contrast, our result demonstrates greater robustness to the choice of $\alpha$. In fact, we can consider a special case to understant the robutst relationship. For sufficiently large $Q$, $\alpha \leq \frac{1}{QL}$ yields negligibly small learning rate, implying the inner update will be close to initialisation $w^t_{\tau_i,0}$, which means the model fails to learn task-specific information effectively, leading to larger generalization error. Furthermore, unlike the generalization bound $\mathcal{O}(\frac{1}{mn^{\mathrm{tr}}})$ provided by [16], our generalization bound grows with $T$ due to our consideration of a general convex setting. In contrast, their analysis assumes a strongly convex setting, which is consistent with [43].

**Theorem 3.** *Let the outer-level step size be chosen as $\eta_t = \frac{c}{t}$ satisfy $c \leq \min\{\frac{1}{L_Q}, \frac{1}{4(2L_Q \ln(T))^2}\}$ and the inner-level step size be chosen as $\alpha \leq \frac{1}{QL}$. Under Assumptions 1-3, and further assume that $\mathbb{E}_{\mathcal{S},\mathcal{A}}[F(w_{\mathcal{S}})] \leq F(w^0)$, the generalization error $\epsilon_{\mathrm{gen}}$ of GDF can be bounded by:*

$$\mathcal{O}\left(\left(\frac{1+\frac{1}{c\gamma}}{m}\Phi_Q\right)^{\frac{1}{c\gamma}}\left(F(w^0)T\right)^{\frac{c\gamma}{1+c\gamma}}\right),$$

*where $\Phi_Q = 1 + \frac{Q}{n^{\mathrm{tr}}}$, $L_Q = \frac{3\rho(1+\alpha L)^{2(Q-1)}}{L} + (1+\alpha L)^Q L$, $\rho_Q = \frac{3\rho(1+\alpha)(1+\alpha L)^{2(Q-1)}}{L} + (1+\alpha L)^{3Q}\rho + \alpha\kappa(1+\alpha L)^{2Q}$ and $\gamma = \min\{L_Q, \mathbb{E}_S[\|\nabla^2 \widehat{F}_i(w^0)\|] + \rho_Q(c\sigma + \sqrt{c(F(w^0) - \min_{\mathcal{W}} F)})\}$.*

*Remark* 3. Note that we find that the trade-off relationship also exists in the non-convex setting, which is similar to Theorem 2. Specifically, on the one hand, we observe that $\epsilon_{\mathrm{gen}}$ increases with $\Phi_Q = 1 + \frac{Q}{n^{\mathrm{tr}}}$. On the other hand, $\epsilon_{\mathrm{gen}}$ decreases with $F(w^0)$, where $F(w^0)$ itself decrease with $Q$. In addition, $\gamma$ also characterizes the effect of $Q$ on $\epsilon_{\mathrm{gen}}$ through the term $\rho_Q(c\sigma + \sqrt{c(F(w^0) - \min_{\mathcal{W}} F)})$. Here, $\rho_Q$ grows with $Q$, while $F(w^0) - \min_{\mathcal{W}} F$ decreases with $Q$. Importantly, an inappropriate choice of the inner step size $\alpha$ can exacerbate the generalization error. Even when the standard loss $\ell(w,z)$ is $L$-smoothness and has $\rho$-Lipschitz hessian, the compositional loss $\ell(w^Q_{\mathcal{T}_i}(w, S^{\mathrm{tr}}_i), z)$ may become $L_Q$-smoothness and $\rho_Q$-Lipschitz, with $L_Q$ and $\rho_Q$ decreasing with $Q$. However, under our principled choice of $\alpha \leq \frac{1}{QL}$, we can ensure $L_Q \leq \mathcal{O}(1)$ and $\rho_Q \leq \mathcal{O}(1)$, thereby mitigating these negative effects.

## 4.2 Generalization bounds of PDF

**Theorem 4.** *Let the outer-level step size be chosen as $\eta_t \leq \frac{1}{L_Q}$, the inner-level step size be fixed and $C > 1$ being a constant. Under Assumptions 2-3, the generalization error $\epsilon_{\mathrm{gen}}$ of PDF can be bounded by:*

$$\mathcal{O}\left(\sum_{t=0}^{T-1}\frac{2\eta_t\lambda}{mC^Q} + \frac{1}{m}\sqrt{F(w^0) - \min_{\mathcal{W}} F + \frac{L_Q\sigma^2}{2}\sum_{t=0}^{T-1}\eta_t^2}\right),$$

*where $L_Q = \frac{\lambda L}{\lambda + L}$. If we further let $\eta_t = \eta \leq \frac{1}{L_Q}$ be a constant, we can obtain a more concise result as follows:*

$$\epsilon_{\mathrm{gen}} \leq \mathcal{O}\left(\frac{T}{mC^Q} + \frac{1}{m}\sqrt{F(w^0) - \min_{\mathcal{W}} F + T}\right).$$

*Remark* 4. An immediate conclusion from the first term, $\frac{T}{mC^Q}$, in the above result is that the generalization error in PDF decreases as the adaption steps $Q$ increase. Additionally, the second term, $\frac{1}{m}\sqrt{F(w^0) - \min_{\mathcal{W}} F + T}$), also benefits from $Q$ for similar reasons discussed in Remark 1. It is worth mentioning that, in a strong-convex setting, the generalization error does not grow with the number of iterations $T$ [43]. However, the objective in (5) that we are minimizing is actually strongly convex w.r.t $w_{\mathcal{T}_i}$ (due to the strongly-convex regularizer, $\frac{\lambda}{2}\|w_{\mathcal{T}_i} - w\|^2$ and convex function $\ell(\cdot, z)$), but only convex w.r.t $w$ [49, 50, 40]. As a result, our generalization bound still grows with $T$. Furthermore, if we let $Q = n^{\mathrm{tr}}$, the generalization error of GDF becomes $\mathcal{O}(\frac{T}{m} + \frac{1}{m}\sqrt{F(w^0) - \min_{\mathcal{W}} F + T})$, which is greater than the generalization error of PDF, $\mathcal{O}(\frac{T}{mC^{n^{\mathrm{tr}}}} + \frac{1}{m}\sqrt{F(w^0) - \min_{\mathcal{W}} F + T})$.

**Theorem 5.** *Let the outer-level step size be chosen as $\eta_t = \frac{c}{t}$ satisfy $c \leq \min\{\frac{1}{L_Q}, \frac{1}{4(2L_Q \ln(T))^2}\}$, inner-level step size be fixed and $C > 1$ being a constant. Under Assumptions 2-3, and further*

Table 2: Summary of our results.

| Frame. | Algorithm | Convex | Non-convex |
|---|---|---|---|
| GDF | MAML | $\mathcal{O}(\sum_{t=0}^{T-1}\eta_t(1+\alpha L)^{Q-1}\frac{(6QG+Q^2\alpha^2G^2\rho)}{mn^{tr}}+\frac{\mathcal{Q}(F(w^0))}{m})$ | $\mathcal{O}(\frac{1+\frac{1}{c\gamma}}{m}(1+(1+\alpha L)^{2(Q-1)}\frac{(6QG+Q^2\alpha^2G^2\rho)}{n^{tr}}))^{\frac{1}{c\gamma}}(F(w^0)T)^{\frac{c\gamma}{1+c\gamma}})$ |
| | FOMAML | $\mathcal{O}(\sum_{t=0}^{T-1}\eta_t\frac{2Q\alpha LG}{mn^{tr}}+\frac{\mathcal{Q}(F(w^0))}{m})$ | $\mathcal{O}(\frac{1+\frac{1}{c\gamma}}{m}(1+\frac{2Q(1+\alpha L)^Q\alpha LG}{n^{tr}}))^{\frac{1}{c\gamma}}(F(w^0)T)^{\frac{c\gamma}{1+c\gamma}})$ |
| | Meta-SGD | $\mathcal{O}(\sum_{t=0}^{T-1}\eta_t(1+\widehat{\alpha}_tL)^{Q-1}\frac{(6QG+Q^2\widehat{\alpha}_t^2G^2\rho)}{mn^{tr}}+\frac{\mathcal{Q}(F(w^0))}{m})$ | $\mathcal{O}(\frac{1+\frac{1}{c\gamma}}{m}(1+(1+\widehat{\alpha}L)^{2(Q-1)}\frac{(6QG+Q^2\widehat{\alpha}^2G^2\rho)}{n^{tr}}))^{\frac{1}{c\gamma}}(F(w^0)T)^{\frac{c\gamma}{1+c\gamma}})$ |
| PDF | iMAML | $\mathcal{O}(\sum_{t=0}^{T-1}\frac{2L\eta_t(G^2+G)}{\lambda mn^{tr}}+\frac{\mathcal{Q}(F(w^0))}{m})$ | $\mathcal{O}(\frac{1+\frac{1}{c\gamma}}{m}(1+\frac{2L(G^2+G)}{(\lambda-L)n^{tr}}))^{\frac{1}{c\gamma}}(F(w^0)T)^{\frac{c\gamma}{1+c\gamma}})$ |
| | Meta-MinibatchProx | $\mathcal{O}(\sum_{t=0}^{T-1}\frac{2\eta_t\lambda}{mCQ}+\frac{\mathcal{Q}(F(w^0))}{m})$ | $\mathcal{O}(\frac{1+\frac{1}{c\gamma}}{m}(1+\frac{G(\lambda+L)}{CQ}))^{\frac{1}{c\gamma}}(F(w^0)T)^{\frac{c\gamma}{1+c\gamma}})$ |
| | Fo-MuML | $\mathcal{O}(\sum_{t=0}^{T-1}\frac{2\eta_t G^2}{\lambda mn^{tr}}+\frac{\mathcal{Q}(F(w^0))}{m})$ | $\mathcal{O}(\frac{1+\frac{1}{c\gamma}}{m}(1+\frac{2G^2}{(\lambda-L)n^{tr}}))^{\frac{1}{c\gamma}}(F(w^0)T)^{\frac{c\gamma}{1+c\gamma}})$ |

assume that $\mathbb{E}_{\mathcal{S},\mathcal{A}}[F(w_{\mathcal{S}})] \leq F(w^0)$ , the generalization error $\epsilon_{\text{gen}}$ of PDF can be bounded by:

$$\mathcal{O}\left((\frac{1+\frac{1}{c\gamma}}{m}\Phi_Q)^{\frac{1}{c\gamma}}\left(F(w^0)T\right)^{\frac{c\gamma}{1+c\gamma}}\right),$$

where $L_Q = \frac{\lambda L}{\lambda+L}$, $\gamma = \min\{L_Q, \mathbb{E}_S[\|\nabla^2\widehat{F}_i(w^0)\|] + \rho_Q(c\sigma + \sqrt{c(F(w^0)-\min_{\mathcal{W}}F)})\}$, $\Phi_Q = 1+\frac{1}{C^Q}$ and $\rho_Q = \rho$.

*Remark* 5. Similarly, we get a lower generalization error since $\Phi_Q = 1+\frac{1}{C^Q}$ and $F(w^0)$ gets smaller when $Q$ increases. In addition, we observe that $L_Q = \frac{\lambda L}{\lambda+L}$ and $\rho_Q = \rho$ are independent of $Q$, because PDF eliminates the compositional structure of $\ell(w_{\mathcal{T}_i}(w, S_i^{\text{tr}}), z)$ in GDF. Furthermore, if we set $Q = n^{\text{tr}}$, we can get $\Phi_Q = 2$ in GDF, whereas $\Phi_Q = 1+\frac{1}{C^{n^{\text{tr}}}}$ in PDF, which indicate that the generalization error of PDF is lower than GDF.

*Remark* 6. Based on Theorems 2-5, we present the generalization bounds for the corresponding algorithms, summarized in Table 2. Here, the term $\mathcal{Q}(F(w^0)) = \sqrt{F(w^0)-\min_{\mathcal{W}}F + \frac{L_Q\sigma^2}{2}\sum_{t=0}^{T-1}\eta_t^2}$ decreases as $Q$ increases. Specifically, Meta-SGD redefines $\alpha$ as a learnable parameter, denoted by $\widehat{\alpha}_t$. As discussed in Remarks 2-3, the choice of the inner step size $\alpha$ is critical, where $\alpha \leq \frac{1}{QL}$ is necessary to mitigate potential adverse effects caused by the inner process. In addition, our results reveal that algorithms based on the GDF framework exhibit a trade-off between the inner process and generalization error, as shown in Theorems 2-3. In contrast, the PDF framework avoids the adverse effects of the adaptation process by implicitly leveraging higher-order information. It is important to emphasize that our analysis focuses on the relationship between $Q$ and $\epsilon_{\text{gen}}$. However, directly comparing $\epsilon_{\text{gen}}$ across algorithms remains challenging due to the influence of algorithm-specific terms, such as $L_Q$ and $\rho_Q$.

## 4.3 New Optimization Objective

Recalling from the Theorems 2-5 and the results in Table 2, we can observe that $F(w^0)$ plays a critical role in reducing the generalization bound, due to the terms, $F(w^0) - \min_{\mathcal{W}}F$ and $F(w^0)T$. To reduce the impact of $F(w^0)$, note that $F(w^0) = \frac{1}{m}\sum_{i=1}^m F_i(w^0)$, then let us consider a simple relaxation technique: given a constant $\beta \in [0,1)$ and $\hat{m} \leq \frac{m}{2}$ with the losses sorted by $\{F_i(w^0)\}$ in *ascending* order, we can obtain that $\sum_{i=1}^{\hat{m}} F_i(w^0)^{\psi(i)} \leq \sum_{i=\hat{m}+1}^m F_i(w^0)^{\psi(i)}$, where $\psi(i)$ is a mapping function associating the index with the original loss $F_i(w^0)$. Based on this, we can derive $F(w^0) = \frac{1}{m}\sum_{i=1}^m F_i(w^0)^{\psi(i)} \geq \frac{1+\beta}{m}\sum_{i=1}^{\hat{m}} F_i(w^0)^{\psi(i)} + \frac{1-\beta}{m}\sum_{i=\hat{m}+1}^m F_i(w^0)^{\psi(i)}$. Motivated by this result, we propose the following optimization objective:

$$F_{\text{new}}(w) = \frac{1+\beta}{\tilde{m}}\sum_{i=1}^m F_i(w)^{\psi(i)} + \frac{1-\beta}{\tilde{m}}\sum_{i=\hat{m}+1}^m F_i(w)^{\psi(i)}, \qquad (7)$$

where $\tilde{m} = (1+\beta)\hat{m} + (1-\beta)(m-\hat{m})$ is to ensure the lower bound derivation. A clear advantage of our new optimization objective is its ability to achieve a smaller initial value, $F_{\text{new}}(w^0)$, thereby reducing the generalization error. In addition, this new optimization objective is designed to easily integrate with all types of algorithms in both the GDF and PDF frameworks without requiring modification of their training procedures. During the training phase, we usually select a batch $\mathcal{B}$ with size $B$. After performing the inner-process on each task, we evaluate their task losses $\widehat{\mathcal{L}}_i(w_{\mathcal{T}_i}^t), i \in [B]$, at the outer-level and sort the losses in *ascending* order, yielding $\{\widehat{\mathcal{L}}_i(w_{\mathcal{T}_i}^t)\}^{\psi(i)}$. Then, based on the sorted losses, we then use a weighting factor $\beta$ to adjust the weight of each

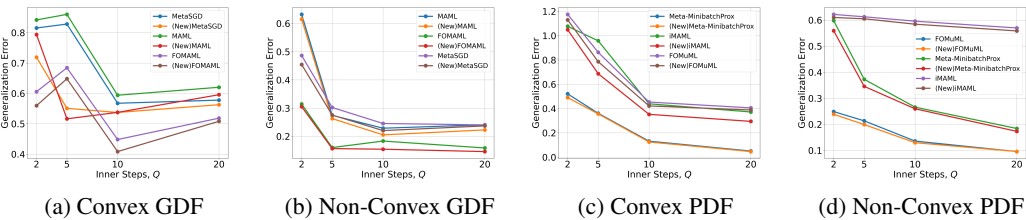

| (a) Convex GDF | (b) Non-Convex GDF | (c) Convex PDF | (d) Non-Convex PDF |

Figure 2: Relationship between generalization errors and $Q$.

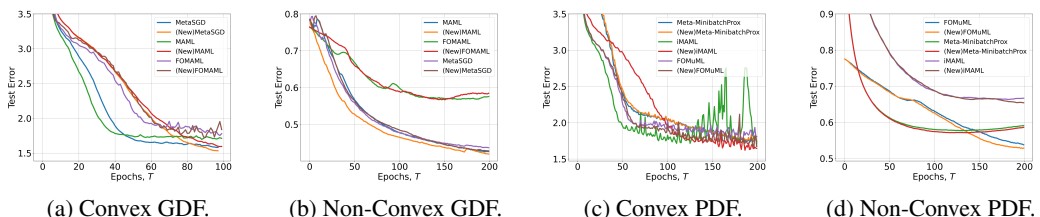

| (a) Convex GDF. | (b) Non-Convex GDF. | (c) Convex PDF. | (d) Non-Convex PDF. |

Figure 3: Convergence results with or without our new objective.

task when updating the meta-model $w^t$. Importantly, our strategy achieves a dynamical fairness for each task without compromising convergence performance. The design most similar to our new optimization goal is [51], however, their goal is to maximize the meta-loss as much as possible to obtain task-robust meta-parameters, whereas our goal is to reduce the original meta-loss, and our approach is easier to practice.

## 5 Experiment

**Few-shot regression.** This problem focuses on approximating a family of sine functions represented as $f(x) = a\sin(bx)$. The task distribution, denoted as $\mathcal{P}$, corresponds to the joint distribution $p(a, b)$, where $a \sim U[0.1, 5]$ is the amplitude and $b \sim U[0, \pi]$ is the phase. Both the training and test tasks are randomly generated from $\mathcal{P} = p(a, b)$. Following [7], we use an MLP network with Mean square error loss function. The generalization error is assessed as the difference between training and test errors. For each training task, we set the number of support samples and query samples to $n^{\mathrm{tr}} = 5$ and $n^{\mathrm{ts}} = 5$, respectively, while for each test task, we use $n^{\mathrm{tr}} = 5, n^{\mathrm{ts}} = 15$.

**Few-shot classification.** We follow the standard experimental setup described in [22] using the real-world Omniglot dataset, which comprises 1,623 characters from 50 different alphabets, with each character having 20 instances drawn by different individuals. We employ a $3 \times 3$ CNN to align with [5, 12], and use the Cross-Entropy Loss as the loss function. For each task, the number of support samples and query samples to $n^{\mathrm{tr}} = 1$ and $n^{\mathrm{ts}} = 5$, respectively. For each test task, we use $n^{\mathrm{tr}} = 1, n^{\mathrm{ts}} = 15$. In addition, each task is formulated as a 5-way classification problem.

We fix the training task number $m = 100$ and generate 10000 new tasks at test time by using the standard library [52]. To ensure a fair comparison, We report the average generalization error during the last 10 iterations of GDF and PDF under convex and non-convex settings. As shown in Figure 2, GDF's generalization ability benefits considerably from smaller inner-levels $Q$, but deteriorates when $Q$ becomes larger. In contrast, PDF's generalization improves with increasing $Q$. These observations align with our findings in Table 2. Under our new optimization Objective, the generalization error of different algorithms can be effectively reduced. Furthermore, we also compare the convergence performance of GDF and PDF with or without our new objective within the same number of epochs. In particular, we set $Q = 5$ for the convex setting and $Q = 10$ for the non-convex setting. In Figure 3, the results indicate that our new optimization objective leads to lower test errors for most algorithms.

# 6    Conclusion

In this paper, we introduce two theoretical frameworks, GDF and PDF. They are summarized by several popular meta-learning algorithms based on inner processes. Within these frameworks, we derive generalization upper bounds for both convex and non-convex settings, which offer a detailed understanding of how the number of inner-levels influences the generalization error of GDF and PDF. Our analysis reveals a trade-off relationship induced by the inner-level in GDF, while PDF demonstrates a more favorable relationship that improves generalization. Building on these insights, we propose a novel meta-objective designed to significantly reduce generalization error. Extensive experiments validate the effectiveness of our findings and the proposed objective. We believe that the insights, the proof techniques, and the new meta-objective can inspire further research and open new directions in meta-learning and related areas.

## Acknowledgment

This work was supported by the Project of Xiangjiang Laboratory (24XJJCYJ01003), the National Natural Science Foundation of China (62302525, 62202293, 62303306, 62302524, 62372472, 62306109, 62320106006), the Hunan Province Natural Science Foundation of China (2024JJ6527, 2024JJ6531, 2024JJ4068), and the Guizhou Provincial Science and Technology Projects (No.[2025]022). It also utilized computing resources at Central South University's High Performance Computing Center of Central South University.

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

# A    Useful Lemmas

**Lemma 1.** *Let $\phi$ be a convex and L-smooth function. Then, for $\eta \leq \frac{1}{L}$, we have*
$$\|(w - \eta\nabla\phi(w)) - (u - \eta\nabla\phi(u))\| \leq \|w - u\|,$$
*for any $w$ and $u$.*

*Proof.* Since $\phi$ is $L$-smooth and convex, we know that
$$\langle \nabla\phi(w) - \nabla\phi(u), w - u \rangle \geq \frac{1}{\eta}\|\nabla\phi(w) - \nabla\phi(u)\|^2.$$

Using this fact,
$$
\begin{aligned}
\|(w - \eta\nabla\phi(w)) - (u - \eta\nabla\phi(u))\|^2 &= \|w - u - \eta(\nabla\phi(w) - \nabla\phi(u))\|^2 \\
&= \|w - u\|^2 + \eta^2\|\nabla\phi(w) - \nabla\phi(u)\|^2 - \eta\langle\nabla\phi(w) - \nabla\phi(u), w - u\rangle \\
&\leq \|w - u\|^2 + \eta(\eta - L^{-1})\|\nabla\phi(w) - \nabla\phi(u)\|^2 \\
&\leq \|w - u\|^2
\end{aligned}
$$
when $\eta \leq \frac{1}{\gamma}$. $\qquad\qquad\square$

**Lemma 2.** *[43] Let $\phi$ be a $\lambda$-strongly convex and L-smooth function. Then, for any $\delta \leq \frac{2}{\lambda + L}$, we have*
$$\|(u - \delta\nabla\phi(u)) - (v - \delta\nabla\phi(v))\| \leq (1 - \frac{L\delta\lambda}{\lambda + L})\|u - v\|$$
*for any $u$ and $v$.*

**Lemma 3.** *Let $f_\lambda(w) = \min_u \left( f(u) + \frac{\lambda}{2}\|w - u\|^2 \right)$, if $f$ is G-Lipschitz, then we have $\|\nabla f_\lambda(w)\| \leq 2G$.*

*Proof.* Define $\phi(u) \triangleq f(u) + \frac{\lambda}{2}\|v - w\|^2$ and $v^* = \arg\min_u \phi(u)$. Now, observe that
$$0 \leq \phi(w) - \phi(u^*) = f(w) - f(u^*) - \frac{\lambda}{2}\|w - u^*\|^2$$

Thus, we have
$$\frac{\lambda}{2}\|w - u^*\|^2 \leq f(w) - f(u^*) \leq L\|w - u^*\|$$
where the last inequality follows from the fact that $f$ is $G$-Lipschitz. Thus, we get $\|w - u^*\| \leq \frac{2G}{\lambda}$. Since $\|\nabla f_\lambda(w)\| = \lambda\|w - u^*\|$. This together with the above bound gives the desired result. $\quad\square$

# B    Stability and Generalization of MAML

### B.1    Proof of Theorem 1

To show the claim, it just suffices to show that for any $i$, we have
$$\mathbb{E}_{\mathcal{A},\mathcal{S}}\left[ F_i(w_\mathcal{S}) - \widehat{F}_i(w_\mathcal{S}, S_i) \right] \leq \epsilon.$$

Take the dataset $\widetilde{\mathcal{S}}$ which is the same as $\mathcal{S}$, except that $\widetilde{S}_{i,k}^{\text{tr}}$ and $\widetilde{S}_{i,j}^{\text{ts}}$ differ from $S_i^{\text{tr}}$ and $S_i^{\text{ts}}$ in at most one data point, respectively. In particular,
$$S_i^{\text{tr}} = \{z_{i,1}^{\text{tr}}, ..., z_{i,n^{\text{tr}}}^{\text{tr}}\}, S_i^{\text{ts}} = \{z_{i,1}^{\text{ts}}, ..., z_{i,n^{\text{ts}}}^{\text{ts}}\},$$
$$\widetilde{S}_{i,k}^{\text{tr}} = \{z_{i,1}^{\text{tr}}, ..., \widetilde{z}_{i,k}^{\text{tr}}, ..., z_{i,n^{\text{tr}}}^{\text{tr}}\}, \widetilde{S}_{i,j}^{\text{ts}} = \{z_{i,1}^{\text{ts}}, ..., \widetilde{z}_{i,j}^{\text{ts}}, ..., z_{i,n^{\text{tr}}}^{\text{ts}}\}.$$
Under Assumption 1, we could assume $\widetilde{z}_{i,j}^{\text{ts}}$ is different with $\widetilde{z}_{i,k}^{\text{tr}}$. Then, we relate empirical risk and population risk by

$$
\begin{aligned}
\mathbb{E}_{\mathcal{S},\mathcal{A}}[\widehat{F}_i(w_\mathcal{S}, S_i)] &= \frac{1}{n^{\text{ts}}}\sum_{j=1}^{n^{\text{ts}}} \mathbb{E}_{\mathcal{S},\mathcal{A}}[\ell\left(w_{\mathcal{T}_i}(w_\mathcal{S}, S_i^{\text{tr}}), z_{i,j}^{\text{ts}}\right)]. \\
&= \frac{1}{n^{\text{ts}}}\sum_{j=1}^{n^{\text{ts}}} \mathbb{E}_{\mathcal{S},\mathcal{A},\widetilde{S}_{i,k}^{\text{tr}},\widetilde{z}_{i,j}^{\text{ts}}}[\ell\left(w_{\mathcal{T}_i}(w_{\widetilde{\mathcal{S}}}, \widetilde{S}_{i,k}^{\text{tr}}), \widetilde{z}_{i,j}^{\text{ts}}\right)].
\end{aligned}
$$
(8)

Moreover, we have

$$\mathbb{E}_{\mathcal{A},\mathcal{S}}[F_i(w_{\mathcal{S}})] = \frac{1}{n^{\text{ts}}} \sum_{j=1}^{n^{\text{ts}}} \mathbb{E}_{\mathcal{S},\mathcal{A},\widetilde{S}_{i,k}^{\text{tr}},\widetilde{z}_{i,j}^{\text{ts}}}[\ell\left(w_{\mathcal{T}_i}(w_{\mathcal{S}}, \widetilde{S}_{i,k}^{\text{tr}}), \widetilde{z}_{i,j}^{\text{ts}}\right)]. \tag{9}$$

Putting (8) and (9) together, we have

$$\mathbb{E}_{\mathcal{A},\mathcal{S}}\left[F(w_{\mathcal{S}}) - \widehat{F}(w_{\mathcal{S}}, \mathcal{S}_i)\right] \leq \frac{1}{m} \sum_{i=1}^{m} \frac{1}{n^{\text{ts}}} \sum_{j=1}^{n^{\text{ts}}} \mathbb{E}_{\mathcal{A},\mathcal{S},\widetilde{S}_{i,k}^{\text{tr}},z_{i,j}^{\text{ts}}}[\ell(w_{\mathcal{T}_i}(w_{\mathcal{S}}, \widetilde{S}_{i,k}^{\text{tr}}), \widetilde{z}_{i,j}^{\text{ts}}) - \ell\left(w_{\mathcal{T}_i}(w_{\widetilde{\mathcal{S}}}, \widetilde{S}_{i,k}^{\text{tr}}), \widetilde{z}_{i,j}^{\text{ts}}\right)]$$

$$\leq \epsilon.$$

Then we obtain the desired result.

## B.2   Lemmas

In the following proofs, for simplicity, we use $\widehat{\mathcal{L}}(w_{\mathcal{T}_i,q}) = \widehat{\mathcal{L}}_i(w_{\mathcal{T}_i,q}, S_i^{\text{tr}}), \ell(w_{\mathcal{T}_i,q}) = \ell(w_{\mathcal{T}_i,q}, z)$ without other explnation, where $q \in [Q]$.

**Lemma 4.** *For any $i \in [m], q = 0, ..., Q - 1$ and $w, u \in \mathbb{R}^d$, if $\ell$ is a convex function, we have*

$$\|w_{\mathcal{T}_i,q+1} - u_{\mathcal{T}_i,q+1}\| \leq \|w - u\|$$

*Proof.* Based on the updates that $w_{\mathcal{T}_i,q+1} = w_{\mathcal{T}_i,q} - \alpha_q \nabla \widehat{\mathcal{L}}_i(w_{\mathcal{T}_i,q})$ and $u_{\mathcal{T}_i,q+1} = u_{\mathcal{T}_i,q} - \alpha_q \nabla \widehat{\mathcal{L}}_i(u_{\mathcal{T}_i,q})$, we obtain

$$\|w_{\mathcal{T}_i,q+1} - u_{\mathcal{T}_i,q+1}\| = \|(w_{\mathcal{T}_i,q} - \alpha_q \nabla \widehat{\mathcal{L}}_i(w_{\mathcal{T}_i,q})) - (u_{\mathcal{T}_i,q} - \alpha_q \nabla \widehat{\mathcal{L}}_i(u_{\mathcal{T}_i,q}))\|$$
$$\leq \|w_{\mathcal{T}_i,q} - u_{\mathcal{T}_i,q}\|$$

where we use the Lemma 1 in the first inequality. Unrolling it over $q$ from 0 to $q + 1$, we obtain

$$\|w_{\mathcal{T}_i,q+1} - u_{\mathcal{T}_i,q+1}\| \leq \|w - u\|$$

□

**Lemma 5.** *Suppose the conditions in Assumtpion 2 are satisfied. Then, if $\alpha \leq \frac{1}{L}$ and $\ell$ is a convex function, we have*

$$\|\ell(w_{\mathcal{T}_i}(w, S_i^{\text{tr}}), z) - \ell(w_{\mathcal{T}_i}(u, S_i^{\text{tr}}), z)\| \leq G\|w - u\|$$

*Proof.* For any $w, u \in \mathcal{W}$, note that

$$\|\ell(w_{\mathcal{T}_i}(w, S_i^{\text{tr}}), z) - \ell(w_{\mathcal{T}_i}(u, S_i^{\text{tr}}), z)\| \leq G\|w_{\mathcal{T}_i}(w, S_i^{\text{tr}}) - w_{\mathcal{T}_i}(u, S_i^{\text{tr}})\|$$
$$= G\|(w_{\mathcal{T}_i,Q-1} - \alpha \nabla \widehat{\mathcal{L}}(w_{\mathcal{T}_i,Q-1}, S_i^{\text{tr}}))$$
$$- (u_{\mathcal{T}_i,Q-1} - \alpha \nabla \widehat{\mathcal{L}}(u_{\mathcal{T}_i,Q-1}, S_i^{\text{tr}}))\|$$
$$\leq G\|w_{\mathcal{T}_i,Q-1} - u_{\mathcal{T}_i,Q-1}\|$$
$$\leq G\|w - u\|$$

where we use Lemma 1 in the second inequality and Lemma 4 in the last inequality.   □

**Lemma 6.** *For any $i \in [m], q = 0, ..., Q - 1$ and $w, u \in \mathbb{R}^d$, we have*

$$\|w_{\mathcal{T}_i,q+1} - u_{\mathcal{T}_i,q+1}\| \leq (1 + \alpha L)^{q+1}\|w - u\|$$

*Proof.* Based on the updates that $w_{\mathcal{T}_i,q+1} = w_{\mathcal{T}_i,q} - \alpha_q \nabla \widehat{\mathcal{L}}_i(w_{\mathcal{T}_i,q})$ and $u_{\mathcal{T}_i,q+1} = u_{\mathcal{T}_i,q} - \alpha_q \nabla \widehat{\mathcal{L}}_i(u_{\mathcal{T}_i,q})$, we obtain

$$\|w_{\mathcal{T}_i,q+1} - u_{\mathcal{T}_i,q+1}\| = \|(w_{\mathcal{T}_i,q} - \alpha_q \nabla \widehat{\mathcal{L}}_i(w_{\mathcal{T}_i,q})) - (u_{\mathcal{T}_i,q} - \alpha_q \nabla \widehat{\mathcal{L}}_i(u_{\mathcal{T}_i,q}))\|$$
$$\leq \|w_{\mathcal{T}_i,q} - u_{\mathcal{T}_i,q}\| + \alpha L\|w_{\mathcal{T}_i,q} - u_{\mathcal{T}_i,q}\|$$
$$= (1 + \alpha L)\|w_{\mathcal{T}_i,q} - u_{\mathcal{T}_i,q}\|$$

where we use the Assumption 2 in the first inequality. Unrolling it over $q$ from 0 to $q + 1$, we obtain

$$\|w_{\mathcal{T}_i,q+1} - u_{\mathcal{T}_i,q+1}\| \leq (1 + \alpha L)^{q+1}\|w - u\|$$

□

**Lemma 7.** *Suppose the conditions in Assumtpion 2 are satisfied. Then, with $\alpha \leq \frac{1}{QL}$ and $\ell$ is a non-convex function, we have*

$$\|\ell(w_{\mathcal{T}_i}(w, S_i^{\mathrm{tr}}), z) - \ell(w_{\mathcal{T}_i}(u, S_i^{\mathrm{tr}}), z)\| \leq eG\|w - u\|$$

*Proof.* For any $w, u \in \mathcal{W}$, note that

$$
\begin{aligned}
\|\ell(w_{\mathcal{T}_i}(w, S_i^{\mathrm{tr}}), z) - \ell(w_{\mathcal{T}_i}(u, S_i^{\mathrm{tr}}), z)\| &\leq G\|w_{\mathcal{T}_i}(w, S_i^{\mathrm{tr}}) - w_{\mathcal{T}_i}(u, S_i^{\mathrm{tr}})\| \\
&= G\|[w_{\mathcal{T}_i, Q-1} - \alpha \nabla \widehat{\mathcal{L}}(w_{\mathcal{T}_i, Q-1}, S_i^{\mathrm{tr}})] - [u_{Q-1} - \alpha \nabla \widehat{\mathcal{L}}(u_{\mathcal{T}_i, Q-1}, S_i^{\mathrm{tr}})]\| \\
&= G\|w_{\mathcal{T}_i, Q-1} - u_{\mathcal{T}_i, Q-1}\| + \alpha\|\nabla \widehat{\mathcal{L}}(w_{\mathcal{T}_i, Q-1}, S_i^{\mathrm{tr}}) - \nabla \widehat{\mathcal{L}}(u_{\mathcal{T}_i, Q-1}, S_i^{\mathrm{tr}})\| \\
&\leq G(1 + \alpha L)\|w_{\mathcal{T}_i, Q-1} - u_{\mathcal{T}_i, Q-1}\| \\
&\leq G(1 + \alpha L)^Q \|w - u\| \\
&\leq eG\|w - u\|
\end{aligned}
$$

where we use Assumption 2 in the first and second inequality, Lemma 6 in the third inequality. If $\alpha \leq \frac{1}{QL}$, we have the last inequality. $\qquad\square$

**Lemma 8.** *For any $i \in [m], q = 0, ..., Q$ and $w \in \mathbb{R}^d$, we have*

$$\|\nabla \widehat{\mathcal{L}}_i(w_{\mathcal{T}_i, q+1})\| \leq (1 + \alpha L)^q \|\nabla \widehat{\mathcal{L}}_i(w)\|.$$

*Proof.* Recalling the update rule $w_{\mathcal{T}_i, q+1} = w_{\mathcal{T}_i, q} - \alpha_q \nabla \widehat{\mathcal{L}}_i(w_{\mathcal{T}_i, q})$, Using Assumption 2, we have

$$
\begin{aligned}
\|\nabla \widehat{\mathcal{L}}_i(w_{\mathcal{T}_i, q+1})\| &= \|\nabla \widehat{\mathcal{L}}_i(w_{\mathcal{T}_i, q+1}) - \nabla \widehat{\mathcal{L}}_i(w_{\mathcal{T}_i, q}) + \nabla \widehat{\mathcal{L}}_i(w_{\mathcal{T}_i, q})\| \\
&\leq \|\nabla \widehat{\mathcal{L}}_i(w_{\mathcal{T}_i, q+1}) - \nabla \widehat{\mathcal{L}}_i(w_{\mathcal{T}_i, q})\| + \|\nabla \widehat{\mathcal{L}}_i(w_{\mathcal{T}_i, q})\| \\
&\leq L\|w_{\mathcal{T}_i, q+1} - w_{\mathcal{T}_i, q}\| + \|\nabla \widehat{\mathcal{L}}_i(w_{\mathcal{T}_i, q})\| \\
&= L\|w_{\mathcal{T}_i, q} - \alpha \nabla \widehat{\mathcal{L}}_i(w_{\mathcal{T}_i, q}) - w_{\mathcal{T}_i, q}\| + \|\nabla \widehat{\mathcal{L}}_i(w_{\mathcal{T}_i, q})\| \leq (1 + \alpha L)\|\nabla \widehat{\mathcal{L}}_i(w_{\mathcal{T}_i, q})\|
\end{aligned}
$$

where we use Assumption 2 in the second inequality. Then, unrolling the above inequality over $q$ from 0 to $q + 1$, we get

$$\|\nabla \widehat{\mathcal{L}}_i(w_{\mathcal{T}_i, q+1})\| \leq (1 + \alpha)^{q+1} \|\nabla \widehat{\mathcal{L}}_i(w)\|$$

$\qquad\square$

**Lemma 9.** *Suppose that Assumption 2 hold and the function $F_i(w)$ defined in Eq (3). Then, for any $w, u \in \mathbb{R}^d$, if $\ell$ is convex, we have*

$$\|\nabla \widehat{F}_i(w) - \nabla \widehat{F}_i(u)\| \leq L_Q \|w - u\|$$

*where $L_Q = \alpha \rho Q(1 + \alpha L)^{Q-1} + (1 + \alpha L)^Q L$ with $\alpha \leq \frac{1}{L}$.*

*Proof.* Similar to the proof of Lemma 9, we have

$$
\|\nabla\widehat{F}_i(w) - \nabla\widehat{F}_i(u)\|
$$

$$
=\|\prod_{q=0}^{Q-1}(I-\alpha\nabla^2\widehat{\mathcal{L}}_i(w_{\mathcal{T}_i,q}))\nabla\ell(w_{\mathcal{T}_i,Q}) - \prod_{q=0}^{Q-1}(I-\alpha\nabla^2\widehat{\mathcal{L}}_i(u_{\mathcal{T}_i,q}))\nabla\ell(u_{\mathcal{T}_i,Q})\|
$$

$$
\leq\|\prod_{q=0}^{Q-1}(I-\alpha\nabla^2\widehat{\mathcal{L}}_i(w_{\mathcal{T}_i,q}))\nabla\ell(w_{\mathcal{T}_i,Q}) - \prod_{j=0}^{Q-1}(I-\alpha\nabla^2\widehat{\mathcal{L}}_i(u_{\mathcal{T}_i,q}))\nabla\ell(w_{\mathcal{T}_i,Q})\|
$$

$$
+\|\prod_{q=0}^{Q-1}(I-\alpha\nabla^2\widehat{\mathcal{L}}_i(u_{\mathcal{T}_i,q}))\nabla\ell(w_{\mathcal{T}_i,Q}) - \prod_{j=0}^{Q-1}(I-\alpha\nabla^2\widehat{\mathcal{L}}_i(u_{\mathcal{T}_i,q}))\nabla\ell(u_{\mathcal{T}_i,Q})\|
$$

$$
\leq\|\prod_{q=0}^{Q-1}(I-\alpha\nabla^2\widehat{\mathcal{L}}_i(w_{\mathcal{T}_i,q})) - \prod_{q=0}^{Q-1}(I-\alpha\nabla^2\widehat{\mathcal{L}}_i(u_{\mathcal{T}_i,q}))\|\|\nabla\ell(w_{\mathcal{T}_i,Q})\|
$$

$$
+\|\prod_{q=0}^{Q-1}(I-\alpha\nabla^2\widehat{\mathcal{L}}_i(u_{\mathcal{T}_i,q}))\|\|\nabla\ell(w_{\mathcal{T}_i,Q}) - \nabla\ell(u_{\mathcal{T}_i,Q})\|
$$

$$
\leq\|\prod_{q=0}^{Q-1}(I-\alpha\nabla^2\widehat{\mathcal{L}}_i(w_{\mathcal{T}_i,q})) - \prod_{q=0}^{Q-1}(I-\alpha\nabla^2\widehat{\mathcal{L}}_i(u_{\mathcal{T}_i,q}))\|\|\nabla\ell(w_{\mathcal{T}_i,Q})\| + (1+\alpha L)^Q\|\nabla\ell(w_{\mathcal{T}_i,Q}) - \nabla\ell(u_{\mathcal{T}_i,Q})\|
$$

$$
\leq\|\prod_{q=0}^{Q-1}(I-\alpha\nabla^2\widehat{\mathcal{L}}_i(w_{\mathcal{T}_i,q})) - \prod_{q=0}^{Q-1}(I-\alpha\nabla^2\widehat{\mathcal{L}}_i(u_{\mathcal{T}_i,q}))\|\|\nabla\ell(w_{\mathcal{T}_i,Q})\| + (1+\alpha L)^Q L\|w_{\mathcal{T}_i,Q} - u_{\mathcal{T}_i,Q}\|
$$

$$
\leq\underbrace{\|\prod_{q=0}^{Q-1}(I-\alpha\nabla^2\widehat{\mathcal{L}}_i(w_{\mathcal{T}_i,q})) - \prod_{q=0}^{Q-1}(I-\alpha\nabla^2\widehat{\mathcal{L}}_i(u_{\mathcal{T}_i,q}))\|}_{\Lambda(Q-1)} G + (1+\alpha L)^Q L\|w-u\|,
$$

(10)

where in the last inequality, we use Assumption 2 and Lemma 4. We next upper-bound $\Lambda$ in the above inequality. Specifically, we have

$$
\Lambda(Q-1)\leq\|\prod_{q=0}^{Q-2}(I-\alpha\nabla^2\widehat{\mathcal{L}}_i(w_{\mathcal{T}_i,q}))(I-\alpha\nabla^2\widehat{\mathcal{L}}_i(w_{\mathcal{T}_i,Q-1})) - \prod_{q=0}^{Q-2}(I-\alpha\nabla^2\widehat{\mathcal{L}}_i(w_{\mathcal{T}_i,q}))(I-\alpha\nabla^2\widehat{\mathcal{L}}_i(u_{\mathcal{T}_i,Q-1}))\|
$$

$$
+\|\prod_{q=0}^{Q-2}(I-\alpha\nabla^2\widehat{\mathcal{L}}_i(w_{\mathcal{T}_i,q}))(I-\alpha\nabla^2\widehat{\mathcal{L}}_i(u_{\mathcal{T}_i,Q-1})) - \prod_{q=0}^{Q-2}(I-\alpha^2\nabla^2\widehat{\mathcal{L}}_i(u_{\mathcal{T}_i,q}))(I-\alpha\nabla^2\widehat{\mathcal{L}}_i(u_{\mathcal{T}_i,Q-1}))\|
$$

$$
\leq\|\prod_{q=0}^{Q-2}(I-\alpha\nabla^2\widehat{\mathcal{L}}_i(w_{\mathcal{T}_i,q}))\|\|\alpha\nabla^2\widehat{\mathcal{L}}_i(w_{\mathcal{T}_i,Q-1}) - \alpha\nabla^2\widehat{\mathcal{L}}_i(u_{\mathcal{T}_i,Q-1})\|
$$

$$
+\|\prod_{q=0}^{Q-2}(I-\alpha\nabla^2\widehat{\mathcal{L}}_i(w_{\mathcal{T}_i,q})) - \prod_{q=0}^{Q-2}(I-\alpha\nabla^2\widehat{\mathcal{L}}_i(u_{\mathcal{T}_i,q}))\|\|(I-\alpha\nabla^2\widehat{\mathcal{L}}_i(u_{\mathcal{T}_i,Q-1}))\|
$$

$$
\leq(1+\alpha L)^{Q-1}\|\alpha\nabla^2\widehat{\mathcal{L}}_i(w_{\mathcal{T}_i,Q-1}) - \alpha\nabla^2\widehat{\mathcal{L}}_i(u_{\mathcal{T}_i,Q-1})\|
$$

$$
+(1+\alpha L)\|\prod_{q=0}^{Q-2}(I-\alpha\nabla^2\widehat{\mathcal{L}}_i(w_{\mathcal{T}_i,q})) - \prod_{q=0}^{Q-2}(I-\alpha\nabla^2\widehat{\mathcal{L}}_i(u_{\mathcal{T}_i,q}))\|
$$

$$
\leq(1+\alpha L)^{Q-1}\alpha\rho\|w_{\mathcal{T}_i,Q-1} - u_{\mathcal{T}_i,Q-1}\| + (1+\alpha L)\Lambda(Q-2)
$$

$$
\leq(1+\alpha L)^{Q-1}\alpha\rho\|w-u\| + (1+\alpha L)\Lambda(Q-2),
$$

where we use Assumption 2 in the second inequality and Lemma 4 in the last inequality. Telescoping the above inequality over $q$ from 1 to $Q-1$ and noting $\Lambda(0) \leq \alpha\rho\|w - u\|$, we have

$$
\begin{aligned}
\Lambda(Q-1) \leq &(1+\alpha L)^{Q-1}\Lambda(0) + \alpha\rho(Q-1)(1+\alpha L)^{Q-1}\|w - u\| \\
\leq &(1+\alpha L)^{Q-1}\alpha\rho\|w - u\| + \alpha\rho(Q-1)(1+\alpha L)^{Q-1}\|w - u\| \\
\leq &\alpha\rho Q(1+\alpha L)^{Q-1}\|w - u\|
\end{aligned}
\tag{11}
$$

Substituting (11) into (10) and let $L_Q = \alpha\rho Q(1+\alpha L)^{Q-1} + (1+\alpha L)^Q L$, we get

$$
\|\nabla F_i(w) - \nabla F_i(u)\| \leq L_Q\|w - u\|,
$$

which completes the proof. $\qquad\square$

**Lemma 10.** *Suppose that Assumption 2 hold and the function $F_i(w)$ defined in Eq (3). Then, for any $w, u \in \mathbb{R}^d$, if $\ell$ is non-convex, we have*

$$
\|\nabla\widehat{F}_i(w) - \nabla\widehat{F}_i(u)\| \leq L_Q\|w - u\|
$$

*where $L_Q = \frac{3\rho(1+\alpha L)^{2(Q-1)}}{L} + (1+\alpha L)^Q L$.*

*Proof.* Similar to the proof of Lemma 9, we have

$$
\begin{aligned}
&\|\nabla\widehat{F}_i(w) - \nabla\widehat{F}_i(u)\| \\
\leq &\Big\|\prod_{q=0}^{Q-1}(I - \alpha\nabla^2\widehat{\mathcal{L}}_i(w_{\mathcal{T}_i,q})) - \prod_{q=0}^{Q-1}(I - \alpha\nabla^2\widehat{\mathcal{L}}_i(u_{\mathcal{T}_i,q}))\Big\|\|\nabla\ell(w_{\mathcal{T}_i,Q})\| + (1+\alpha L)^Q L\|w_{\mathcal{T}_i,Q} - u_{\mathcal{T}_i,Q}\| \\
\leq &\underbrace{\Big\|\prod_{q=0}^{Q-1}(I - \alpha\nabla^2\widehat{\mathcal{L}}_i(w_{\mathcal{T}_i,q})) - \prod_{q=0}^{Q-1}(I - \alpha\nabla^2\widehat{\mathcal{L}}_i(u_{\mathcal{T}_i,q}))\Big\|}_{\Lambda(Q-1)} G + (1+\alpha L)^Q L\|w - u\|,
\end{aligned}
\tag{12}
$$

where in the last inequality, we use Assumption 2 and Lemma 4. We next upper-bound $\Lambda$ in the above inequality which is also similar to the proof of (11). Specifically, we have

$$
\begin{aligned}
\Lambda(Q-1) \leq &(1+\alpha L)^{Q-1}\Lambda(0) + \sum_{l=0}^{Q-2}\alpha\rho(1+\alpha L)^{2(Q-1-l)+l}\|w - u\| \\
\leq &(1+\alpha L)^{Q-1}\alpha\rho\|w - u\| + \sum_{l=0}^{Q-2}\alpha\rho(1+\alpha L)^{2(Q-1-l)+l}\|w - u\| \\
\leq &\left[(1+\alpha L)^{Q-1}\alpha\rho + \alpha\rho(1+\alpha L)^Q \sum_{l=0}^{Q-2}(1+\alpha L)^l\right]\|w - u\| \\
\leq &\left[(1+\alpha L)^{Q-1}\alpha\rho + \frac{\rho}{L}(1+\alpha L)^Q((1+\alpha L)^{Q-1} - 1)\right]\|w - u\| \\
\leq &\frac{3\rho(1+\alpha L)^{2(Q-1)}}{L}\|w - u\|
\end{aligned}
\tag{13}
$$

where we use $\alpha \leq \frac{1}{L}$ in the last inequality. Substituting (13) into (12) and let $L_Q = \frac{3\rho(1+\alpha L)^{2(Q-1)}}{L} + (1+\alpha L)^Q L$, we get

$$
\|\nabla F_i(w) - \nabla F_i(u)\| \leq L_Q\|w - u\|,
$$

which completes the proof. $\qquad\square$

**Lemma 11.** *Suppose that Assumption 2 hold and the function $F_i(w)$ defined in Eq (3). Then, for any $w, u \in \mathbb{R}^d$, if $\ell$ is convex, we have*

$$
\|\nabla^2 F_i(w) - \nabla^2 F_i(u)\| \leq \rho_Q\|w - u\|
$$

*where $\rho_Q = (1+\alpha L)^{2Q-1}\left[\alpha\rho Q + 2\rho + Q\alpha\rho + G\alpha\kappa + G\alpha^2\rho^2 Q\right]$*

*Proof.* For simplicity, we denote $J_q = I - \alpha \nabla^2 \widehat{\mathcal{L}}_i(w_{\mathcal{T}_i,q})$.

$$\|\nabla^2 \widehat{F}_i(w) - \nabla^2 \widehat{F}_i(u)\|$$

$$= \| \prod_{q=0}^{Q-1} (I - \alpha \nabla^2 \widehat{\mathcal{L}}_i(w_{\mathcal{T}_i,q}))^2 \nabla^2 \ell(w_{\mathcal{T}_i,Q}) + \sum_{q=0}^{Q-1} \left( \prod_{l=0}^{q-1} J_l \right) \frac{\partial J_q}{\partial w} \left( \prod_{l=q+1}^{Q-1} J_l \right) \nabla \ell(w_{\mathcal{T}_i,Q})$$

$$- \prod_{q=0}^{Q-1} (I - \alpha \nabla^2 \widehat{\mathcal{L}}_i(u_{\mathcal{T}_i,q}))^2 \nabla^2 \ell(u_{\mathcal{T}_i,Q}) - \sum_{q=0}^{Q-1} \left( \prod_{l=0}^{q-1} J_l \right) \frac{\partial J_q}{\partial u} \left( \prod_{l=q+1}^{Q-1} J_l \right) \nabla \ell(u_{\mathcal{T}_i,Q}) \|$$

$$\leq \| \prod_{q=0}^{Q-1} (I - \alpha \nabla^2 \widehat{\mathcal{L}}_i(w_{\mathcal{T}_i,q}))^2 \nabla^2 \ell(w_{\mathcal{T}_i,Q}) - \prod_{q=0}^{Q-1} (I - \alpha \nabla^2 \widehat{\mathcal{L}}_i(u_{\mathcal{T}_i,q}))^2 \nabla^2 \ell(u_{\mathcal{T}_i,Q}) \|$$

$$+ \| \sum_{q=0}^{Q-1} \left( \prod_{l=0}^{q-1} J_l \right) \frac{\partial J_q}{\partial w} \left( \prod_{l=q+1}^{Q-1} J_l \right) \nabla \ell(w_{\mathcal{T}_i,Q}) - \sum_{q=0}^{Q-1} \left( \prod_{l=0}^{q-1} J_l \right) \frac{\partial J_q}{\partial u} \left( \prod_{l=q+1}^{Q-1} J_l \right) \nabla \ell(u_{\mathcal{T}_i,Q}) \|$$

$$\tag{14}$$

For the first term, we have

$$\| \prod_{q=0}^{Q-1} (I - \alpha \nabla^2 \widehat{\mathcal{L}}_i(w_{\mathcal{T}_i,q}))^2 \nabla^2 \ell(w_{\mathcal{T}_i,Q}) - \prod_{q=0}^{Q-1} (I - \alpha \nabla^2 \widehat{\mathcal{L}}_i(u_{\mathcal{T}_i,q}))^2 \nabla^2 \ell(u_{\mathcal{T}_i,Q}) \|$$

$$\leq \| \prod_{q=0}^{Q-1} (I - \alpha \nabla^2 \widehat{\mathcal{L}}_i(w_{\mathcal{T}_i,q}))^2 \nabla^2 \ell(w_{\mathcal{T}_i,Q}) - \prod_{q=0}^{Q-1} (I - \alpha \nabla^2 \widehat{\mathcal{L}}_i(u_{\mathcal{T}_i,q}))^2 \nabla^2 \ell(w_{\mathcal{T}_i,Q}) \|$$

$$+ \| \prod_{q=0}^{Q-1} (I - \alpha \nabla^2 \widehat{\mathcal{L}}_i(u_{\mathcal{T}_i,q}))^2 \nabla^2 \ell(w_{\mathcal{T}_i,Q}) - \prod_{q=0}^{Q-1} (I - \alpha \nabla^2 \widehat{\mathcal{L}}_i(u_{\mathcal{T}_i,q}))^2 \nabla^2 \ell(u_{\mathcal{T}_i,Q}) \|$$

$$\leq \| \prod_{q=0}^{Q-1} (I - \alpha \nabla^2 \widehat{\mathcal{L}}_i(w_{\mathcal{T}_i,q}))^2 - \prod_{q=0}^{Q-1} (I - \alpha \nabla^2 \widehat{\mathcal{L}}_i(u_{\mathcal{T}_i,q}))^2 \| \| \nabla^2 \ell(w_{\mathcal{T}_i,Q}) \|$$

$$\tag{15}$$

$$+ \| \prod_{q=0}^{Q-1} (I - \alpha \nabla^2 \widehat{\mathcal{L}}_i(u_{\mathcal{T}_i,q}))^2 \| \| \nabla^2 \ell(w_{\mathcal{T}_i,Q}) - \nabla^2 \ell(u_{\mathcal{T}_i,Q}) \|$$

$$\leq \underbrace{\| \prod_{q=0}^{Q-1} (I - \alpha \nabla^2 \widehat{\mathcal{L}}_i(w_{\mathcal{T}_i,q}))^2 - \prod_{q=0}^{Q-1} (I - \alpha \nabla^2 \widehat{\mathcal{L}}_i(u_{\mathcal{T}_i,q}))^2 \|}_{\Lambda(Q-1)} L + (1 + \alpha L)^{2Q} \rho \|w - u\|$$

where we use Assumption 2 and Lemma 4 in the last inequality. We next upper-bound $\Lambda(Q-1)$ in the above inequality. Specifically, we have

$$
\begin{aligned}
\Lambda(Q-1) \leq & \| \prod_{q=0}^{Q-1}(I - \alpha\nabla^2\widehat{\mathcal{L}}_i(w_{\mathcal{T}_i,q}))^2 - \prod_{q=0}^{Q-1}(I - \alpha\nabla^2\widehat{\mathcal{L}}_i(w_{\mathcal{T}_i,q}))(I - \alpha\nabla^2\widehat{\mathcal{L}}_i(u_{\mathcal{T}_i,q}))\| \\
& + \| \prod_{q=0}^{Q-1}(I - \alpha\nabla^2\widehat{\mathcal{L}}_i(w_{\mathcal{T}_i,q}))(I - \alpha\nabla^2\widehat{\mathcal{L}}_i(u_{\mathcal{T}_i,q})) - \prod_{q=0}^{Q-1}(I - \alpha\nabla^2\widehat{\mathcal{L}}_i(u_{\mathcal{T}_i,q}))^2\| \\
\leq & \| \prod_{q=0}^{Q-1}(I - \alpha\nabla^2\widehat{\mathcal{L}}_i(w_{\mathcal{T}_i,q}))\|\| \prod_{q=0}^{Q-1}(I - \alpha\nabla^2\widehat{\mathcal{L}}_i(w_{\mathcal{T}_i,q})) - \prod_{q=0}^{Q-1}(I - \alpha\nabla^2\widehat{\mathcal{L}}_i(u_{\mathcal{T}_i,q}))\| \\
& + \| \prod_{q=0}^{Q-1}(I - \alpha\nabla^2\widehat{\mathcal{L}}_i(u_{\mathcal{T}_i,q}))\|\| \prod_{q=0}^{Q-1}(I - \alpha\nabla^2\widehat{\mathcal{L}}_i(w_{\mathcal{T}_i,q})) - \prod_{q=0}^{Q-1}(I - \alpha\nabla^2\widehat{\mathcal{L}}_i(u_{\mathcal{T}_i,q}))\| \\
\leq & (1 + \alpha L)^Q \| \prod_{q=0}^{Q-1}(I - \alpha\nabla^2\widehat{\mathcal{L}}_i(w_{\mathcal{T}_i,q})) - \prod_{q=0}^{Q-1}(I - \alpha\nabla^2\widehat{\mathcal{L}}_i(u_{\mathcal{T}_i,q}))\| \\
\leq & (1 + \alpha L)^Q \alpha\rho Q(1 + \alpha L)^{Q-1}\|w - u\|.
\end{aligned}
\tag{16}
$$

where we use (11) in the last inequality. Putting (16) into (15), we have

$$
\| \prod_{q=0}^{Q-1}(I - \alpha\nabla^2\widehat{\mathcal{L}}_i(w_{\mathcal{T}_i,q}))^2\nabla^2\ell(w_{\mathcal{T}_i,Q}) - \prod_{q=0}^{Q-1}(I - \alpha\nabla^2\widehat{\mathcal{L}}_i(u_{\mathcal{T}_i,q}))^2\nabla^2\ell(u_{\mathcal{T}_i,Q})\|
\tag{17}
$$
$$
\leq \left[(1 + \alpha L)^Q \alpha\rho Q(1 + \alpha L)^{Q-1} + (1 + \alpha L)^{2Q}\rho\right]\|w - u\|
$$

To bound the second term in (14), we have

$$
\begin{aligned}
& \| \sum_{q=0}^{Q-1}\left(\prod_{l=0}^{q-1}J_l\right)\frac{\partial J_q}{\partial w}\left(\prod_{l=q+1}^{Q-1}J_l\right)\nabla\ell(w_{\mathcal{T}_i,Q}) - \sum_{q=0}^{Q-1}\left(\prod_{l=0}^{q-1}J_l\right)\frac{\partial J_q}{\partial u}\left(\prod_{l=q+1}^{Q-1}J_l\right)\nabla\ell(u_{\mathcal{T}_i,Q})\| \\
\leq & \| \sum_{q=0}^{Q-1}\left(\prod_{l=0}^{q-1}J_l\right)\frac{\partial J_q}{\partial w}\left(\prod_{l=q+1}^{Q-1}J_l\right)\nabla\ell(w_{\mathcal{T}_i,Q}) - \sum_{q=0}^{Q-1}\left(\prod_{l=0}^{q-1}J_l\right)\frac{\partial J_q}{\partial w}\left(\prod_{l=q+1}^{Q-1}J_l\right)\nabla\ell(u_{\mathcal{T}_i,Q})\| \\
& + \| \sum_{q=0}^{Q-1}\left(\prod_{l=0}^{q-1}J_l\right)\frac{\partial J_q}{\partial w}\left(\prod_{l=q+1}^{Q-1}J_l\right)\nabla\ell(u_{\mathcal{T}_i,Q}) - \sum_{q=0}^{Q-1}\left(\prod_{l=0}^{q-1}J_l\right)\frac{\partial J_q}{\partial u}\left(\prod_{l=q+1}^{Q-1}J_l\right)\nabla\ell(u_{\mathcal{T}_i,Q})\| \\
\leq & \| \sum_{q=0}^{Q-1}\left(\prod_{l=0}^{q-1}J_l\right)\frac{\partial J_q}{\partial w}\left(\prod_{l=q+1}^{Q-1}J_l\right)\|\|\nabla\ell(w_{\mathcal{T}_i,Q}) - \nabla\ell(u_{\mathcal{T}_i,Q})\| \\
& + \| \sum_{q=0}^{Q-1}\left(\prod_{l=0}^{q-1}J_l\right)\frac{\partial J_q}{\partial w}\left(\prod_{l=q+1}^{Q-1}J_l\right) - \sum_{q=0}^{Q-1}\left(\prod_{l=0}^{q-1}J_l\right)\frac{\partial J_q}{\partial u}\left(\prod_{l=q+1}^{Q-1}J_l\right)\|\|\nabla\ell(u_{\mathcal{T}_i,Q})\| \\
\leq & \sum_{q=0}^{Q-1}\|\left(\prod_{l=0}^{q-1}J_l\right)\frac{\partial J_q}{\partial w}\left(\prod_{l=q+1}^{Q-1}J_l\right)\|\|w - u\| + \sum_{q=0}^{Q-1}\|\left(\prod_{l=0}^{q-1}J_l\right)\frac{\partial J_q}{\partial w}\left(\prod_{l=q+1}^{Q-1}J_l\right) - \left(\prod_{l=0}^{q-1}J_l\right)\frac{\partial J_q}{\partial u}\left(\prod_{l=q+1}^{Q-1}J_l\right)\|G \\
\leq & \sum_{q=0}^{Q-1}(1 + \alpha L)^{Q-1}\|\frac{\partial J_q}{\partial w}\|\|w - u\| + \sum_{q=0}^{Q-1}(1 + \alpha L)^{Q-1}G\|\frac{\partial J_q}{\partial w} - \frac{\partial J_q}{\partial u}\|.
\end{aligned}
\tag{18}
$$

Firstly, we have

$$
\begin{aligned}
\|\frac{\partial J_q}{\partial w}\| = & \|\alpha\nabla^3\widehat{\mathcal{L}}(w_{\mathcal{T}_i,q})\prod_{l=0}^{q-1}(I - \alpha\nabla^2\widehat{\mathcal{L}}(w_{\mathcal{T}_i,q}))\| \\
\leq & (1 + \alpha L)^{q-1}\alpha\rho,
\end{aligned}
\tag{19}
$$

where we use Assumption 2. Secondly, we have

$$
\begin{aligned}
\|\frac{\partial J_q}{\partial w} - \frac{\partial J_q}{\partial u}\| =& \|\alpha \nabla^3 \widehat{\mathcal{L}}(w_{\mathcal{T}_i,q}) \prod_{l=0}^{q-1}(I - \alpha \nabla^2 \widehat{\mathcal{L}}(w_{\mathcal{T}_i,q})) - \alpha \nabla^3 \widehat{\mathcal{L}}(u_{\mathcal{T}_i,q}) \prod_{l=0}^{q-1}(I - \alpha \nabla^2 \widehat{\mathcal{L}}(u_{\mathcal{T}_i,q}))\| \\
\leq& \|\alpha \nabla^3 \widehat{\mathcal{L}}(w_{\mathcal{T}_i,q}) \prod_{l=0}^{q-1}(I - \alpha \nabla^2 \widehat{\mathcal{L}}(w_{\mathcal{T}_i,q})) - \alpha \nabla^3 \widehat{\mathcal{L}}(u_{\mathcal{T}_i,q}) \prod_{l=0}^{q-1}(I - \alpha \nabla^2 \widehat{\mathcal{L}}(w_{\mathcal{T}_i,q}))\| \\
&+ \|\alpha \nabla^3 \widehat{\mathcal{L}}(u_{\mathcal{T}_i,q}) \prod_{l=0}^{q-1}(I - \alpha \nabla^2 \widehat{\mathcal{L}}(w_{\mathcal{T}_i,q})) - \alpha \nabla^3 \widehat{\mathcal{L}}(u_{\mathcal{T}_i,q}) \prod_{l=0}^{q-1}(I - \alpha \nabla^2 \widehat{\mathcal{L}}(u_{\mathcal{T}_i,q}))\| \\
\leq& \alpha(1 + \alpha L)^q \|\nabla^3 \widehat{\mathcal{L}}(w_{\mathcal{T}_i,q}) - \nabla^3 \widehat{\mathcal{L}}(u_{\mathcal{T}_i,q})\| + \alpha \rho \|\prod_{l=0}^{q-1}(I - \alpha \nabla^2 \widehat{\mathcal{L}}(w_{\mathcal{T}_i,q})) - \prod_{l=0}^{q-1}(I - \alpha \nabla^2 \widehat{\mathcal{L}}(u_{\mathcal{T}_i,q}))\| \\
\leq& \alpha \kappa (1 + \alpha L)^q \|w - u\| + \alpha^2 \rho^2 q (1 + \alpha L)^{q-1} \|w - u\|
\end{aligned}
$$
(20)

Putting (20) and (19) into (18), we have

$$
\begin{aligned}
& \|\sum_{q=0}^{Q-1}\left(\prod_{l=0}^{q-1} J_l\right) \frac{\partial J_q}{\partial w}\left(\prod_{l=q+1}^{Q-1} J_l\right) \nabla \ell(w_{\mathcal{T}_i,Q}) - \sum_{q=0}^{Q-1}\left(\prod_{l=0}^{q-1} J_l\right) \frac{\partial J_q}{\partial u}\left(\prod_{l=q+1}^{Q-1} J_l\right) \nabla \ell(u_{\mathcal{T}_i,Q})\| \\
& \leq Q\left[\alpha \rho(1 + \alpha L)^{2(Q-2)} + G \alpha \kappa (1 + \alpha L)^{2Q-1} + G \alpha^2 \rho^2 Q (1 + \alpha L)^{2(Q-1)}\right] \|w - u\|.
\end{aligned}
$$
(21)

Putting (17) and (21) into (14), we get

$$
\begin{aligned}
\|\nabla^2 F_i(w) - \nabla^2 F_i(u)\| \leq& \left[\alpha \rho Q(1 + \alpha L)^{2Q-1} + (1 + \alpha L)^{2Q} \rho\right] \|w - u\| \\
&+ Q\left[\alpha \rho(1 + \alpha L)^{2(Q-2)} + G \alpha \kappa (1 + \alpha L)^{2Q-1} + G \alpha^2 \rho^2 Q (1 + \alpha L)^{2(Q-1)}\right] \|w - u\| \\
=& (1 + \alpha L)^{2Q-1}\left[\alpha \rho Q + 2\rho + Q \alpha \rho + G \alpha \kappa + G \alpha^2 \rho^2 Q\right] \|w - u\| \\
=& \rho_Q \|w - u\|,
\end{aligned}
$$

where we denote $\rho_Q = (1 + \alpha L)^{2Q-1}\left[\alpha \rho Q + 2\rho + Q \alpha \rho + G \alpha \kappa + G \alpha^2 \rho^2 Q\right]$. $\qquad\square$

**Lemma 12.** *Suppose that Assumption 2 hold and the function $F_i(w)$ defined in Eq (3). Then, for any $w, u \in \mathbb{R}^d$, if $\ell$ is non-convex, we have*

$$
\|\nabla^2 F_i(w) - \nabla^2 F_i(u)\| \leq \rho_Q \|w - u\|
$$

*where $\rho_Q = \left[\frac{3\rho(1 + \alpha L)^{2(Q-1)}}{L} + (1 + \alpha L)^{3Q} \rho + \alpha \kappa (1 + \alpha L)^{2q} + \frac{3\alpha \rho^2 (1 + \alpha L)^{2(Q-1)}}{L}\right]$*

*Proof.* Similar to the proof of (14), we have

$$
\begin{aligned}
\|\nabla^2 F_i(w) - \nabla^2 F_i(u)\| \leq& \|\prod_{q=0}^{Q-1}(I - \alpha \nabla^2 \mathcal{L}_i(w_{i,q}))^2 \nabla^2 \ell(w_{i,Q}) - \prod_{q=0}^{Q-1}(I - \alpha \nabla^2 \mathcal{L}_i(u_{i,q}))^2 \nabla^2 \ell(u_{i,Q})\| \\
&+ \|\sum_{q=0}^{Q-1}\left(\prod_{k=0}^{q-1} J_k\right) \frac{\partial J_q}{\partial w}\left(\prod_{k=q+1}^{Q-1} J_k\right) \nabla \ell(w_{i,Q}) \\
&- \sum_{q=0}^{Q-1}\left(\prod_{k=0}^{q-1} J_k\right) \frac{\partial J_q}{\partial u}\left(\prod_{k=q+1}^{Q-1} J_k\right) \nabla \ell(u_{i,Q})\|
\end{aligned}
$$
(22)

For the first term, we have

$$\|\prod_{q=0}^{Q-1}(I - \alpha\nabla^2\mathcal{L}_i(w_{i,q}))^2\nabla^2\ell(w_{i,Q}) - \prod_{q=0}^{Q-1}(I - \alpha\nabla^2\mathcal{L}_i(u_{i,q}))^2\nabla^2\ell(u_{i,Q})\|$$

$$\leq\|\prod_{q=0}^{Q-1}(I - \alpha\nabla^2\mathcal{L}_i(w_{i,q}))^2 - \prod_{q=0}^{Q-1}(I - \alpha\nabla^2\mathcal{L}_i(u_{i,q}))^2\|\|\nabla^2\ell(w_{i,Q})\|$$

$$+ \|\prod_{q=0}^{Q-1}(I - \alpha\nabla^2\mathcal{L}_i(u_{i,q}))^2\|\|\nabla^2\ell(w_{i,Q}) - \nabla^2\ell(u_{i,Q})\| \tag{23}$$

$$\leq\underbrace{\|\prod_{q=0}^{Q-1}(I - \alpha\nabla^2\mathcal{L}_i(w_{i,q}))^2 - \prod_{q=0}^{Q-1}(I - \alpha\nabla^2\mathcal{L}_i(u_{i,q}))^2\|}_{\Lambda(Q-1)} L + (1+\alpha L)^{3Q}\rho\|w - u\|$$

where we use Assumption 2 and Lemma 6 in the last inequality. We next upper-bound $\Lambda(Q-1)$ in the above inequality. Similar to the proof of (16), we have

$$\Lambda(Q-1) \leq \frac{3\rho(1+\alpha L)^{2(Q-1)}}{L}\|w - u\| \tag{24}$$

Putting (24) into (23), we have

$$\|\prod_{q=0}^{Q-1}(I - \alpha\nabla^2\widehat{\mathcal{L}}_i(w_{\mathcal{T}_i,q}))^2\nabla^2\ell(w_{\mathcal{T}_i,Q}) - \prod_{q=0}^{Q-1}(I - \alpha\nabla^2\widehat{\mathcal{L}}_i(u_{\mathcal{T}_i,q}))^2\nabla^2\ell(u_{\mathcal{T}_i,Q})\|$$

$$\leq \left[\frac{3\rho(1+\alpha L)^{2(Q-1)}}{L} + (1+\alpha L)^{3Q}\rho\right]\|w - u\| \tag{25}$$

To bound the second term, similar to the proof of (18), we have

$$\|\sum_{q=0}^{Q-1}\left(\prod_{k=0}^{q-1}J_k\right)\frac{\partial J_q}{\partial w}\left(\prod_{k=q+1}^{Q-1}J_k\right)\nabla\ell(w_{i,Q}) - \sum_{q=0}^{Q-1}\left(\prod_{k=0}^{q-1}J_k\right)\frac{\partial J_q}{\partial u}\left(\prod_{k=q+1}^{Q-1}J_k\right)\nabla\ell(u_{i,Q})\|$$

$$\leq\|\sum_{q=0}^{Q-1}\left(\prod_{k=0}^{q-1}J_k\right)\frac{\partial J_q}{\partial w}\left(\prod_{k=q+1}^{Q-1}J_k\right)\|\|\nabla\ell(w_{i,Q}) - \nabla\ell(u_{i,Q})\|$$

$$+ \|\sum_{q=0}^{Q-1}\left(\prod_{k=0}^{q-1}J_k\right)\frac{\partial J_q}{\partial w}\left(\prod_{k=q+1}^{Q-1}J_k\right) - \sum_{q=0}^{Q-1}\left(\prod_{k=0}^{q-1}J_k\right)\frac{\partial J_q}{\partial u}\left(\prod_{k=q+1}^{Q-1}J_k\right)\|\|\nabla\ell(u_{i,Q})\|$$

$$\leq\sum_{q=0}^{Q-1}\|\left(\prod_{k=0}^{q-1}J_k\right)\frac{\partial J_q}{\partial w}\left(\prod_{k=q+1}^{Q-1}J_k\right)\|(1+\alpha L)^Q L\|w - u\|$$

$$+ \sum_{q=0}^{Q-1}\|\left(\prod_{k=0}^{q-1}J_k\right)\frac{\partial J_q}{\partial w}\left(\prod_{k=q+1}^{Q-1}J_k\right) - \left(\prod_{k=0}^{q-1}J_k\right)\frac{\partial J_q}{\partial u}\left(\prod_{k=q+1}^{Q-1}J_k\right)\|G$$

$$\leq\sum_{q=0}^{Q-1}(1+\alpha L)^{2Q-1}L\|\frac{\partial J_q}{\partial w}\|\|w - u\| + \sum_{q=0}^{Q-1}(1+\alpha L)^{Q-1}G\|\frac{\partial J_q}{\partial w} - \frac{\partial J_q}{\partial u}\|,$$

$$\tag{26}$$

where we use Assumption 2 and Lemma 6 in the second inequality. First, we have

$$\|\frac{\partial J_q}{\partial w}\| = \|\alpha\nabla^3 L(w_{i,q})\prod_{k=0}^{q-1}(I - \alpha\nabla^2 L_i(w_{i,q}))\|$$

$$\leq(1+\alpha L)^{q-1}\alpha\rho \tag{27}$$

Secondly, we have

$$
\begin{aligned}
\|\frac{\partial J_q}{\partial w} - \frac{\partial J_q}{\partial u}\| = & \|\alpha\nabla^3 L(w_{i,q})\prod_{k=0}^{q-1}(I-\alpha\nabla^2 L_i(w_{i,q})) - \alpha\nabla^3 L(u_{i,q})\prod_{k=0}^{q-1}(I-\alpha\nabla^2 L_i(u_{i,q}))\| \\
\leq & \|\alpha\nabla^3 L(w_{i,q})\prod_{k=0}^{q-1}(I-\alpha\nabla^2 L_i(w_{i,q})) - \alpha\nabla^3 L(u_{i,q})\prod_{k=0}^{q-1}(I-\alpha\nabla^2 L_i(w_{i,q}))\| \\
& + \|\alpha\nabla^3 L(u_{i,q})\prod_{k=0}^{q-1}(I-\alpha\nabla^2 L_i(w_{i,q})) - \alpha\nabla^3 L(u_{i,q})\prod_{k=0}^{q-1}(I-\alpha\nabla^2 L_i(u_{i,q}))\| \\
\leq & \alpha(1+\alpha L)^q\|\nabla^3 L(w_{i,q}) - \nabla^3 L(u_{i,q})\| + \alpha\rho\|\prod_{k=0}^{q-1}(I-\alpha\nabla^2 L_i(w_{i,q})) - \prod_{k=0}^{q-1}(I-\alpha\nabla^2 L_i(u_{i,q}))\| \\
\leq & \alpha\kappa(1+\alpha L)^{2q}\|w-u\| + \frac{3\alpha\rho^2(1+\alpha L)^{2(Q-1)}}{L}\|w-u\|.
\end{aligned}
\tag{28}
$$

Putting (20) and (19) into (18), we have

$$
\begin{aligned}
& \|\sum_{q=0}^{Q-1}\left(\prod_{k=0}^{q-1} J_k\right)\frac{\partial J_q}{\partial w}\left(\prod_{k=q+1}^{Q-1} J_k\right)\nabla\ell(w_{i,Q}) - \sum_{q=0}^{Q-1}\left(\prod_{k=0}^{q-1} J_k\right)\frac{\partial J_q}{\partial u}\left(\prod_{k=q+1}^{Q-1} J_k\right)\nabla\ell(u_{i,Q})\| \\
& \leq Q\left[\alpha\rho(1+\alpha L)^{3Q-2} + \frac{3G\alpha\rho^2(1+\alpha L)^{3(Q-1)}}{L}\right]\|w-u\|.
\end{aligned}
\tag{29}
$$

Putting (25) and (29) into (22), we get

$$
\begin{aligned}
\|\nabla^2 F_i(w) - \nabla^2 F_i(u)\| \leq & \left[\frac{3\rho(1+\alpha L)^{2(Q-1)}}{L} + (1+\alpha L)^{3Q}\rho\right]\|w-u\| \\
& + \alpha\kappa(1+\alpha L)^{2Q}\|w-u\| + \frac{3\alpha\rho^2(1+\alpha L)^{2(Q-1)}}{L}\|w-u\|. \\
= & \rho_Q\|w-u\|,
\end{aligned}
$$

where we denote $\rho_Q = \left[\frac{3\rho(1+\alpha)(1+\alpha L)^{2(Q-1)}}{L} + (1+\alpha L)^{3Q}\rho + \alpha\kappa(1+\alpha L)^{2Q}\right]$. $\qquad\square$

As shown in Algotihm 1, in the outer-level, We are also performing SGD by considering $S$ as a meta-sample (which is equivalent to $z$), then we can obtain the following Lemmas.

**Lemma 13.** *(Lemma 4 in [47])Suppose that Assumption 2 and Assumption 3 hold and $F$ is $L_Q$-smoothness. Then, for any $w \in \mathcal{W}$, we have*

$$
\mathbb{E}_S[\sum_{t=0}^{T-1}\eta_t\|\nabla\widehat{F}(w^t, S_t)\|] \leq 2\sqrt{\sum_{t=0}^{T-1}\eta_t}\sqrt{F(w^{t+1}) - \min_{\mathcal{W}} F + \frac{L_Q\sigma^2}{2}\sum_{t=0}^{T-1}\eta_t^2} + \sum_{t=0}^{T-1}\sigma\eta_t.
$$

**Lemma 14.** *(Lemma 6 in [47])Let $G_t(w) := w - \eta_t\nabla F(w, S_i)$ and assume that the loss function $F(\cdot, S_i)$ is $L_Q$-smooth and that its Hessian is $\rho_Q$-Lipschitz. Then,*

$$
\left\|G_t(w_{\mathcal{S},t}) - G_t(w_{\widetilde{\mathcal{S}}^{(i)},t})\right\| \leq (1+\eta_t\xi_t)\|w_{\mathcal{S},t} - w_{\widetilde{\mathcal{S}}^{(i)},t}\|,
$$

*where $\xi_t := \|\nabla^2 F(w_0, S_t)\| + \frac{\rho_Q}{2}\left\|\sum_{l=1}^{t-1}\beta_l\nabla F(w_{\mathcal{S}}^l, S_l)\right\| + \frac{\rho_Q}{2}\left\|\sum_{l=1}^{t-1}\beta_l\nabla F(w_{\widetilde{\mathcal{S}}}^l, S_l)\right\|$. Furthermore, we have $\mathbb{E}_{\mathcal{S},S}[\xi_t] = \mathbb{E}_{\mathcal{S},S}\|\nabla^2 F(w_0, S_t)\| + \rho_Q\mathbb{E}_{\mathcal{S},S}\left\|\sum_{k=1}^{t-1}\beta_k\nabla F(w_{\mathcal{S}}^k, S_k)\right\|$. Furthermore, for any $t \in [T]$,*

$$
\begin{aligned}
\mathbb{E}_{\mathcal{S},S}\left[\xi_t(\mathcal{S}, S)\right] \leq & \mathbb{E}_{\mathcal{S},S}\left[\left\|\nabla^2 F(w_0, S_t)\right\|\right] \\
& + 2\rho_Q\sqrt{(F(w_0) - \min_{\mathcal{W}} F)c(1+\ln(T))} \\
& + \sigma\rho_Q\left(\sqrt{2cL_Q} + c(1+\ln(T))\right).
\end{aligned}
$$

**Lemma 15.** *(Lemma 5 in [47])Suppose that Assumption 2 hold, then for every $t_0 \in \{0, 1, 2, ..., T\}$ we have that*

$$\mathbb{E}_{\mathcal{S},\widetilde{z},\mathcal{A}}[\ell(w_{\mathcal{T}_i}(w_{\mathcal{S}}^T, \widetilde{S}_i^{\mathrm{tr}}), \widetilde{z}) - \ell(w_{\mathcal{T}_i}(\widetilde{w}_{\widetilde{\mathcal{S}}}^T, \widetilde{S}_i^{\mathrm{tr}}), \widetilde{z})|] \leq eG\mathbb{E}_{\mathcal{S},\widetilde{z},\mathcal{A}}[w_{\mathcal{S}}^T - w_{\widetilde{\mathcal{S}}}^T | w_{\mathcal{S}}^{t_0} - w_{\widetilde{\mathcal{S}}}^{t_0} = 0] + \mathbb{E}_{\mathcal{S},\mathcal{A}}[F(w_{\mathcal{S}})]\frac{t_0}{m}$$

### B.3 Proof of Theorem 2(convex)

Let's consider two parallel processes of generating iterates $\{w^t\}$ and $\{\widetilde{w}^t\}$ by using datasets $\mathcal{S}$ and $\widetilde{\mathcal{S}}$, respectively. We use the tilde superscript to refer to the second process throughout the proof. By Lemma 9, we know $\ell$ is $L_Q$-smooth and convex function. Hence, for a given time index $t$, with probability $1 - \frac{1}{m}$, the task $\mathcal{T}_j$ is selected, where $j \neq i$. By using Lemma 1, we have

$$
\begin{aligned}
\|w^{t+1} - \widetilde{w}^{t+1}\| = & \|(w^t - \eta_t\nabla\widehat{\mathcal{L}}(w_{\mathcal{T}_j}(w^t, S_j^{\mathrm{tr}}), S_j^{\mathrm{ts}})) - (\widetilde{w}^t - \eta_t\nabla\widehat{\mathcal{L}}(w_{\mathcal{T}_j}(\widetilde{w}^t, S_j^{\mathrm{tr}}), S_j^{\mathrm{ts}}))\| \\
\leq & \frac{1}{n^{\mathrm{ts}}}\sum_{z^{\mathrm{ts}} \in S_j^{\mathrm{ts}}} \|(w^t - \eta_t\nabla\ell(w_{\mathcal{T}_j}(w^t, S_j^{\mathrm{tr}}), z^{\mathrm{ts}})) - (\widetilde{w}^t - \eta_t\nabla\ell(w_{\mathcal{T}_j}(\widetilde{w}^t, S_j^{\mathrm{tr}}), z^{\mathrm{ts}}))\| \\
\leq & \|w^t - \widetilde{w}^t\|
\end{aligned}
$$
(30)

Next, for a given time index $t$, with probability $\frac{1}{m}$, the task $\mathcal{T}_i$ is selected. In this case, we have

$$
\begin{aligned}
\mathbb{E}\|w^{t+1} - \widetilde{w}^{t+1}\| = & \mathbb{E}\|(w^t - \eta_t\nabla\widehat{\mathcal{L}}(w_{\mathcal{T}_i}(w^t, S_i^{\mathrm{tr}}), S_i^{\mathrm{ts}})) - (\widetilde{w}^t - \eta_t\nabla\widehat{\mathcal{L}}(w_{\mathcal{T}_i}(\widetilde{w}^t, \widetilde{S}_i^{\mathrm{tr}}), \widetilde{S}_i^{\mathrm{ts}}))\| \\
\leq & \mathbb{E}\|(w^t - \eta_t\nabla\widehat{\mathcal{L}}(w_{\mathcal{T}_i}(w^t, S_i^{\mathrm{tr}}), S_i^{\mathrm{ts}})) - (\widetilde{w}^t - \eta_t\nabla\widehat{\mathcal{L}}(w_{\mathcal{T}_i}(\widetilde{w}^t, \widetilde{S}_i^{\mathrm{tr}}), S_i^{\mathrm{ts}}))\| \\
& + \mathbb{E}\|\eta_t\nabla\widehat{\mathcal{L}}(w_{\mathcal{T}_i}(\widetilde{w}^t, \widetilde{S}_i^{\mathrm{tr}}), \widetilde{S}_i^{\mathrm{ts}}) - \eta_t\nabla\widehat{\mathcal{L}}(w_{\mathcal{T}_i}(\widetilde{w}^t, \widetilde{S}_i^{\mathrm{tr}}), S_i^{\mathrm{ts}})\| \\
\leq & \frac{1}{n^{\mathrm{ts}}}\sum_{z^{\mathrm{ts}} \in S_i^{\mathrm{ts}}} \mathbb{E}\|(w^t - \eta_t\nabla\widehat{\mathcal{L}}(w_{\mathcal{T}_i}(w^t, S_i^{\mathrm{tr}}), z^{\mathrm{ts}})) - (\widetilde{w}^t - \eta_t\nabla\widehat{\mathcal{L}}(w_{\mathcal{T}_i}(\widetilde{w}^t, \widetilde{S}_i^{\mathrm{tr}}), z^{\mathrm{ts}}))\| \\
& + \eta_t\mathbb{E}\|\nabla\widehat{\mathcal{L}}(w_{\mathcal{T}_i}(\widetilde{w}^t, \widetilde{S}_i^{\mathrm{tr}}), \widetilde{S}_i^{\mathrm{ts}}) - \nabla\widehat{\mathcal{L}}(w_{\mathcal{T}_i}(\widetilde{w}^t, \widetilde{S}_i^{\mathrm{tr}}), S_i^{\mathrm{ts}})\| \\
\leq & \frac{1}{n^{\mathrm{ts}}}\sum_{z \in S_i^{\mathrm{ts}}} \mathbb{E}\|(w^t - \eta_t\nabla\ell(w_{\mathcal{T}_i}(w^t, S_i^{\mathrm{tr}}), z^{\mathrm{ts}})) - (\widetilde{w}^t - \eta_t\nabla\ell(w_{\mathcal{T}_i}(\widetilde{w}^t, \widetilde{S}_i^{\mathrm{tr}}), z^{\mathrm{ts}}))\| \\
& + 2\eta_t\mathbb{E}\|\nabla\widehat{F}_i(w)\|,
\end{aligned}
$$
(31)

where the last inequality follows that $\widetilde{S}_i^{\mathrm{ts}}$ and $S_i^{\mathrm{ts}}$ are sampled from the same distribution, then $\mathbb{E}\|\nabla\widehat{\mathcal{L}}(w_{\mathcal{T}_i}(\widetilde{w}^t, \widetilde{S}_i^{\mathrm{tr}}), \widetilde{S}_i^{\mathrm{ts}})\| = \mathbb{E}\|\nabla\widehat{\mathcal{L}}(w_{\mathcal{T}_i}(\widetilde{w}^t, \widetilde{S}_i^{\mathrm{tr}}), S_i^{\mathrm{ts}}) = \mathbb{E}\|\nabla\widehat{F}_i(w)\|$. Note that

$$
\begin{aligned}
& \mathbb{E}\|(w^t - \eta_t\nabla\ell(w_{\mathcal{T}_i}(w^t, S_i^{\mathrm{tr}}), z^{\mathrm{ts}})) - (\widetilde{w}^t - \eta_t\nabla\ell(w_{\mathcal{T}_i}(\widetilde{w}^t, \widetilde{S}_i^{\mathrm{tr}}), z^{\mathrm{ts}}))\| \\
\leq & \mathbb{E}\|(w^t - \eta_t\nabla\ell(w_{\mathcal{T}_i}(w^t, S_i^{\mathrm{tr}}), z^{\mathrm{ts}})) - (\widetilde{w}^t - \eta_t\nabla\ell(w_{\mathcal{T}_i}(\widetilde{w}^t, S_i^{\mathrm{tr}}), z^{\mathrm{ts}}))\| \\
& + \eta_t\mathbb{E}\|\nabla\ell(w_{\mathcal{T}_i}(\widetilde{w}^t, S_i^{\mathrm{tr}}), z^{\mathrm{ts}}) - \nabla\ell(w_{\mathcal{T}_i}(\widetilde{w}^t, \widetilde{S}_i^{\mathrm{tr}}), z^{\mathrm{ts}})\|.
\end{aligned}
$$
(32)

Let us bound the two terms on the RHS of (32) separately. First, similar to how we derived (30), we could bound the first term by

$$\mathbb{E}\|(w^t - \eta_t\nabla\ell(w_{\mathcal{T}_i}(w^t, S_i^{\mathrm{tr}}), z^{\mathrm{ts}})) - (\widetilde{w}^t - \eta_t\nabla\ell(w_{\mathcal{T}_i}(\widetilde{w}^t, S_i^{\mathrm{tr}}), z^{\mathrm{ts}}))\| \leq \mathbb{E}\|w^t - \widetilde{w}^t\|.$$

To bound the second term on the RHS of (32), we consider two parallel processes of generating iterates $\{\widetilde{w}^t_{\mathcal{T}_i,q}\}$ and $\{\widetilde{w}^{t,\prime}_{\mathcal{T}_i,q}\}$ by using datasets $S^{\text{tr}}_i$ and $\widetilde{S}^{\text{tr}}_i$, respectively. Note that

$$
\mathbb{E}\|\nabla\ell\big(w_{\mathcal{T}_i}(\widetilde{w}^t, S^{\text{tr}}_i), z^{\text{ts}}\big) - \nabla\ell\big(w_{\mathcal{T}_i}(\widetilde{w}^t, \widetilde{S}^{\text{tr}}_i), z^{\text{ts}}\big)\|
$$

$$
=\mathbb{E}\| \prod_{q=0}^{Q-1}(I - \alpha\nabla^2\widehat{\mathcal{L}}(\widetilde{w}^t_{\mathcal{T}_i,q}, S^{\text{tr}}_i))\nabla\ell(\widetilde{w}^t_{\mathcal{T}_i,Q}, z^{\text{ts}}) - \prod_{q=0}^{Q-1}(I - \alpha\nabla^2\widehat{\mathcal{L}}(\widetilde{w}^{t,\prime}_{i,q}, \widetilde{S}^{\text{tr}}_i))\nabla\ell(\widetilde{w}^{t,\prime}_{i,Q}, z^{\text{ts}})\|
$$

$$
\leq\mathbb{E}\| \prod_{q=0}^{Q-1}(I - \alpha\nabla^2\widehat{\mathcal{L}}(\widetilde{w}^t_{\mathcal{T}_i,q}, S^{\text{tr}}_i))\nabla\ell(\widetilde{w}^t_{\mathcal{T}_i,q}, z^{\text{ts}}) - \prod_{q=0}^{Q-1}(I - \alpha\nabla^2\widehat{\mathcal{L}}(\widetilde{w}^t_{\mathcal{T}_i,q}, S^{\text{tr}}_i))\nabla\ell(\widetilde{w}^{t,\prime}_{i,Q}, z^{\text{ts}})\|
$$

$$
+ \mathbb{E}\| \prod_{q=0}^{Q-1}(I - \alpha\nabla^2\widehat{\mathcal{L}}(\widetilde{w}^t_{\mathcal{T}_i,q}, S^{\text{tr}}_i))\nabla\ell(\widetilde{w}^{t,\prime}_{i,Q}, z^{\text{ts}}) - \prod_{q=0}^{Q-1}(I - \alpha\nabla^2\widehat{\mathcal{L}}(\widetilde{w}^{t,\prime}_{i,q}, \widetilde{S}^{\text{tr}}_i))\nabla\ell(\widetilde{w}^{t,\prime}_{i,Q}, z^{\text{ts}})\|
$$

$$
\leq\mathbb{E}\| \prod_{q=0}^{Q-1}(I - \alpha\nabla^2\widehat{\mathcal{L}}(\widetilde{w}^t_{\mathcal{T}_i,q}, S^{\text{tr}}_i))\|\|\nabla\ell(\widetilde{w}^t, z^{\text{ts}}) - \nabla\ell(\widetilde{w}^{t,\prime}_{i,Q}, z^{\text{ts}})\|
$$

$$
+ \mathbb{E}\| \prod_{q=0}^{Q-1}(I - \alpha\nabla^2\widehat{\mathcal{L}}(\widetilde{w}^t_{\mathcal{T}_i,q}, S^{\text{tr}}_i)) - \prod_{q=0}^{Q-1}(I - \alpha\nabla^2\widehat{\mathcal{L}}(\widetilde{w}^{t,\prime}_{i,q}, \widetilde{S}^{\text{tr}}_i))\|\|\nabla\ell(\widetilde{w}^{t,\prime}_{i,Q}, z^{\text{ts}})\|
$$

$$
\leq(1 + \alpha L)^Q\mathbb{E}\|\nabla\ell(\widetilde{w}^t_{\mathcal{T}_i,Q}, z^{\text{ts}}) - \nabla\ell(\widetilde{w}^{t,\prime}_{i,Q}, z^{\text{ts}})\|
$$

$$
+ G\mathbb{E}\| \underbrace{\prod_{q=0}^{Q-1}(I - \alpha\nabla^2\widehat{\mathcal{L}}(\widetilde{w}^t_{\mathcal{T}_i,Q}, S^{\text{tr}}_i)) - \prod_{q=0}^{Q-1}(I - \alpha\nabla^2\widehat{\mathcal{L}}(\widetilde{w}^{t,\prime}_{i,q}, \widetilde{S}^{\text{tr}}_i))}_{V(Q)}\|
$$

$$
\tag{33}
$$

where we use Assumption 2 in the last inequality. Hence, what remains is to bound the two terms in (33). To do so, notice that

$$
\begin{aligned}
\mathbb{E}\|\nabla\ell(\widetilde{w}^t_{\mathcal{T}_i,Q}, z^{\text{ts}}) - \nabla\ell(\widetilde{w}^{t,\prime}_{i,Q}, z^{\text{ts}})\| &\leq L\mathbb{E}\|\widetilde{w}^t_{\mathcal{T}_i,Q} - \widetilde{w}^{t,\prime}_{i,Q}\| \\
&= L\mathbb{E}\| \left[\widetilde{w}^t_{i,Q-1} - \alpha\nabla\widehat{\mathcal{L}}(\widetilde{w}^t_{i,Q-1}, S^{\text{tr}}_i)\right] - \left[\widetilde{w}^{t,\prime}_{i,Q-1} - \alpha\nabla\widehat{\mathcal{L}}(\widetilde{w}^{t,\prime}_{i,Q-1}, \widetilde{S}^{\text{tr}}_i)\right]\| \\
&\leq L\mathbb{E}\| \left[\widetilde{w}^t_{i,Q-1} - \alpha\nabla\widehat{\mathcal{L}}(\widetilde{w}^t_{i,Q-1}, S^{\text{tr}}_i))\right] - \left[\widetilde{w}^{t,\prime}_{i,Q-1} - \alpha\nabla\ell(\widetilde{w}^{t,\prime}_{i,Q-1}, S^{\text{tr}}_i)\right]\| \\
&\quad + L\mathbb{E}\|\alpha\nabla\widehat{\mathcal{L}}(\widetilde{w}^{t,\prime}_{i,Q-1}, S^{\text{tr}}_i) - \alpha\nabla\widehat{\mathcal{L}}(\widetilde{w}^{t,\prime}_{i,Q-1}, \widetilde{S}^{\text{tr}}_i)\| \\
&\leq L\mathbb{E}\|\widetilde{w}^t_{i,Q-1} - \widetilde{w}^{t,\prime}_{i,Q-1}\| + \frac{2\alpha LG}{n^{\text{tr}}} \\
&\leq L\mathbb{E}\| \left[\widetilde{w}^t - \alpha\nabla\widehat{\mathcal{L}}(\widetilde{w}^t, S^{\text{tr}}_i)\right] - \left[\widetilde{w}^t - \alpha\nabla\widehat{\mathcal{L}}(\widetilde{w}^t, \widetilde{S}^{\text{tr}}_i)\right]\| + \frac{2(Q-1)\alpha LG}{n^{\text{tr}}} \\
&= \alpha L\mathbb{E}\|\nabla\widehat{\mathcal{L}}(\widetilde{w}^t, S^{\text{tr}}_i) - \nabla\widehat{\mathcal{L}}(\widetilde{w}^t, \widetilde{S}^{\text{tr}}_i)\| + \frac{2(Q-1)\alpha LG}{n^{\text{tr}}} \\
&\leq \frac{2Q\alpha LG}{n^{\text{tr}}},
\end{aligned}
$$

$$
\tag{34}
$$

where we use Lemma 1 and Assumption 2 in the third inequality. Next,

$$
\begin{aligned}
V(Q) =& \| \prod_{q=0}^{Q-1} (I - \alpha\nabla^2\widehat{\mathcal{L}}(\widetilde{w}^t_{\mathcal{T}_i,q}, S^{\mathrm{tr}}_i)) - \prod_{q=0}^{Q-1} (I - \alpha\nabla^2\widehat{\mathcal{L}}(\widetilde{w}^{t,\prime}_{i,q}, \widetilde{S}^{\mathrm{tr}}_i)) \| \\
=& \| \prod_{q=0}^{Q-2} (I - \alpha\nabla^2\widehat{\mathcal{L}}(\widetilde{w}^t_{\mathcal{T}_i,q}, S^{\mathrm{tr}}_i))(I - \alpha\nabla^2\widehat{\mathcal{L}}(\widetilde{w}^t_{i,Q-1}, S^{\mathrm{tr}}_i)) \\
& - \prod_{q=0}^{Q-2} (I - \alpha\nabla^2\widehat{\mathcal{L}}(\widetilde{w}^{t,\prime}_{i,q}, \widetilde{S}^{\mathrm{tr}}_i))(I - \alpha\nabla^2\widehat{\mathcal{L}}(\widetilde{w}^{t,\prime}_{i,Q-1}, \widetilde{S}^{\mathrm{tr}}_i)) \| \\
\leq& \| \prod_{q=0}^{Q-2} (I - \alpha\nabla^2\widehat{\mathcal{L}}(\widetilde{w}^t_{\mathcal{T}_i,q}, S^{\mathrm{tr}}_i))(I - \alpha\nabla^2\widehat{\mathcal{L}}(\widetilde{w}^t_{i,Q-1}, S^{\mathrm{tr}}_i)) \\
& - \prod_{q=0}^{Q-2} (I - \alpha\nabla^2\widehat{\mathcal{L}}(\widetilde{w}^t_{\mathcal{T}_i,Q}, S^{\mathrm{tr}}_i))(I - \alpha\nabla^2\widehat{\mathcal{L}}(\widetilde{w}^{t,\prime}_{i,Q-1}, \widetilde{S}^{\mathrm{tr}}_i)) \| \\
& + \| \prod_{q=0}^{Q-2} (I - \alpha\nabla^2\widehat{\mathcal{L}}(\widetilde{w}^t_{\mathcal{T}_i,q}, S^{\mathrm{tr}}_i))(I - \alpha\nabla^2\widehat{\mathcal{L}}(\widetilde{w}^{t,\prime}_{i,Q-1}, \widetilde{S}^{\mathrm{tr}}_i)) \\
& - \prod_{q=0}^{Q-2} (I - \alpha\nabla^2\widehat{\mathcal{L}}(\widetilde{w}^{t,\prime}_{i,q}, \widetilde{S}^{\mathrm{tr}}_i))(I - \alpha\nabla^2\widehat{\mathcal{L}}(\widetilde{w}^{t,\prime}_{i,Q-1}, \widetilde{S}^{\mathrm{tr}}_i)) \| \\
=& \| \prod_{q=0}^{Q-2} (I - \alpha\nabla^2\widehat{\mathcal{L}}(\widetilde{w}^t_{\mathcal{T}_i,q}, S^{\mathrm{tr}}_i)) \| \| \alpha\nabla^2\widehat{\mathcal{L}}(\widetilde{w}^t_{i,Q-1}, S^{\mathrm{tr}}_i) - \alpha\nabla^2\widehat{\mathcal{L}}(\widetilde{w}^{t,\prime}_{i,Q-1}, \widetilde{S}^{\mathrm{tr}}_i) \| \\
& + V(Q-1)\| I - \alpha\nabla^2\widehat{\mathcal{L}}(\widetilde{w}^{t,\prime}_{i,Q-1}, \widetilde{S}^{\mathrm{tr}}_i) \| \\
\leq& (1+\alpha L)^{Q-1}\| \alpha\nabla^2\widehat{\mathcal{L}}(\widetilde{w}^t_{i,Q-1}, S^{\mathrm{tr}}_i) - \alpha\nabla^2\widehat{\mathcal{L}}(\widetilde{w}^{t,\prime}_{i,Q-1}, \widetilde{S}^{\mathrm{tr}}_i) \| + (1+\alpha L)V(Q-1)
\end{aligned}
$$

where we use Assumption 2 in the last inequality. To bound the above inequality, we first consider the first term.

$$
\begin{aligned}
& (1+\alpha L)^{Q-1}\| \alpha\nabla^2\widehat{\mathcal{L}}(\widetilde{w}^t_{i,Q-1}, S^{\mathrm{tr}}_i) - \alpha\nabla^2\widehat{\mathcal{L}}(\widetilde{w}^{t,\prime}_{i,Q-1}, \widetilde{S}^{\mathrm{tr}}_i) \| \\
\leq& \alpha(1+\alpha L)^{Q-1}\left[ \| \nabla^2\widehat{\mathcal{L}}(\widetilde{w}^t_{i,Q-1}, S^{\mathrm{tr}}_i) - \nabla^2\widehat{\mathcal{L}}(\widetilde{w}^{t,\prime}_{i,Q-1}, S^{\mathrm{tr}}_i) \| + \| \nabla^2\widehat{\mathcal{L}}(\widetilde{w}^{t,\prime}_{i,Q-1}, S^{\mathrm{tr}}_i) - \nabla^2\widehat{\mathcal{L}}(\widetilde{w}^{t,\prime}_{i,Q-1}, \widetilde{S}^{\mathrm{tr}}_i) \| \right] \\
\leq& \alpha(1+\alpha L)^{Q-1}\left[ \rho\| \widetilde{w}^t_{i,Q-1} - \widetilde{w}^{t,\prime}_{i,Q-1} \| + \frac{2L}{n^{\mathrm{tr}}} \right] \\
=& \alpha(1+\alpha L)^{Q-1}\left[ \rho\| \big(\widetilde{w}^t_{i,Q-2} - \alpha\nabla\widehat{\mathcal{L}}(\widetilde{w}^t_{i,Q-2}, S^{\mathrm{tr}}_i)\big) - \big(\widetilde{w}^{t,\prime}_{i,Q-2} - \alpha\nabla\widehat{\mathcal{L}}(\widetilde{w}^{t,\prime}_{i,Q-2}, \widetilde{S}^{\mathrm{tr}}_i)\big) \| + \frac{2L}{n^{\mathrm{tr}}} \right] \\
\leq& \alpha(1+\alpha L)^{Q-1}\left[ \rho\| \widetilde{w}^t_{i,Q-2} - \widetilde{w}^{t,\prime}_{i,Q-2} \| + \frac{2(L+G\alpha\rho)}{n^{\mathrm{tr}}} \right] \\
\leq& \alpha(1+\alpha L)^{Q-1}\left[ \rho\| \big[\widetilde{w}^t - \alpha\nabla\widehat{\mathcal{L}}(\widetilde{w}^t, S^{\mathrm{tr}}_i)\big] - \big[\widetilde{w}^t - \alpha\nabla\widehat{\mathcal{L}}(\widetilde{w}^t, \widetilde{S}^{\mathrm{tr}}_i)\big] \| + \frac{2(L+G\alpha\rho(Q-2))}{n^{\mathrm{tr}}} \right] \\
\leq& \alpha(1+\alpha L)^{Q-1}\left[ \alpha\rho\| \nabla\widehat{\mathcal{L}}(\widetilde{w}^t, S^{\mathrm{tr}}_i) - \nabla\widehat{\mathcal{L}}(\widetilde{w}^t, \widetilde{S}^{\mathrm{tr}}_i) \| + \frac{2(L+G\alpha\rho(Q-2))}{n^{\mathrm{tr}}} \right] \\
\leq& \alpha(1+\alpha L)^{Q-1}\frac{2(L+G\alpha\rho(Q-1))}{n^{\mathrm{tr}}},
\end{aligned}
$$

where we use Assumption 2 to derive the inequalities from the second to the last step. Putting it in to $V(Q)$ and Unrolling it, noting that $V(1) = \| \alpha\nabla^2\widehat{\mathcal{L}}(\widetilde{w}^t, S^{\mathrm{tr}}_i) - \alpha\nabla^2\widehat{\mathcal{L}}(\widetilde{w}^t, \widetilde{S}^{\mathrm{tr}}_i) \| \leq \frac{2\alpha L}{n^{\mathrm{tr}}}$, then we

have

$$V(Q) \leq (1 + \alpha L)^{Q-1} V(1) + \sum_{k=1}^{Q-1} (1 + \alpha L)^{Q-1} \frac{2\alpha(L + G\alpha\rho(Q - k))}{n^{\text{tr}}}$$

$$\leq \frac{2\alpha L(1 + \alpha L)^{Q-1}}{n^{\text{tr}}} + \sum_{k=1}^{Q-1} (1 + \alpha L)^{Q-1} \frac{2\alpha(L + G\alpha\rho k)}{n^{\text{tr}}} \qquad (35)$$

$$= \sum_{k=0}^{Q-1} (1 + \alpha L)^{Q-1} \frac{2\alpha(L + G\alpha\rho k)}{n^{\text{tr}}}.$$

By plugging (34) and (35) into (33), then we have

$$\mathbb{E}\|\nabla\ell\big(w_{\mathcal{T}_i}(\widetilde{w}^t, S_i^{\text{tr}}), z^{\text{ts}}\big) - \nabla\ell\big(w_{\mathcal{T}_i}(\widetilde{w}^t, \widetilde{S}_i^{\text{tr}}), z^{\text{ts}}\big)\| \leq (1 + \alpha L)^{Q-1} \left[ \frac{2(1 + \alpha L)Q\alpha LG}{n^{\text{tr}}} + \sum_{k=0}^{Q-1} \frac{2\alpha G(L + G\alpha\rho k)}{n^{\text{tr}}} \right]$$

$$\leq (1 + \alpha L)^{Q-1} \frac{(6QG + Q^2\alpha^2 G^2\rho)}{n^{\text{tr}}}.$$
(36)

where we using $\alpha L \leq 1$ in the second inequality. Substituting (36) and (32) into (31), we have

$$\mathbb{E}\|w^{t+1} - \widetilde{w}^{t+1}\| \leq \mathbb{E}\|w^t - \widetilde{w}^t\| + \eta_t(1 + \alpha L)^{Q-1} \frac{(6QG + Q^2\alpha^2 G^2\rho)}{n^{\text{tr}}} + 2\eta_t \mathbb{E}\|\nabla\widehat{F}_i(w^t)\|.$$

Combing the above two cases, we obtain

$$\mathbb{E}\|w^{t+1} - \widetilde{w}^{t+1}\| \leq (1 - \frac{1}{m})\mathbb{E}\|w^t - \widetilde{w}^t\| + \frac{1}{m}\mathbb{E}\|w^t - \widetilde{w}^t\|$$

$$+ \frac{1}{m}\eta_t(1 + \alpha L)^{Q-1} \frac{(6QG + Q^2\alpha^2 G^2\rho)}{n^{\text{tr}}} + \frac{2}{m}\eta_t \mathbb{E}\|\nabla\widehat{F}_i(w^t)\|$$

$$= \mathbb{E}\|w^t - \widetilde{w}^t\| + \eta_t(1 + \alpha L)^{Q-1} \frac{(6QG + Q^2\alpha^2 G^2\rho)}{mn^{\text{tr}}} + \frac{2\eta_t}{m}\mathbb{E}\|\nabla\widehat{F}_i(w^t)\|.$$

Unrolling it and noting that $\|w^0 - \widetilde{w}^0\| = 0$, we have

$$\mathbb{E}[\|w^T - \widetilde{w}^T\|] \leq \sum_{t=0}^{T-1} \eta_t(1 + \alpha L)^{Q-1} \frac{(6QG + Q^2\alpha^2 G^2\rho)}{mn^{\text{tr}}} + \sum_{t=0}^{T-1} \frac{2\eta_t}{m}\mathbb{E}\|\nabla\widehat{F}\big(w^t, S_i\big)\|$$

$$\leq \sum_{t=0}^{T-1} \eta_t(1 + \alpha L)^{Q-1} \frac{(6QG + Q^2\alpha^2 G^2\rho)}{mn^{\text{tr}}} + \frac{1}{m}\sqrt{F(w^0) - \min_{\mathcal{W}} F + \frac{L_Q\sigma^2}{2}\sum_{t=0}^{T-1}\eta_t^2},$$

where we use Lemma 13 in the last inequality. Now we are ready to conclude. For any $i \in [m]$, we have

$$\epsilon_{gen} \leq \mathbb{E}\|\ell(w_{\mathcal{T}_i}(w^T, \widetilde{S}_i^{\text{tr}}), \widetilde{z}) - \ell(w_{\mathcal{T}_i}(\widetilde{w}^T, \widetilde{S}_i^{\text{tr}}), \widetilde{z})\|$$

$$\leq G\mathbb{E}\|w^T - \widetilde{w}^T\|$$

which completes the proof.

## B.4 Proof of Theorem 3

In this section, we establish stability results that do not rely on convexity, and we consider two cases. For the first case, using Lemma 10 and Lemma 14, we have

$$\|w^{t+1} - \widetilde{w}^{t+1}\| = \|\big(w^t - \eta_t\nabla\widehat{\mathcal{L}}\big(w_{\mathcal{T}_j}(w^t, S_j^{\text{tr}}), S_j^{\text{ts}}\big)\big) - \big(\widetilde{w}^t - \eta_t\nabla\widehat{\mathcal{L}}\big(w_{\mathcal{T}_j}(\widetilde{w}^t, S_j^{\text{tr}}), S_j^{\text{ts}}\big)\big)\|$$
$$\leq (1 + \eta_t\phi_t)\|w^t - \widetilde{w}^t\|,$$
(37)

where $\phi_t = \min\{L_Q, \xi_t\}$ with $\xi_t = \|\nabla^2 F(w_0, S_t)\| + \frac{\rho_Q}{2}\left\|\sum_{l=1}^{t-1}\beta_l\nabla\widehat{F}(w_{\mathcal{S}}^l)\right\| + \frac{\rho_Q}{2}\left\|\sum_{l=1}^{t-1}\beta_l\nabla\widehat{F}(w_{\widetilde{\mathcal{S}}}^l)\right\|$, $\rho_Q = \left[\frac{3\rho(1+\alpha L)^{2(Q-1)}}{L} + (1 + \alpha L)^{3Q}\rho + \alpha\kappa(1 + \alpha L)^{2q} + \frac{3\alpha\rho^2(1+\alpha L)^{2(Q-1)}}{L}\right]$, $L_Q = \frac{3\rho(1+\alpha L)^{2(Q-1)}}{L} +$

$(1 + \alpha L)^Q L$.

Next, for the second case, similar to the proof of (31), we have

$$\mathbb{E}\|w^{t+1} - \widetilde{w}^{t+1}\| \leq \frac{1}{n^{\mathrm{ts}}} \sum_{z \in S_i^{\mathrm{ts}}} \mathbb{E}\|\big(w^t - \eta_t \nabla\ell\big(w_{\mathcal{T}_i}(w^t, S_i^{\mathrm{tr}}), z^{\mathrm{ts}}\big)\big) - \big(\widetilde{w}^t - \eta_t \nabla\ell\big(w_{\mathcal{T}_i}(\widetilde{w}^t, \widetilde{S}_i^{\mathrm{tr}}), z^{\mathrm{ts}}\big)\big)\| $$
$$+ 2\eta_t \mathbb{E}\|\nabla \widehat{F}_i(w)\|. \tag{38}$$

Note that

$$\mathbb{E}\|\big(w^t - \eta_t \nabla\ell\big(w_{\mathcal{T}_i}(w^t, S_i^{\mathrm{tr}}), z^{\mathrm{ts}}\big)\big) - \big(\widetilde{w}^t - \eta_t \nabla\ell\big(w_{\mathcal{T}_i}(\widetilde{w}^t, \widetilde{S}_i^{\mathrm{tr}}), z^{\mathrm{ts}}\big)\big)\|$$
$$\leq \mathbb{E}\|\big(w^t - \eta_t \nabla\ell\big(w_{\mathcal{T}_i}(w^t, S_i^{\mathrm{tr}}), z^{\mathrm{ts}}\big)\big) - \big(\widetilde{w}^t - \eta_t \nabla\ell\big(w_{\mathcal{T}_i}(\widetilde{w}^t, S_i^{\mathrm{tr}}), z^{\mathrm{ts}}\big)\big)\| \tag{39}$$
$$+ \eta_t \mathbb{E}\|\nabla\ell\big(w_{\mathcal{T}_i}(\widetilde{w}^t, S_i^{\mathrm{tr}}), z^{\mathrm{ts}}\big) - \nabla\ell\big(w_{\mathcal{T}_i}(\widetilde{w}^t, \widetilde{S}_i^{\mathrm{tr}}), z^{\mathrm{ts}}\big)\|.$$

Let us bound the two terms on the RHS of (39), separately. First, similar to how we bound (37), we could bound the first term by

$$\mathbb{E}\|\big(w^t - \eta_t \nabla\ell\big(w_{\mathcal{T}_i}(w^t, S_i^{\mathrm{tr}}), z^{\mathrm{ts}}\big)\big) - \big(\widetilde{w}^t - \eta_t \nabla\ell\big(w_{\mathcal{T}_i}(\widetilde{w}^t, S_i^{\mathrm{tr}}), z^{\mathrm{ts}}\big)\big)\| \leq (1 + \eta_t \phi_t)\mathbb{E}\|w^t - \widetilde{w}^t\|.$$

To bound the second term on the RHS of (39), similar to the proof of (33), we have

$$\mathbb{E}\|\nabla\ell\big(w_{\mathcal{T}_i}(\widetilde{w}^t, S_i^{\mathrm{tr}}), z^{\mathrm{ts}}\big) - \nabla\ell\big(w_{\mathcal{T}_i}(\widetilde{w}^t, \widetilde{S}_i^{\mathrm{tr}}), z^{\mathrm{ts}}\big)\|$$
$$\leq (1 + \alpha L)^Q \mathbb{E}\|\nabla\ell(\widetilde{w}_{\mathcal{T}_i, Q}^t, z^{\mathrm{ts}}) - \nabla\ell(\widetilde{w}_{i,Q}^{t,\prime}, z^{\mathrm{ts}})\|$$
$$+ G\mathbb{E}\| \underbrace{\prod_{q=0}^{Q-1} (I - \alpha\nabla^2\widehat{\mathcal{L}}(\widetilde{w}_{\mathcal{T}_i, Q}^t, S_i^{\mathrm{tr}})) - \prod_{q=0}^{Q-1} (I - \alpha\nabla^2\widehat{\mathcal{L}}(\widetilde{w}_{i,q}^{t,\prime}, \widetilde{S}_i^{\mathrm{tr}}))}_{V(Q)}\| \tag{40}$$

where we use Assumption 2 in the last inequality. Hence, what remains is to bound the two terms in (40). To do so, notice that

$$\mathbb{E}\|\nabla\ell(\widetilde{w}_{\mathcal{T}_i, Q}^t, z^{\mathrm{ts}}) - \nabla\ell(\widetilde{w}_{i,Q}^{t,\prime}, z^{\mathrm{ts}})\|$$
$$\leq L\mathbb{E}\|\widetilde{w}_{\mathcal{T}_i, Q}^t - \widetilde{w}_{i,Q}^{t,\prime}\|$$
$$= L\mathbb{E}\| \left[\widetilde{w}_{i,Q-1}^t - \alpha\nabla\widehat{\mathcal{L}}(\widetilde{w}_{i,Q-1}^t, S_i^{\mathrm{tr}})\right] - \left[\widetilde{w}_{i,Q-1}^{t,\prime} - \alpha\nabla\widehat{\mathcal{L}}(\widetilde{w}_{i,Q-1}^{t,\prime}, \widetilde{S}_i^{\mathrm{tr}})\right] \|$$
$$\leq L\mathbb{E}\| \left[\widetilde{w}_{i,Q-1}^t - \alpha\nabla\widehat{\mathcal{L}}(\widetilde{w}_{i,Q-1}^t, S_i^{\mathrm{tr}}))\right] - \left[\widetilde{w}_{i,Q-1}^{t,\prime} - \alpha\nabla\ell(\widetilde{w}_{i,Q-1}^{t,\prime}, S_i^{\mathrm{tr}})\right] \|$$
$$+ L\mathbb{E}\|\alpha\nabla\widehat{\mathcal{L}}(\widetilde{w}_{i,Q-1}^{t,\prime}, S_i^{\mathrm{tr}}) - \alpha\nabla\widehat{\mathcal{L}}(\widetilde{w}_{i,Q-1}^{t,\prime}, \widetilde{S}_i^{\mathrm{tr}})\|$$
$$\leq L(1 + \alpha L)\mathbb{E}\|\widetilde{w}_{i,Q-1}^t - \widetilde{w}_{i,Q-1}^{t,\prime}\| + \frac{2\alpha L G}{n^{\mathrm{tr}}}$$
$$\leq L(1 + \alpha L)^Q \mathbb{E}\| \left[\widetilde{w}^t - \alpha\nabla\widehat{\mathcal{L}}(\widetilde{w}^t, S_i^{\mathrm{tr}})\right] - \left[\widetilde{w}^t - \alpha\nabla\widehat{\mathcal{L}}(\widetilde{w}^t, \widetilde{S}_i^{\mathrm{tr}})\right] \| + \frac{2(Q-1)(1 + \alpha L)^{Q-1}\alpha L G}{n^{\mathrm{tr}}}$$
$$= \alpha L(1 + \alpha L)^Q \mathbb{E}\|\nabla\widehat{\mathcal{L}}(\widetilde{w}^t, S_i^{\mathrm{tr}}) - \nabla\widehat{\mathcal{L}}(\widetilde{w}^t, \widetilde{S}_i^{\mathrm{tr}})\| + \frac{2(Q-1)(1 + \alpha L)^{Q-1}\alpha L G}{n^{\mathrm{tr}}}$$
$$\leq \frac{2Q(1 + \alpha L)^Q \alpha L G}{n^{\mathrm{tr}}}, \tag{41}$$

Next, we have

$$V(Q) \leq (1 + \alpha L)^{Q-1}\|\alpha\nabla^2\widehat{\mathcal{L}}(\widetilde{w}_{i,Q-1}^t, S_i^{\mathrm{tr}}) - \alpha\nabla^2\widehat{\mathcal{L}}(\widetilde{w}_{i,Q-1}^{t,\prime}, \widetilde{S}_i^{\mathrm{tr}})\| + (1 + \alpha L)V(Q-1), \tag{42}$$

To bound the above inequality, we first consider the first term,

$$(1 + \alpha L)^{Q-1}\|\alpha\nabla^2\widehat{\mathcal{L}}(\widetilde{w}_{i,Q-1}^t, S_i^{\mathrm{tr}}) - \alpha\nabla^2\widehat{\mathcal{L}}(\widetilde{w}_{i,Q-1}^{t,\prime}, \widetilde{S}_i^{\mathrm{tr}})\| \leq \alpha(1 + \alpha L)^{Q-1}\frac{2(L + G\alpha\rho(Q-1))}{n^{\mathrm{tr}}},$$

Putting it in to $V(Q)$ and Unrolling it, noting that $V(1) = \|\alpha\nabla^2\widehat{\mathcal{L}}(\widetilde{w}^t, S_i^{\text{tr}}) - \alpha\nabla^2\widehat{\mathcal{L}}(\widetilde{w}^t, \widetilde{S}_i^{\text{tr}})\| \le \frac{2\alpha L}{n^{\text{tr}}}$, then we have

$$
\begin{aligned}
V(Q) \le& (1+\alpha L)^{Q-1}V(1) + \sum_{k=1}^{Q-1}(1+\alpha L)^{Q-1}\frac{2\alpha(L+G\alpha\rho(Q-k))}{n^{\text{tr}}} \\
\le& \frac{2\alpha L(1+\alpha L)^{Q-1}}{n^{\text{tr}}} + \sum_{k=1}^{Q-1}(1+\alpha L)^{Q-1}\frac{2\alpha(L+G\alpha\rho k)}{n^{\text{tr}}} \\
=& \sum_{k=0}^{Q-1}(1+\alpha L)^{Q-1}\frac{2\alpha(L+G\alpha\rho k)}{n^{\text{tr}}}.
\end{aligned}
\tag{43}
$$

By plugging (41) and (42) into (40), then we have

$$
\begin{aligned}
\mathbb{E}\|\nabla\ell\big(w_{\mathcal{T}_i}(\widetilde{w}^t, S_i^{\text{tr}}), z^{\text{ts}}\big) - \nabla\ell\big(w_{\mathcal{T}_i}(\widetilde{w}^t, \widetilde{S}_i^{\text{tr}}), z^{\text{ts}}\big)\| &\le (1+\alpha L)^{2(Q-1)}\left[\frac{2(1+\alpha L)Q\alpha LG}{n^{\text{tr}}} + \sum_{k=0}^{Q-1}\frac{2\alpha G(L+G\alpha\rho k)}{n^{\text{tr}}}\right] \\
&\le (1+\alpha L)^{2(Q-1)}\frac{(6QG + Q^2\alpha^2 G^2\rho)}{n^{\text{tr}}}.
\end{aligned}
\tag{44}
$$

where we using $\alpha L \le 1$ in the second inequality. Substituting (44) and (39) into (38), we obtain

$$
\mathbb{E}\|w^{t+1} - \widetilde{w}^{t+1}\| \le (1+\eta_t\phi_t)\mathbb{E}\|w^t - \widetilde{w}^t\| + \eta_t(1+\alpha L)^{2(Q-1)}\frac{(6QG + Q^2\alpha^2 G^2\rho)}{n^{\text{tr}}} + 2\eta_t\mathbb{E}\|\nabla\widehat{F}_i(w^t)\|.
$$

From Lemma 7, we can know $\mathbb{E}\|\nabla\widehat{F}_i(w^t)\| \le eG$. Then combing the above two cases, we obtain

$$
\begin{aligned}
\mathbb{E}\|w^{t+1} - \widetilde{w}^{t+1}\| \le& (1-\frac{1}{m})(1+\eta_t\phi_t)\mathbb{E}\|w^t - \widetilde{w}^t\| + \frac{1}{m}(1+\eta_t\phi_t)\mathbb{E}\|w^t - \widetilde{w}^t\| \\
&+ \frac{1}{m}\eta_t(1+\alpha L)^{2(Q-1)}\frac{(6QG+Q^2\alpha^2G^2\rho)}{n^{\text{tr}}} + \frac{2\eta_t eG}{m} \\
\le& \exp(\eta_t\phi_t)\mathbb{E}\|w^t - \widetilde{w}^t\| + \frac{\eta_t\Phi}{m}.
\end{aligned}
$$

where we use $1 + x \le \exp(x)$ and $\alpha \le \frac{1}{QL}$ in the second inequality, and we denote $\Phi = 2eG + (1+\alpha L)^{2(Q-1)}\frac{(6QG+Q^2\alpha^2G^2\rho)}{n^{\text{tr}}}$. Following the same proof technique in [47](Eq (23) in Theorem 4), we can easily get

$$
\begin{aligned}
\mathbb{E}\|w^T - \widetilde{w}^T\| \le& \sum_{t=t_0+1}^{T}\exp(2c\gamma\sum_{l=t+1}^{T}\frac{1}{k})\frac{2c\Phi}{mt} \\
\le& \sum_{t=t_0+}^{T}\exp(2c\gamma\ln(\frac{T}{t}))\frac{2c\Phi}{mt} \\
=& \frac{2c\Phi}{m}(T^{2c\gamma})\sum_{t=t_0+}^{T}t^{-2c\gamma-1} \\
\le& \frac{1}{2c\gamma}\frac{2c\Phi}{m}(\frac{T}{t_0})^{2c\gamma}
\end{aligned}
$$

and

$$
\mathbb{E}\|\ell(w_{\mathcal{T}_i}(w^T, \widetilde{S}_i^{\text{tr}}), \widetilde{z}) - \ell(w_{\mathcal{T}_i}(\widetilde{w}^T, \widetilde{S}_i^{\text{tr}}), \widetilde{z})\| \le \frac{eG\Phi}{\gamma m}(\frac{T}{t_0})^{2c\gamma} + r\frac{t_0}{m},
\tag{45}
$$

where $r = \mathbb{E}_{\mathcal{S},\mathcal{A}}[F(w_{\mathcal{S}})], \gamma = \mathcal{O}(\min\{L_Q, \mathbb{E}_S[\|\nabla^2 F_i(w^0, S\|] + \rho_Q(c\sigma + \sqrt{c(F(w^0) - \min_{\mathcal{W}} F)})\})$. Next, let $b = 2c\gamma$. Then, setting

$$
t_0 = (\frac{2ceG\Phi}{r})^{\frac{1}{1+b}}T^{\frac{b}{1+b}}
$$

minimizes (45). Plugging $t_0$ back we get that (44) equals to

$$
\frac{1+\frac{1}{b}}{m}(2ceG\Phi)^{\frac{1}{1+b}}(rT)^{\frac{b}{1+b}}
$$

This completes the proof.

# C Stability and Generalization of PDF

## C.1 Proof of stability of PDF

To show the claim, it just suffices to show that for any $i$, we have

$$\mathbb{E}_{\mathcal{A},\mathcal{S}}\left[F_i(w_{\mathcal{S}}) - \widehat{F}_i(w_{\mathcal{S}}, S_i)\right] \leq \epsilon.$$

Take the dataset $\widetilde{\mathcal{S}}^{(i)}$ which is the same as $\mathcal{S}$, except that $\widetilde{S}_i$ differ from $S_i$ in at most one data point. In particular,

$$S_i^{\mathrm{tr}} = \{z_{i,1}, ..., z_{i,n}\}, \widetilde{S}_i = \{z_{i,1}, ..., \widetilde{z}_{i,j}, ..., z_{i,n}\}.$$

Then, we relate empirical risk and population risk by

$$
\begin{aligned}
\mathbb{E}_{\mathcal{S},\mathcal{A}}[\widehat{F}_i(w_{\mathcal{S}}, z_{i,j})] &= \frac{1}{n}\sum_{j=1}^{n}\mathbb{E}_{\mathcal{S},\mathcal{A}}\ell_\lambda(w_{\mathcal{S}}, z_{i,j}) \\
&= \frac{1}{n}\sum_{j=1}^{n}\mathbb{E}_{\mathcal{S},\mathcal{A},\widetilde{z}_{i,j}}\ell_\lambda(w_{\widetilde{\mathcal{S}}}, \widetilde{z}_{i,j}).
\end{aligned}
\tag{46}
$$

Moreover, we have

$$\mathbb{E}_{\mathcal{A},\mathcal{S}}[F_i(w_{\mathcal{S}})] = \frac{1}{n}\sum_{j=1}^{n}\mathbb{E}_{\mathcal{S},\mathcal{A},\widetilde{z}_{i,j}}\ell_\lambda(w_{\mathcal{S}}, \widetilde{z}_{i,j}). \tag{47}$$

Putting (46) and (47) together, we have

$$\mathbb{E}_{\mathcal{A},\mathcal{S}}\left[F(w_{\mathcal{S}}) - \widehat{F}(w_{\mathcal{S}}, \mathcal{S}_i)\right] \leq \frac{1}{m}\sum_{i=1}^{m}\frac{1}{n}\sum_{j=1}^{n}\mathbb{E}_{\mathcal{S},\mathcal{A},\widetilde{z}_{i,j}}\ell_\lambda(w_{\mathcal{S}}, \widetilde{z}_{i,j}) - \ell_\lambda(w_{\widetilde{\mathcal{S}}}, \widetilde{z}_{i,j})$$

$$\leq \epsilon.$$

Then we obtain the desired result.

## C.2 Lemmas

**Lemma 16.** *Assume that $\hat{\mathcal{L}}$ is differentiable and $w_{\mathcal{T}}^*$ is the unique minimizer of $\hat{\mathcal{L}}(w_{\mathcal{T}}) + \frac{\lambda}{2}\|w_{\mathcal{T}} - w\|^2$. Then the gradient of $\widehat{\mathcal{L}}_\lambda(w) = \widehat{\mathcal{L}}(w_{\mathcal{T}}^*) + \frac{\lambda}{2}\|w_{\mathcal{T}}^* - w\|^2$ is given by $\nabla\hat{\mathcal{L}}_\lambda = \lambda(w - w_{\mathcal{T}}^*)$*

*Proof.* Since $\hat{\mathcal{L}}$ is differentiable, from the first-order optimality condition we know that

$$\nabla\widehat{\mathcal{L}}(w_{\mathcal{T}_i}^*) + \lambda(w_{\mathcal{T}}^* - w) = 0$$

From the chain rule we have

$$
\begin{aligned}
\nabla\widehat{\mathcal{L}}_\lambda(w) &= (\frac{\partial w_{\mathcal{T}^*}}{\partial w})^\top\nabla\hat{\mathcal{L}}(w_{\mathcal{T}}^*) + \lambda\left(I - \left(\frac{\partial w_{\mathcal{T}}^*}{\partial w}\right)^\top\right)(w - w_{\mathcal{T}}^*) \\
&= \lambda(w - w_{\mathcal{T}}^*) + (\frac{\partial w_{\mathcal{T}^*}}{\partial w})^\top(\nabla\hat{\mathcal{L}}(w_{\mathcal{T}}^*) + \lambda(w - w_{\mathcal{T}}^*)) \\
&= \lambda(w - w_{\mathcal{T}}^*)
\end{aligned}
$$

$\square$

**Lemma 17.** *Suppose Assumption 2 hold. Then if $\lambda \geq L$, $\ell_\lambda(w, z) = \ell(w_{\mathcal{T}}^*, z) + \frac{\lambda}{2}\|w_{\mathcal{T}_i}^* - w\|^2$ is $L_Q$-smoothness with $L_Q = \frac{\lambda L}{\lambda + L}$ and $\rho$-Lipschitz Hessian with respect to $w$, where $w_{\mathcal{T}_i}^* = argmin_{w_{\mathcal{T}_i}}\ell(w_{\mathcal{T}_i}, z) + \frac{\lambda}{2}\|w_{\mathcal{T}_i} - w\|^2$.*

*Proof.* From the first-order optimiality condition we know that

$$\nabla\ell(w_{\mathcal{T}_i}^*) + \lambda(w_{\mathcal{T}_i}^* - w) = 0.$$

Therefore, we can further obtain

$$\nabla^2 \ell(w^*_{\mathcal{T}_i}) \frac{\partial w^*_{\mathcal{T}_i}}{\partial w} + \lambda(\frac{\partial w^*_{\mathcal{T}_i}}{\partial w} - I) = 0.$$

and

$$\nabla^3 \ell(w^*_{\mathcal{T}_i}) \left(\frac{\partial w^*_{\mathcal{T}_i}}{\partial w}\right)^2 + \nabla^2 \ell(w^*_{\mathcal{T}_i}) \frac{\partial^2 w^*_{\mathcal{T}_i}}{\partial w^2} + \lambda \frac{\partial^2 w^*_{\mathcal{T}_i}}{\partial w^2} = 0.$$

This implies

$$\frac{\partial w^*_{\mathcal{T}_i}}{\partial w} = \lambda(\nabla^2 \ell(w^*_{\mathcal{T}_i}) + \lambda I)^{-1}, \ \frac{\partial^2 w^*_{\mathcal{T}}}{\partial w^2} = -\left(\nabla^2 \ell(w^*_{\mathcal{T}_i}) + \lambda I\right)^{-1} \nabla^3 \ell(w^*_{\mathcal{T}_i}) \left(\frac{\partial w^*_{\mathcal{T}_i}}{\partial w}\right)^2.$$

From Lemma , we have $\nabla \ell_\lambda(w) = \lambda(w - w^*_{\mathcal{T}_i})$. Therefore, we can further have

$$\nabla^2 \ell_\lambda(w) = \lambda(I - \frac{\partial w^*_{\mathcal{T}_i}}{\partial w}) = \lambda(I - \lambda(\nabla^2 \ell(w^*_{\mathcal{T}_i}) + \lambda I)^{-1})$$

and

$$\nabla^3 \ell_\lambda(w) = -\lambda \frac{\partial^2 w^*_{\mathcal{T}_i}}{\partial w^2} = \lambda^3 (\nabla^2 \ell(w^*_{\mathcal{T}_i}) + \lambda I)^{-3} \nabla^3 \ell(w^*_{\mathcal{T}_i}).$$

Note that $\ell(w_{\mathcal{T}_i})$ is $L$-smooth and $\rho$-Hessian Lipschitz with respect to $w_{\mathcal{T}_i}$. Then it yields

$$\|\nabla^2 \ell_\lambda(w)\| \le \frac{\lambda L}{\lambda + L}, \ \|\nabla^3 \ell_\lambda(w)\| \le \rho$$

$\square$

## C.3 Proof of Theorem 4

To facilitate the analysis of stability, we rewrite $\widehat{\mathcal{K}}(w_{\mathcal{T}}, w, S)) = \widehat{\mathcal{L}}(w_{\mathcal{T}}, S) + \frac{\lambda}{2}\|w_{\mathcal{T}} - w\|^2, \widehat{\mathcal{L}}_\lambda(w, S) = \min_{w_{\mathcal{T}}}\{\widehat{\mathcal{L}}(w_{\mathcal{T}}, S) + \frac{\lambda}{2}\|w_{\mathcal{T}} - w\|^2\}$. From Lemma 17, we can know $\hat{\mathcal{L}}_\lambda$ is $L_Q$-smooth. Hence, by using Lemma 1, for the first case that task $\mathcal{T}_j$ is selected, we have

$$
\begin{aligned}
\mathbb{E}\|w^{t+1} - \widetilde{w}^{t+1}\| =& \mathbb{E}\|\left(w^t - \eta_t \nabla\widehat{\mathcal{K}}\left(w^t_{\mathcal{T}_j, Q}, w^t; S_j\right)\right) - \left(\widetilde{w}^t - \eta_t \nabla\widehat{\mathcal{K}}\left(w^t_{\mathcal{T}_j, Q}, \widetilde{w}^t; S_j\right)\right)\| \\
\le& \mathbb{E}\|\left(w^t - \eta_t \nabla\widehat{\mathcal{L}}_\lambda\left(w^t, S_j\right)\right) - \left(\widetilde{w}^t - \eta_t \nabla\widehat{\mathcal{L}}_\lambda\left(\widetilde{w}^t, S_j\right)\right)\| \\
& + 2\eta_t \mathbb{E}\|\nabla\widehat{\mathcal{K}}\left(w^t_{\mathcal{T}_j, Q}, w^t; S_j\right) - \nabla\widehat{\mathcal{L}}_\lambda\left(w^t, S_j\right)\| \\
=& \mathbb{E}\|\left(w^t - \eta_t \nabla\widehat{\mathcal{L}}_\lambda\left(w^t, S_j\right)\right) - \left(\widetilde{w}^t - \eta_t \nabla\widehat{\mathcal{L}}_\lambda\left(\widetilde{w}^t, S_j\right)\right)\| \\
& + 2\eta_t \lambda \mathbb{E}\|w^t_{j, Q} - w^*_{\mathcal{T}_i}(w^t)\| \\
\le& \mathbb{E}\|w^t - \widetilde{w}^t\| + 2\eta_t \lambda \|w^t_{j, Q} - w^*_{\mathcal{T}_i}(w^t)\|
\end{aligned}
\tag{48}
$$

where $w^*_{\mathcal{T}}(w) = \arg\min_{w_{\mathcal{T}}} \widehat{\mathcal{L}}(w_{\mathcal{T}}, S) + \frac{\lambda}{2}\|w_{\mathcal{T}} - w\|$. where we use Lemma 1 in the second inequality. In the second case, we have

$$
\begin{aligned}
\mathbb{E}\|w^{t+1} - \widetilde{w}^{t+1}\| =& \mathbb{E}\|\left(w^t - \eta_t \nabla\mathcal{K}\left(w^t_{\mathcal{T}_i, Q}, w^t; S_i\right)\right) - \left(\widetilde{w}^t - \eta_t \nabla\mathcal{K}\left(w^{t,'}_{\mathcal{T}_i, Q}, w^t; \widetilde{S}_i\right)\right)\| \\
\le& \mathbb{E}\|\left(w^t - \eta_t \nabla\widehat{\mathcal{L}}_\lambda\left(w^t, S_i\right)\right) - \left(\widetilde{w}^t - \eta_t \nabla\widehat{\mathcal{L}}_\lambda\left(\widetilde{w}^t, \widetilde{S}_i\right)\right)\| \\
& + 2\mathbb{E}\|\nabla\mathcal{K}\left(w^t_{\mathcal{T}_i, Q}, w^t; S_i\right) - \nabla\widehat{\mathcal{L}}_\lambda\left(w^t, S_i\right)\|.
\end{aligned}
\tag{49}
$$

For the first term in (49), we have

$$
\begin{aligned}
& \mathbb{E}\|\left(w^t - \eta_t \nabla\widehat{\mathcal{L}}_\lambda\left(w^t, S_i\right)\right) - \left(\widetilde{w}^t - \eta_t \nabla\widehat{\mathcal{L}}_\lambda\left(\widetilde{w}^t, \widetilde{S}_i\right)\right)\| \\
\le& \mathbb{E}\|\left(w^t - \eta_t \nabla\widehat{\mathcal{L}}_\lambda\left(w^t, S_i\right)\right) - \left(\widetilde{w}^t - \eta_t \nabla\widehat{\mathcal{L}}_\lambda\left(\widetilde{w}^t, S_i\right)\right)\| + \eta_t \mathbb{E}\|\nabla\widehat{\mathcal{L}}_\lambda\left(\widetilde{w}^t, S_i\right) - \nabla\widehat{\mathcal{L}}_\lambda\left(w^t, \widetilde{S}_i\right)\| \\
\le& \mathbb{E}\|w^t - \widetilde{w}^t\| + 2\eta_t \mathbb{E}\|\nabla\widehat{F}_i(w^t, S_i)\|
\end{aligned}
\tag{50}
$$

For the second term in (50), we have

$$\mathbb{E}\|\nabla\mathcal{K}\left(\Phi_Q(w^t, j), w^t; j\right) - \nabla\widehat{\mathcal{L}}_\lambda\left(w^t, i\right)\| \le \lambda \mathbb{E}\|w^t_{i, Q} - w^*_{\mathcal{T}_i}(w^t)\| \tag{51}$$

Putting (50) and (51) into (49), then we have

$$\mathbb{E}\|w^{t+1} - \widetilde{w}^{t+1}\| \le \mathbb{E}\|w^t - \widetilde{w}^t\| + 2\eta_t \mathbb{E}\|\nabla \widehat{F}_i(w^t, S_i)\| + 2\eta_t \lambda \mathbb{E}\|w_{i,Q}^t - w_{\mathcal{T}_i}^*(w^t)\| \qquad (52)$$

Combining (48) and (52), we obtain

$$\mathbb{E}\|w^{t+1} - \widetilde{w}^{t+1}\| \le (1 - \frac{1}{m})\mathbb{E}\|w^t - \widetilde{w}^t\| + \frac{1}{m}\mathbb{E}\|w^t - \widetilde{w}^t\| + \frac{1}{m}\left[2\eta_t \mathbb{E}\|\nabla \widehat{F}(w^t, _i)\| + 2\eta_t \lambda \mathbb{E}\|w_{i,Q}^t - w_{\mathcal{T}_i}^*(w^t)\|\right].$$

$$= \mathbb{E}\|w^t - \widetilde{w}^t\| + \frac{2\eta_t}{m}\mathbb{E}\|\nabla \widehat{F}(w^t, S_i)\| + \frac{2\eta_t \lambda}{m}\mathbb{E}\|w_{i,Q}^t - w_{\mathcal{T}_i}^*(w^t)\|.$$

Unrolling it and noting that $\|w^0 - \widetilde{w}^0\| = 0$, we have

$$\mathbb{E}[\|w^T - \widetilde{w}^T\|] \le \sum_{t=0}^{T-1}\frac{2\eta_t \lambda}{m}\mathbb{E}\|w_{i,Q}^t - w_{\mathcal{T}_i}^*(w^t)\| + \sum_{t=0}^{T-1}\frac{2\eta_t}{m}\mathbb{E}\|\nabla \widehat{F}(w^t, S_i)\|$$

$$\le \sum_{t=0}^{T-1}\frac{2\eta_t \lambda}{m}\mathbb{E}\|w_{i,Q}^t - w_{\mathcal{T}_i}^*(w^t)\| + \frac{1}{m}\sqrt{F(w^0) - \min_{\mathcal{W}} F + \frac{L_Q \sigma^2}{2}\sum_{t=0}^{T-1}\eta_t^2}$$

where we use Lemma 14 and Lemma 17 in the second inequality, and we denote $L_Q = \frac{\lambda L}{\lambda + L}$. Now we are ready to conclude. By using Lemma A, we have

$$\mathbb{E}\|\ell_\lambda(w^T, \widetilde{z}) - \ell_\lambda(\widetilde{w}^T, \widetilde{z})\| \le 2G\mathbb{E}\|w^T - \widetilde{w}^T\|$$

$$\le \sum_{t=0}^{T-1}\frac{2\eta_t \lambda}{m}\mathbb{E}\|w_{i,Q}^t - w_{\mathcal{T}_i}^*(w^t)\| + \frac{1}{m}\sqrt{F(w^0) - \min_{\mathcal{W}} F + \frac{L_Q \sigma^2}{2}\sum_{t=0}^{T-1}\eta_t^2}$$

Then we completes the proof.

## C.4   Proof of Theorem 5

For the non-convex case, under Lemma 17, for the first case we have

$$\mathbb{E}\|w^{t+1} - \widetilde{w}^{t+1}\| = \mathbb{E}\|\left(w^t - \eta_t \nabla\widehat{\mathcal{K}}\left(w_{\mathcal{T}_j,q}^t, w^t; S_j\right)\right) - \left(\widetilde{w}^t - \eta_t \nabla\widehat{\mathcal{K}}\left(w_{\mathcal{T}_j,q}^t, \widetilde{w}^t; S_j\right)\right)\|$$

$$\le \mathbb{E}\|\left(w^t - \eta_t \nabla\widehat{\mathcal{L}}_\lambda\left(w^t, S_j\right)\right) - \left(\widetilde{w}^t - \eta_t \nabla\widehat{\mathcal{L}}_\lambda\left(\widetilde{w}^t, S_j\right)\right)\|$$

$$+ 2\eta_t \mathbb{E}\|\nabla\widehat{\mathcal{K}}\left(w_{\mathcal{T}_j,q}^t, w^t; S_j\right) - \nabla\widehat{\mathcal{L}}_\lambda\left(w^t, S_j\right)\|.$$

$$= \mathbb{E}\|\left(w^t - \eta_t \nabla\widehat{\mathcal{L}}_\lambda\left(w^t, S_j\right)\right) - \left(\widetilde{w}^t - \eta_t \nabla\widehat{\mathcal{L}}_\lambda\left(\widetilde{w}^t, S_j\right)\right)\| \qquad (53)$$

$$+ 2\eta_t \lambda\|w_{j,Q}^t - w_{\mathcal{T}_i}^*(w^t)\|.$$

$$\le (1 + \eta_t \phi_t)\mathbb{E}\|w^t - \widetilde{w}^t\| + 2\eta_t \lambda\|w_{j,Q}^t - w_{\mathcal{T}_i}^*(w^t)\|$$

where $\phi_t = \min\{L_Q, \xi_t\}$ with $\xi_t = \|\nabla^2 F(w_0, S_t)\| + \frac{\rho}{2}\left\|\sum_{l=1}^{t-1}\beta_l \nabla\widehat{F}(w_{\mathcal{S}}^l)\right\| + \frac{\rho}{2}\left\|\sum_{l=1}^{t-1}\beta_l \nabla\widehat{F}(w_{\widetilde{\mathcal{S}}}^l)\right\|$, $w_{\mathcal{T}}^*(w) = \arg\min_{w_{\mathcal{T}}}\widehat{\mathcal{L}}(w_{\mathcal{T}}, S) + \frac{\lambda}{2}\|w_{\mathcal{T}} - w\|$. In the second case, we have

$$\mathbb{E}\|w^{t+1} - \widetilde{w}^{t+1}\| = \mathbb{E}\|\left(w^t - \eta_t \nabla\mathcal{K}\left(w_{\mathcal{T}_i,q}^t, w^t; S_i\right)\right) - \left(\widetilde{w}^t - \eta_t \nabla\mathcal{K}\left(w_{\mathcal{T}_i,q}^{t,\prime}, w^t; \widetilde{S}_i\right)\right)\|$$

$$\le \mathbb{E}\|\left(w^t - \eta_t \nabla\widehat{\mathcal{L}}_\lambda\left(w^t, S_i\right)\right) - \left(\widetilde{w}^t - \eta_t \nabla\widehat{\mathcal{L}}_\lambda\left(\widetilde{w}^t, \widetilde{S}_i\right)\right)\|$$

$$+ 2\mathbb{E}\|\nabla\mathcal{K}\left(w_{\mathcal{T}_i,q}^t, w^t; S_i\right) - \nabla\widehat{\mathcal{L}}_\lambda\left(w^t, S_i\right)\| \qquad (54)$$

$$\le \mathbb{E}\|\left(w^t - \eta_t \nabla\widehat{\mathcal{L}}_\lambda\left(w^t, S_i\right)\right) - \left(\widetilde{w}^t - \eta_t \nabla\widehat{\mathcal{L}}_\lambda\left(\widetilde{w}^t, \widetilde{S}_i\right)\right)\|$$

$$+ 2\eta_t \lambda\|w_{j,Q}^t - w_{\mathcal{T}_i}^*(w^t)\|$$

For the first term in (54),

$$\mathbb{E}\|\left(w^t - \eta_t \nabla\widehat{\mathcal{L}}_\lambda\left(w^t, _i\right)\right) - \left(\widetilde{w}^t - \eta_t \nabla\widehat{\mathcal{L}}_\lambda\left(\widetilde{w}^t, \widetilde{_i}\right)\right)\|$$

$$\le \mathbb{E}\|\left(w^t - \eta_t \nabla\widehat{\mathcal{L}}_\lambda\left(w^t, _i\right)\right) - \left(\widetilde{w}^t - \eta_t \nabla\widehat{\mathcal{L}}_\lambda\left(\widetilde{w}^t, _i\right)\right)\| + \eta_t \mathbb{E}\|\nabla\widehat{\mathcal{L}}_\lambda\left(\widetilde{w}^t, _i\right) - \nabla\widehat{\mathcal{L}}_\lambda\left(w^t, \widetilde{_i}\right)\|$$

Putting it into (54), then we have

$$\mathbb{E}\|w^{t+1} - \widetilde{w}^{t+1}\| \le (1 + \eta_t \phi_t)\mathbb{E}\|w^t - \widetilde{w}^t\| + 2\eta_t \mathbb{E}\|\nabla \widehat{F}_i(w^t, S_i)\| + 2\eta_t \lambda \|w_{j,Q}^t - w_{\mathcal{T}_i}^*(w^t)\| \quad (55)$$

we can know $\mathbb{E}\|\nabla \widehat{F}_i(w^t)\| \le G$. Then combing two cases, we obtain

$$\mathbb{E}\|w^{t+1} - \widetilde{w}^{t+1}\|$$
$$\le (1 - \frac{1}{m})(1 + \eta_t \phi_t)\mathbb{E}\|w^t - \widetilde{w}^t\| + \frac{1}{m}(1 + \eta_t \phi_t)\mathbb{E}\|w^t - \widetilde{w}^t\| + \frac{1}{m}\left[2\eta_t \mathbb{E}\|\nabla \widehat{F}(w^t,_i)\| + 2\eta_t \lambda \mathbb{E}\|w_{i,Q}^t - w_{\mathcal{T}_i}^*(w^t)\|\right].$$
$$\le (1 + \eta_t \phi_t)\mathbb{E}\|w^t - \widetilde{w}^t\| + \frac{1}{m}\left[2\eta_t \mathbb{E}\|\nabla \widehat{F}(w^t,_i)\| + \frac{2\eta_t \lambda G}{C^Q}\right]$$
$$\le \exp(\eta_t \phi_t)\mathbb{E}\|w^t - \widetilde{w}^t\| + \frac{2\eta_t \Phi}{m}$$

where we use the typical convergence rate of GD results in the second inequality, $1 + x \le \exp(x)$ in the third inequality. Additionally, we denote $\Phi = G + \frac{(\lambda + L)G}{C^Q}$. Following the same proof technique in [47](Eq (23) in Theorem 4), we can easily get

$$\mathbb{E}\|w^T - \widetilde{w}^T\| \le \sum_{t=t_0+1}^{T} \exp(2c\gamma \sum_{l=t+1}^{T} \frac{1}{k})\frac{2c\Phi}{mt}$$
$$\le \sum_{t=t_0+}^{T} \exp(2c\gamma \ln(\frac{T}{t}))\frac{2c\Phi}{mt}$$
$$= \frac{2c\Phi}{m}(T^{2c\gamma}) \sum_{t=t_0+}^{T} t^{-2c\gamma-1}$$
$$\le \frac{1}{2c\gamma}\frac{2c\Phi}{m}(\frac{T}{t_0})^{2c\gamma}$$

and

$$\mathbb{E}\|\ell(w_{\mathcal{T}_i}(w^T, \widetilde{\mathcal{S}}_i^{\text{tr}}), \widetilde{z}) - \ell(w_{\mathcal{T}_i}(\widetilde{w}^T, \widetilde{\mathcal{S}}_i^{\text{tr}}), \widetilde{z})\| \le \frac{eG\Phi}{\gamma m}(\frac{T}{t_0})^{2c\gamma} + r\frac{t_0}{m}, \quad (56)$$

where $r = \mathbb{E}_{\mathcal{S},\mathcal{A}}[F(w_{\mathcal{S}})], \gamma = \mathcal{O}(\min\{L_Q, \mathbb{E}_S[\|\nabla^2 F_i(w^0, S\|)] + \rho(c\sigma + \sqrt{c(F(w^0) - \min_{\mathcal{W}} F))}\})$. Next, let $b = 2c\gamma$. Then, setting

$$t_0 = (\frac{2cG\Phi}{r})^{\frac{1}{1+b}} T^{\frac{b}{1+b}}$$

minimizes (56). Plugging $t_0$ back we get that (44) equals to

$$\frac{1 + \frac{1}{b}}{m}(2cG\Phi)^{\frac{1}{1+b}}(rT)^{\frac{b}{1+b}}$$

This completes the proof.

# D    Results of Table 2

In the subsequent proofs of other algorithms, we provide only the proof of generalization bound under the assumptions of the $L_Q$-smoothness constant and $\rho_Q$-Hessian Lipschitz continuity, which can be established by referring our previous proof. We first present their corrosponding algorithm.

## D.1    Algorithms

---

**Algorithm 2** MAML

---

**Require:** The set of datasets $\mathcal{S} = \{S_i\}_{i=1}^m$ with $S_i = \{S_i^{\text{tr}}, S_i^{\text{ts}}\}$, outer iterations $T$, adaptation steps $Q$.

**Require:** Choose arbitrary initial point $w^0 \in W$;

1: **for** $t = 0$ **to** $T - 1$ **do**
2:     Randomly choose the task $i$.
3:     Inner-Level: $w_{\mathcal{T}_i,0}^t = w_t$
4:     **for** $q = 0, 1, ..., Q - 1$ **do**
5:        $w_{\mathcal{T}_i,q+1}^t = w_{\mathcal{T}_i,q}^t - \alpha \nabla \widehat{\mathcal{L}}(w_{\mathcal{T}_i,q}^t, S_i^{\text{tr}})$;
6:     **end for**
7:     Outer-level: $w_{\mathcal{T}_i} = w_{\mathcal{T}_i,Q}^t$
8:     $w^{t+1} := w^t - \eta_t \nabla_w \widehat{\mathcal{L}}_i(w_{\mathcal{T}_i}, S_i^{\text{ts}})$
9: **end for**
10: $w^T$ and $\overline{w}^T := \frac{1}{T+1} \sum_{t=0}^T w^t$;

---

---

**Algorithm 3** FOMAML

---

**Require:** The set of datasets $\mathcal{S} = \{S_i\}_{i=1}^m$ with $S_i = \{S_i^{\text{tr}}, S_i^{\text{ts}}\}$, outer iterations $T$, adaptation steps $Q$.

**Require:** Choose arbitrary initial point $w^0 \in W$;

1: **for** $t = 0$ **to** $T - 1$ **do**
2:     Randomly choose the task $i$.
3:     Inner-Level: $w_{\mathcal{T}_i,0}^t = w_t$
4:     **for** $q = 0, 1, ..., Q - 1$ **do**
5:        $w_{\mathcal{T}_i,q+1}^t = w_{\mathcal{T}_i,q}^t - \alpha \nabla \widehat{\mathcal{L}}(w_{\mathcal{T}_i,q}^t, S_i^{\text{tr}})$;
6:     **end for**
7:     Outer-level: $w_{\mathcal{T}_i} = w_{\mathcal{T}_i,Q}^t$
8:     $w^{t+1} := w^t - \eta_t \nabla_{w_{\mathcal{T}_i}} \widehat{\mathcal{L}}_i(w_{\mathcal{T}_i}, S_i^{\text{ts}})$
9: **end for**
10: $w^T$ and $\overline{w}^T := \frac{1}{T+1} \sum_{t=0}^T w^t$;

---

---

**Algorithm 4** MetaSGD

---

**Require:** The set of datasets $\mathcal{S} = \{S_i\}_{i=1}^m$ with $S_i = \{S_i^{\text{tr}}, S_i^{\text{ts}}\}$, outer iterations $T$, adaptation steps $Q$.

**Require:** Choose arbitrary initial point $w^0 \in W$;

1: **for** $t = 0$ **to** $T - 1$ **do**
2:     Randomly choose the task $i$.
3:     Inner-Level: $w_{\mathcal{T}_i,0}^t = w_t$
4:     **for** $q = 0, 1, ..., Q - 1$ **do**
5:        $w_{\mathcal{T}_i,q+1}^t = w_{\mathcal{T}_i,q}^t - \alpha \circ \nabla \widehat{\mathcal{L}}(w_{\mathcal{T}_i,q}^t, S_i^{\text{tr}})$;
6:     **end for**
7:     Outer-level: $w_{\mathcal{T}_i} = w_{\mathcal{T}_i,Q}^t$
8:     $w^{t+1} := w^t - \eta_t \nabla_w \widehat{\mathcal{L}}_i(w_{\mathcal{T}_i}, S_i^{\text{ts}})$
9:     $\alpha^{t+1} := \alpha^t - \eta_t \nabla_\alpha \widehat{\mathcal{L}}_i(w_{\mathcal{T}_i}, S_i^{\text{ts}})$
10: **end for**
11: $w^T$ and $\overline{w}^T := \frac{1}{T+1} \sum_{t=0}^T w^t$;

---

---

**Algorithm 5** iMAML

---

**Require:** The set of datasets $\mathcal{S} = \{S_i\}_{i=1}^m$ with $S_i = \{S_i^{\text{tr}}, S_i^{\text{ts}}\}$, outer iterations $T$, adaptation steps $Q$,regularization constant $\lambda$

**Require:** Choose arbitrary initial point $w^0 \in W$;

1: **for** $t = 0$ **to** $T - 1$ **do**
2:     Randomly choose the task $i$.
3:     Inner-Level: $w_{\mathcal{T}_i,0}^t = w_t$
4:     **for** $q = 0, 1, ..., Q - 1$ **do**
5:         $w_{\mathcal{T}_i,q+1}^t = w_{\mathcal{T}_i,q}^t - \alpha \nabla \widehat{\mathcal{L}}_\lambda(w_{\mathcal{T}_i,q}^t, S_i^{\text{tr}})$;
6:     **end for**
7:     Outer-level: $w_{\mathcal{T}_i} = w_{\mathcal{T}_i,Q}^t$
8:     $w^{t+1} := w^t - \eta_t (I + \frac{1}{\lambda} \nabla_{w_{\mathcal{T}_i}}^2 \widehat{\mathcal{L}}_i(w_{\mathcal{T}_i}, S_i^{\text{tr}}))^{-1} \nabla_{w_{\mathcal{T}_i}} \widehat{\mathcal{L}}_i(w_{\mathcal{T}_i}, S_i^{\text{ts}})$
9: **end for**
10: $w^T$ and $\overline{w}^T := \frac{1}{T+1} \sum_{t=0}^T w^t$;

---

**Algorithm 6** Meta-MinibatchProx

---

**Require:** The set of datasets $\mathcal{S} = \{S_i\}_{i=1}^m$, outter iterations $T$, adpation steps $Q$, regularization constant $\lambda$.

**Require:** Choose arbitrary initial point $w^0 \in W$;

1: **for** $t = 0$ **to** $T - 1$ **do**
2:     Randomly choose the task $i$.
3:     Inner-Level: $w_{\mathcal{T}_i,0}^t = w_t$
4:     **for** $q = 0, 1, ..., Q - 1$ **do**
5:         $w_{\mathcal{T}_i,q+1}^t = w_{\mathcal{T}_i,q}^t - \alpha \nabla \widehat{\mathcal{K}}(w_{\mathcal{T}_i,q}^t, S_i)$;
6:     **end for**
7:     Outer-level: $w_{\mathcal{T}_i} = w_{\mathcal{T}_i,Q}^t$
8:     $w^{t+1} := w^t - \eta_t \lambda (w^t - w_{\mathcal{T}_i})$
9: **end for**
10: $w^T$ and $\overline{w}^T := \frac{1}{T+1} \sum_{t=0}^T w^t$;

---

**Algorithm 7** FoMuML

---

**Require:** The set of datasets $\mathcal{S} = \{S_i\}_{i=1}^m$ with $S_i = \{S_i^{\text{tr}}, S_i^{\text{ts}}\}$, outer iterations $T$, adaptation steps $Q$, regularization constant $\lambda$.

**Require:** Choose arbitrary initial point $w^0 \in W$;

1: **for** $t = 0$ **to** $T - 1$ **do**
2:     Randomly choose the task $i$.
3:     Inner-Level: $w_{\mathcal{T}_i,0}^t = w_t$
4:     **for** $q = 0, 1, ..., Q - 1$ **do**
5:         $w_{\mathcal{T}_i,q+1}^t = w_{\mathcal{T}_i,q}^t - \alpha \nabla \widehat{\mathcal{L}}_\lambda(w_{\mathcal{T}_i,q}^t, S_i^{\text{tr}})$;
6:     **end for**
7:     Outer-level: $w_{\mathcal{T}_i} = w_{\mathcal{T}_i,Q}^t$
8:     $w^{t+1} := w^t - \eta_t \nabla_{w_{\mathcal{T}_i}} \widehat{\mathcal{L}}_i(w_{\mathcal{T}_i}, S_i^{\text{ts}})$
9: **end for**
10: $w^T$ and $\overline{w}^T := \frac{1}{T+1} \sum_{t=0}^T w^t$;

---

### D.2 FOMAML(convex)

This proof is similar to the proof of Theorem 2, we have

$$
\begin{aligned}
\|w^{t+1} - \widetilde{w}^{t+1}\| &= \|\left(w^t - \eta_t \nabla \widehat{\mathcal{L}}\left(w_{\mathcal{T}_j}(w^t, S_j^{\text{tr}}), S_j^{\text{ts}}\right)\right) - \left(\widetilde{w}^t - \eta_t \nabla \widehat{\mathcal{L}}\left(w_{\mathcal{T}_j}(\widetilde{w}^t, S_j^{\text{tr}}), S_j^{\text{ts}}\right)\right)\| \\
&\leq \frac{1}{n^{\text{ts}}} \sum_{z^{\text{ts}} \in S_j^{\text{ts}}} \|\left(w^t - \eta_t \nabla \ell\left(w_{\mathcal{T}_j}(w^t, S_j^{\text{tr}}), z^{\text{ts}}\right)\right) - \left(\widetilde{w}^t - \eta_t \nabla \ell\left(w_{\mathcal{T}_j}(\widetilde{w}^t, S_j^{\text{tr}}), z^{\text{ts}}\right)\right)\| \\
&\leq \|w^t - \widetilde{w}^t\|
\end{aligned}
$$

$$(57)$$

Next, for a given time index $t$, with probability $\frac{1}{m}$, the task $\mathcal{T}_i$ is selected. In this case, we have

$$
\begin{aligned}
\mathbb{E}\|w^{t+1} - \widetilde{w}^{t+1}\| =& \mathbb{E}\|\big(w^t - \eta_t \nabla\widehat{\mathcal{L}}(w_{\mathcal{T}_i}(w^t, S_i^{\text{tr}}), S_i^{\text{ts}})\big) - \big(\widetilde{w}^t - \eta_t \nabla\widehat{\mathcal{L}}(w_{\mathcal{T}_i}(\widetilde{w}^t, \widetilde{S}_i^{\text{tr}}), \widetilde{S}_i^{\text{ts}})\big)\| \\
\leq& \mathbb{E}\|\big(w^t - \eta_t \nabla\widehat{\mathcal{L}}(w_{\mathcal{T}_i}(w^t, S_i^{\text{tr}}), S_i^{\text{ts}})\big) - \big(\widetilde{w}^t - \eta_t \nabla\widehat{\mathcal{L}}(w_{\mathcal{T}_i}(\widetilde{w}^t, \widetilde{S}_i^{\text{tr}}), S_i^{\text{ts}})\big)\| \\
& + \mathbb{E}\|\eta_t \nabla\widehat{\mathcal{L}}(w_{\mathcal{T}_i}(\widetilde{w}^t, \widetilde{S}_i^{\text{tr}}), \widetilde{S}_i^{\text{ts}}) - \eta_t \nabla\widehat{\mathcal{L}}(w_{\mathcal{T}_i}(\widetilde{w}^t, \widetilde{S}_i^{\text{tr}}), S_i^{\text{ts}})\| \\
\leq& \frac{1}{n^{\text{ts}}} \sum_{z^{\text{ts}} \in S_i^{\text{ts}}} \mathbb{E}\|\big(w^t - \eta_t \nabla\widehat{\mathcal{L}}(w_{\mathcal{T}_i}(w^t, S_i^{\text{tr}}), z^{\text{ts}})\big) - \big(\widetilde{w}^t - \eta_t \nabla\widehat{\mathcal{L}}(w_{\mathcal{T}_i}(\widetilde{w}^t, \widetilde{S}_i^{\text{tr}}), z^{\text{ts}})\big)\| \\
& + \eta_t \mathbb{E}\|\nabla\widehat{\mathcal{L}}(w_{\mathcal{T}_i}(\widetilde{w}^t, \widetilde{S}_i^{\text{tr}}), \widetilde{S}_i^{\text{ts}}) - \nabla\widehat{\mathcal{L}}(w_{\mathcal{T}_i}(\widetilde{w}^t, \widetilde{S}_i^{\text{tr}}), S_i^{\text{ts}})\| \\
\leq& \frac{1}{n^{\text{ts}}} \sum_{z \in S_i^{\text{ts}}} \mathbb{E}\|\big(w^t - \eta_t \nabla\ell(w_{\mathcal{T}_i}(w^t, S_i^{\text{tr}}), z^{\text{ts}})\big) - \big(\widetilde{w}^t - \eta_t \nabla\ell(w_{\mathcal{T}_i}(\widetilde{w}^t, \widetilde{S}_i^{\text{tr}}), z^{\text{ts}})\big)\| \\
& + 2\eta_t \mathbb{E}\|\nabla\widehat{F}_i(w)\|,
\end{aligned}
\tag{58}
$$

where the last inequality follows that $\widetilde{S}_i^{\text{ts}}$ and $S_i^{\text{ts}}$ are sampled from the same distribution, then $\mathbb{E}\|\nabla\widehat{\mathcal{L}}(w_{\mathcal{T}_i}(\widetilde{w}^t, \widetilde{S}_i^{\text{tr}}), \widetilde{S}_i^{\text{ts}})\| = \mathbb{E}\|\nabla\widehat{\mathcal{L}}(w_{\mathcal{T}_i}(\widetilde{w}^t, \widetilde{S}_i^{\text{tr}}), S_i^{\text{ts}}) = \mathbb{E}\|\nabla\widehat{F}_i(w)\|$. Note that

$$
\begin{aligned}
& \mathbb{E}\|\big(w^t - \eta_t \nabla\ell(w_{\mathcal{T}_i}(w^t, S_i^{\text{tr}}), z^{\text{ts}})\big) - \big(\widetilde{w}^t - \eta_t \nabla\ell(w_{\mathcal{T}_i}(\widetilde{w}^t, \widetilde{S}_i^{\text{tr}}), z^{\text{ts}})\big)\| \\
\leq& \mathbb{E}\|\big(w^t - \eta_t \nabla\ell(w_{\mathcal{T}_i}(w^t, S_i^{\text{tr}}), z^{\text{ts}})\big) - \big(\widetilde{w}^t - \eta_t \nabla\ell(w_{\mathcal{T}_i}(\widetilde{w}^t, S_i^{\text{tr}}), z^{\text{ts}})\big)\| \\
& + \eta_t \mathbb{E}\|\nabla\ell(w_{\mathcal{T}_i}(\widetilde{w}^t, S_i^{\text{tr}}), z^{\text{ts}}) - \nabla\ell(w_{\mathcal{T}_i}(\widetilde{w}^t, \widetilde{S}_i^{\text{tr}}), z^{\text{ts}})\|.
\end{aligned}
\tag{59}
$$

Let us bound the two terms on the RHS of (59) separately. First, similar to how we derived (57), we could bound the first term by

$$
\mathbb{E}\|\big(w^t - \eta_t \nabla\ell(w_{\mathcal{T}_i}(w^t, S_i^{\text{tr}}), z^{\text{ts}})\big) - \big(\widetilde{w}^t - \eta_t \nabla\ell(w_{\mathcal{T}_i}(\widetilde{w}^t, S_i^{\text{tr}}), z^{\text{ts}})\big)\| \leq \mathbb{E}\|w^t - \widetilde{w}^t\|.
$$

To bound the second term on the RHS of (59), we consider two parallel processes of generating iterates $\{\widetilde{w}_{\mathcal{T}_i, q}^t\}$ and $\{\widetilde{w}_{\mathcal{T}_i, q}^{t,\prime}\}$ by using datasets $S_i^{\text{tr}}$ and $\widetilde{S}_i^{\text{tr}}$, respectively. Note that

$$
\begin{aligned}
& \mathbb{E}\|\nabla\ell(w_{\mathcal{T}_i}(\widetilde{w}^t, S_i^{\text{tr}}), z^{\text{ts}}) - \nabla\ell(w_{\mathcal{T}_i}(\widetilde{w}^t, \widetilde{S}_i^{\text{tr}}), z^{\text{ts}})\| \\
=& \mathbb{E}\|\nabla\ell(\widetilde{w}_{\mathcal{T}_i, Q}^t, z^{\text{ts}}) - \nabla\ell(\widetilde{w}_{i, Q}^{t,\prime}, z^{\text{ts}})\| \\
\leq& L\mathbb{E}\|\widetilde{w}_{\mathcal{T}_i, Q}^t - \widetilde{w}_{i, Q}^{t,\prime}\| \\
=& L\mathbb{E}\|\big[\widetilde{w}_{i, Q-1}^t - \alpha\nabla\widehat{\mathcal{L}}(\widetilde{w}_{i, Q-1}^t, S_i^{\text{tr}})\big] - \big[\widetilde{w}_{i, Q-1}^{t,\prime} - \alpha\nabla\widehat{\mathcal{L}}(\widetilde{w}_{i, Q-1}^{t,\prime}, \widetilde{S}_i^{\text{tr}})\big]\| \\
\leq& L\mathbb{E}\|\big[\widetilde{w}_{i, Q-1}^t - \alpha\nabla\widehat{\mathcal{L}}(\widetilde{w}_{i, Q-1}^t, S_i^{\text{tr}}))\big] - \big[\widetilde{w}_{i, Q-1}^{t,\prime} - \alpha\nabla\ell(\widetilde{w}_{i, Q-1}^{t,\prime}, S_i^{\text{tr}})\big]\| \\
& + L\mathbb{E}\|\alpha\nabla\widehat{\mathcal{L}}(\widetilde{w}_{i, Q-1}^{t,\prime}, S_i^{\text{tr}}) - \alpha\nabla\widehat{\mathcal{L}}(\widetilde{w}_{i, Q-1}^{t,\prime}, \widetilde{S}_i^{\text{tr}})\| \\
\leq& L\mathbb{E}\|\widetilde{w}_{i, Q-1}^t - \widetilde{w}_{i, Q-1}^{t,\prime}\| + \frac{2\alpha LG}{n^{\text{tr}}} \\
\leq& L\mathbb{E}\|\big[\widetilde{w}^t - \alpha\nabla\widehat{\mathcal{L}}(\widetilde{w}^t, S_i^{\text{tr}})\big] - \big[\widetilde{w}^t - \alpha\nabla\widehat{\mathcal{L}}(\widetilde{w}^t, \widetilde{S}_i^{\text{tr}})\big]\| + \frac{2(Q-1)\alpha LG}{n^{\text{tr}}} \\
=& \alpha L\mathbb{E}\|\nabla\widehat{\mathcal{L}}(\widetilde{w}^t, S_i^{\text{tr}}) - \nabla\widehat{\mathcal{L}}(\widetilde{w}^t, \widetilde{S}_i^{\text{tr}})\| + \frac{2(Q-1)\alpha LG}{n^{\text{tr}}} \\
\leq& \frac{2Q\alpha LG}{n^{\text{tr}}},
\end{aligned}
\tag{60}
$$

Substituting (60) and (59) into (58), we have

$$
\mathbb{E}\|w^{t+1} - \widetilde{w}^{t+1}\| \leq \mathbb{E}\|w^t - \widetilde{w}^t\| + \eta_t \frac{2Q\alpha LG}{n^{\text{tr}}} + 2\eta_t \mathbb{E}\|\nabla\widehat{F}_i(w^t)\|.
$$

Combing the above two cases, we obtain

$$
\begin{aligned}
\mathbb{E}\|w^{t+1} - \widetilde{w}^{t+1}\| \leq& (1 - \frac{1}{m})\mathbb{E}\|w^t - \widetilde{w}^t\| + \frac{1}{m}\mathbb{E}\|w^t - \widetilde{w}^t\| + \frac{1}{m}\eta_t \frac{2Q\alpha LG}{n^{\text{tr}}} + \frac{2}{m}\eta_t \mathbb{E}\|\nabla\widehat{F}_i(w^t)\| \\
=& \mathbb{E}\|w^t - \widetilde{w}^t\| + \eta_t \frac{2Q\alpha LG}{mn^{\text{tr}}} + \frac{2\eta_t}{m}\mathbb{E}\|\nabla\widehat{F}_i(w^t)\|.
\end{aligned}
$$

Unrolling it and noting that $\|w^0 - \widetilde{w}^0\| = 0$, we have

$$
\begin{aligned}
\mathbb{E}[\|w^T - \widetilde{w}^T\|] &\leq \sum_{t=0}^{T-1} \eta_t \frac{2Q\alpha LG}{mn^{\mathrm{tr}}} + \sum_{t=0}^{T-1} \frac{2\eta_t}{m} \mathbb{E}\|\nabla \widehat{F}(w^t, S_i)\| \\
&\leq \sum_{t=0}^{T-1} \eta_t \frac{2Q\alpha LG}{mn^{\mathrm{tr}}} + \frac{1}{m} \sqrt{F(w^0) - \min_{\mathcal{W}} F + \frac{L_Q \sigma^2}{2} \sum_{t=0}^{T-1} \eta_t^2},
\end{aligned}
$$

where we use Lemma 13 in the last inequality. which completes the proof.

### D.3   FOMAML(non-convex)

This proof is similar to the proof of Theorem 3, we have

$$
\begin{aligned}
\|w^{t+1} - \widetilde{w}^{t+1}\| = &\|\big(w^t - \eta_t \nabla\widehat{\mathcal{L}}\big(w_{\mathcal{T}_j}(w^t, S_j^{\mathrm{tr}}), S_j^{\mathrm{ts}}\big)\big) - \big(\widetilde{w}^t - \eta_t \nabla\widehat{\mathcal{L}}\big(w_{\mathcal{T}_j}(\widetilde{w}^t, S_j^{\mathrm{tr}}), S_j^{\mathrm{ts}}\big)\big)\| \\
&\leq (1 + \eta_t \phi_t)\|w^t - \widetilde{w}^t\|,
\end{aligned} \tag{61}
$$

where $\phi_t = \min\{L_Q, \xi_t\}$ with $\xi_t = \|\nabla^2 F(w_0, S_t)\| + \frac{\rho_Q}{2} \left\|\sum_{l=1}^{t-1} \beta_l \nabla\widehat{F}(w_{\mathcal{S}}^l)\right\| + \frac{\rho_Q}{2} \left\|\sum_{l=1}^{t-1} \beta_l \nabla\widehat{F}(w_{\widetilde{\mathcal{S}}}^l)\right\|$.

Next, for the second case, similar to the proof of (58), we have

$$
\begin{aligned}
\mathbb{E}\|w^{t+1} - \widetilde{w}^{t+1}\| \leq &\frac{1}{n^{\mathrm{ts}}} \sum_{z \in S_i^{\mathrm{ts}}} \mathbb{E}\|\big(w^t - \eta_t \nabla\ell\big(w_{\mathcal{T}_i}(w^t, S_i^{\mathrm{tr}}), z^{\mathrm{ts}}\big)\big) - \big(\widetilde{w}^t - \eta_t \nabla\ell\big(w_{\mathcal{T}_i}(\widetilde{w}^t, \widetilde{S}_i^{\mathrm{tr}}), z^{\mathrm{ts}}\big)\big)\| \\
&+ 2\eta_t \mathbb{E}\|\nabla\widehat{F}_i(w)\|.
\end{aligned} \tag{62}
$$

Note that

$$
\begin{aligned}
&\mathbb{E}\|\big(w^t - \eta_t \nabla\ell\big(w_{\mathcal{T}_i}(w^t, S_i^{\mathrm{tr}}), z^{\mathrm{ts}}\big)\big) - \big(\widetilde{w}^t - \eta_t \nabla\ell\big(w_{\mathcal{T}_i}(\widetilde{w}^t, \widetilde{S}_i^{\mathrm{tr}}), z^{\mathrm{ts}}\big)\big)\| \\
&\leq \mathbb{E}\|\big(w^t - \eta_t \nabla\ell\big(w_{\mathcal{T}_i}(w^t, S_i^{\mathrm{tr}}), z^{\mathrm{ts}}\big)\big) - \big(\widetilde{w}^t - \eta_t \nabla\ell\big(w_{\mathcal{T}_i}(\widetilde{w}^t, S_i^{\mathrm{tr}}), z^{\mathrm{ts}}\big)\big)\| \\
&+ \eta_t \mathbb{E}\|\nabla\ell\big(w_{\mathcal{T}_i}(\widetilde{w}^t, S_i^{\mathrm{tr}}), z^{\mathrm{ts}}\big) - \nabla\ell\big(w_{\mathcal{T}_i}(\widetilde{w}^t, \widetilde{S}_i^{\mathrm{tr}}), z^{\mathrm{ts}}\big)\|.
\end{aligned} \tag{63}
$$

Let us bound the two terms on the RHS of (63), separately. First, similar to how we bound (61), we could bound the first term by

$$
\mathbb{E}\|\big(w^t - \eta_t \nabla\ell\big(w_{\mathcal{T}_i}(w^t, S_i^{\mathrm{tr}}), z^{\mathrm{ts}}\big)\big) - \big(\widetilde{w}^t - \eta_t \nabla\ell\big(w_{\mathcal{T}_i}(\widetilde{w}^t, S_i^{\mathrm{tr}}), z^{\mathrm{ts}}\big)\big)\| \leq (1 + \eta_t \phi_t)\mathbb{E}\|w^t - \widetilde{w}^t\|.
$$

To bound the second term on the RHS of (63), similar to the proof of (60), we have

$$
\mathbb{E}\|\nabla\ell\big(w_{\mathcal{T}_i}(\widetilde{w}^t, S_i^{\mathrm{tr}}), z^{\mathrm{ts}}\big) - \nabla\ell\big(w_{\mathcal{T}_i}(\widetilde{w}^t, \widetilde{S}_i^{\mathrm{tr}}), z^{\mathrm{ts}}\big)\| \leq \frac{2Q(1+\alpha L)^Q \alpha LG}{n^{\mathrm{tr}}} \tag{64}
$$

Substituting (64) and (63) into (62), we obtain

$$
\mathbb{E}\|w^{t+1} - \widetilde{w}^{t+1}\| \leq (1 + \eta_t \phi_t)\mathbb{E}\|w^t - \widetilde{w}^t\| + \eta_t \frac{2Q(1+\alpha L)^Q \alpha LG}{n^{\mathrm{tr}}} + 2\eta_t \mathbb{E}\|\nabla\widehat{F}_i(w^t)\|.
$$

From Lemma 7, we can know $\mathbb{E}\|\nabla\widehat{F}_i(w^t)\| \leq eG$. Then combing the above two cases, we obtain

$$
\begin{aligned}
\mathbb{E}\|w^{t+1} - \widetilde{w}^{t+1}\| \leq &(1 - \frac{1}{m})(1 + \eta_t \phi_t)\mathbb{E}\|w^t - \widetilde{w}^t\| + \frac{1}{m}(1 + \eta_t \phi_t)\mathbb{E}\|w^t - \widetilde{w}^t\| \\
&+ \frac{2Q(1+\alpha L)^Q \alpha LG}{mn^{\mathrm{tr}}} + \frac{2\eta_t eG}{m} \\
\leq &\exp(\eta_t \phi_t)\mathbb{E}\|w^t - \widetilde{w}^t\| + \frac{\eta_t \Phi}{m}.
\end{aligned}
$$

where we use $1 + x \leq \exp(x)$ and $\alpha \leq \frac{1}{QL}$ in the second inequality, and we denote $\Phi = 2eG + \frac{2Q(1+\alpha L)^Q \alpha LG}{n^{\mathrm{tr}}}$. Using the same technique in Theorem 3. Then we complete the proof.

## D.4 iMAML(convex)

Since our goal is to demonstrate the extensibility of our framework analysis, here we consider the exact version of the solved iMAML algorithm, and of course we believe that this can be equally generalized to the iMAML algorithm with an error term. Let $w_{\mathcal{T}_i}^{t,*} = \operatorname{argmin}_{w_{\mathcal{T}_i}} \{\widehat{\mathcal{L}}(w_{\mathcal{T}_i}, S_j^{\mathrm{tr}}) + \frac{\lambda}{2}\|w_{\mathcal{T}_i} - w^t\|^2\}$, we have

$$
\begin{aligned}
\|w^{t+1} - \widetilde{w}^{t+1}\| =& \|(w^t - \eta_t\nabla\widehat{\mathcal{L}}(w_{\mathcal{T}_j}^*(w^t, S_j^{\mathrm{tr}}), S_j^{\mathrm{ts}})) - (\widetilde{w}^t - \eta_t\nabla\widehat{\mathcal{L}}(w_{\mathcal{T}_j}^*(\widetilde{w}^t, S_j^{\mathrm{tr}}), S_j^{\mathrm{ts}}))\| \\
\leq& \frac{1}{n^{\mathrm{ts}}}\sum_{z^{\mathrm{ts}}\in S_j^{\mathrm{ts}}}\|(w^t - \eta_t\nabla\ell(w_{\mathcal{T}_j}^*(w^t, S_j^{\mathrm{tr}}), z^{\mathrm{ts}})) - (\widetilde{w}^t - \eta_t\nabla\ell(w_{\mathcal{T}_j}^*(\widetilde{w}^t, S_j^{\mathrm{tr}}), z^{\mathrm{ts}}))\| \\
\leq& \|w^t - \widetilde{w}^t\|
\end{aligned}
$$
(65)

Next, for a given time index $t$, with probability $\frac{1}{m}$, the task $\mathcal{T}_i$ is selected. In this case, we have

$$
\begin{aligned}
\mathbb{E}\|w^{t+1} - \widetilde{w}^{t+1}\| =& \mathbb{E}\|(w^t - \eta_t\nabla\widehat{\mathcal{L}}(w_{\mathcal{T}_i}^*(w^t, S_i^{\mathrm{tr}}), S_i^{\mathrm{ts}})) - (\widetilde{w}^t - \eta_t\nabla\widehat{\mathcal{L}}(w_{\mathcal{T}_i}^*(\widetilde{w}^t, \widetilde{S}_i^{\mathrm{tr}}), \widetilde{S}_i^{\mathrm{ts}}))\| \\
\leq& \mathbb{E}\|(w^t - \eta_t\nabla\widehat{\mathcal{L}}(w_{\mathcal{T}_i}^*(w^t, S_i^{\mathrm{tr}}), S_i^{\mathrm{ts}})) - (\widetilde{w}^t - \eta_t\nabla\widehat{\mathcal{L}}(w_{\mathcal{T}_i}^*(\widetilde{w}^t, \widetilde{S}_i^{\mathrm{tr}}), S_i^{\mathrm{ts}}))\| \\
& + \mathbb{E}\|\eta_t\nabla\widehat{\mathcal{L}}(w_{\mathcal{T}_i}^*(\widetilde{w}^t, \widetilde{S}_i^{\mathrm{tr}}), \widetilde{S}_i^{\mathrm{ts}}) - \eta_t\nabla\widehat{\mathcal{L}}(w_{\mathcal{T}_i}^*(\widetilde{w}^t, \widetilde{S}_i^{\mathrm{tr}}), S_i^{\mathrm{ts}})\| \\
\leq& \frac{1}{n^{\mathrm{ts}}}\sum_{z^{\mathrm{ts}}\in S_i^{\mathrm{ts}}}\mathbb{E}\|(w^t - \eta_t\nabla\widehat{\mathcal{L}}(w_{\mathcal{T}_i}^*(w^t, S_i^{\mathrm{tr}}), z^{\mathrm{ts}})) - (\widetilde{w}^t - \eta_t\nabla\widehat{\mathcal{L}}(w_{\mathcal{T}_i}^*(\widetilde{w}^t, \widetilde{S}_i^{\mathrm{tr}}), z^{\mathrm{ts}}))\| \\
& + \eta_t\mathbb{E}\|\nabla\widehat{\mathcal{L}}(w_{\mathcal{T}_i}^*(\widetilde{w}^t, \widetilde{S}_i^{\mathrm{tr}}), \widetilde{S}_i^{\mathrm{ts}}) - \nabla\widehat{\mathcal{L}}(w_{\mathcal{T}_i}^*(\widetilde{w}^t, \widetilde{S}_i^{\mathrm{tr}}), S_i^{\mathrm{ts}})\| \\
\leq& \frac{1}{n^{\mathrm{ts}}}\sum_{z\in S_i^{\mathrm{ts}}}\mathbb{E}\|(w^t - \eta_t\nabla\ell(w_{\mathcal{T}_i}^*(w^t, S_i^{\mathrm{tr}}), z^{\mathrm{ts}})) - (\widetilde{w}^t - \eta_t\nabla\ell(w_{\mathcal{T}_i}^*(\widetilde{w}^t, \widetilde{S}_i^{\mathrm{tr}}), z^{\mathrm{ts}}))\| \\
& + 2\eta_t\mathbb{E}\|\nabla\widehat{F}_i(w)\|,
\end{aligned}
$$
(66)

where the last inequality follows that $\widetilde{S}_i^{\mathrm{ts}}$ and $S_i^{\mathrm{ts}}$ are sampled from the same distribution, then $\mathbb{E}\|\nabla\widehat{\mathcal{L}}(w_{\mathcal{T}_i}^*(\widetilde{w}^t, \widetilde{S}_i^{\mathrm{tr}}), \widetilde{S}_i^{\mathrm{ts}})\| = \mathbb{E}\|\nabla\widehat{\mathcal{L}}(w_{\mathcal{T}_i}^*(\widetilde{w}^t, \widetilde{S}_i^{\mathrm{tr}}), S_i^{\mathrm{ts}}) = \mathbb{E}\|\nabla\widehat{F}_i(w)\|$. Note that

$$
\begin{aligned}
& \mathbb{E}\|(w^t - \eta_t\nabla\ell(w_{\mathcal{T}_i}^*(w^t, S_i^{\mathrm{tr}}), z^{\mathrm{ts}})) - (\widetilde{w}^t - \eta_t\nabla\ell(w_{\mathcal{T}_i}^*(\widetilde{w}^t, \widetilde{S}_i^{\mathrm{tr}}), z^{\mathrm{ts}}))\| \\
\leq& \mathbb{E}\|(w^t - \eta_t\nabla\ell(w_{\mathcal{T}_i}^*(w^t, S_i^{\mathrm{tr}}), z^{\mathrm{ts}})) - (\widetilde{w}^t - \eta_t\nabla\ell(w_{\mathcal{T}_i}^*(\widetilde{w}^t, S_i^{\mathrm{tr}}), z^{\mathrm{ts}}))\| \\
& + \eta_t\mathbb{E}\|\nabla\ell(w_{\mathcal{T}_i}^*(\widetilde{w}^t, S_i^{\mathrm{tr}}), z^{\mathrm{ts}}) - \nabla\ell(w_{\mathcal{T}_i}^*(\widetilde{w}^t, \widetilde{S}_i^{\mathrm{tr}}), z^{\mathrm{ts}})\|.
\end{aligned}
$$
(67)

Let us bound the two terms on the RHS of (67) separately. First, similar to how we derived (65), we could bound the first term by

$$
\mathbb{E}\|(w^t - \eta_t\nabla\ell(w_{\mathcal{T}_i}^*(w^t, S_i^{\mathrm{tr}}), z^{\mathrm{ts}})) - (\widetilde{w}^t - \eta_t\nabla\ell(w_{\mathcal{T}_i}^*(\widetilde{w}^t, S_i^{\mathrm{tr}}), z^{\mathrm{ts}}))\| \leq \mathbb{E}\|w^t - \widetilde{w}^t\|.
$$

To bound the second term on the RHS of (67), note that

$$
\begin{aligned}
& \mathbb{E}\|\nabla\ell(w_{\mathcal{T}_i}^*(\widetilde{w}^t, S_i^{\mathrm{tr}}), z^{\mathrm{ts}}) - \nabla\ell(w_{\mathcal{T}_i}^*(\widetilde{w}^t, \widetilde{S}_i^{\mathrm{tr}}), z^{\mathrm{ts}})\| \\
=& \mathbb{E}\|(I + \frac{1}{\lambda}\nabla^2\widehat{\mathcal{L}}(w_{\mathcal{T}_i}^*, S_i^{\mathrm{tr}}))^{-1}\nabla\ell(w_{\mathcal{T}_i}^*, z^{\mathrm{ts}}) - (I + \frac{1}{\lambda}\nabla^2\widehat{\mathcal{L}}(w_{\mathcal{T}_i}^*, \widetilde{S}_i^{\mathrm{tr}}))^{-1}\nabla\ell(\widetilde{w}_{\mathcal{T}_i}^*, z^{\mathrm{ts}})\| \\
\leq& \mathbb{E}\|(I + \frac{1}{\lambda}\nabla^2\widehat{\mathcal{L}}(w_{\mathcal{T}_i}^*, S_i^{\mathrm{tr}}))^{-1}\nabla\ell(w_{\mathcal{T}_i}^*, z^{\mathrm{ts}}) - (I + \frac{1}{\lambda}\nabla^2\widehat{\mathcal{L}}(w_{\mathcal{T}_i}^*, S_i^{\mathrm{tr}}))^{-1}\nabla\ell(\widetilde{w}_{\mathcal{T}_i}^*, z^{\mathrm{ts}})\| \\
& + \mathbb{E}\|(I + \frac{1}{\lambda}\nabla^2\widehat{\mathcal{L}}(w_{\mathcal{T}_i}^*, S_i^{\mathrm{tr}}))^{-1}\nabla\ell(\widetilde{w}_{\mathcal{T}_i}^*, z^{\mathrm{ts}}) - (I + \frac{1}{\lambda}\nabla^2\widehat{\mathcal{L}}(w_{\mathcal{T}_i}^*, \widetilde{S}_i^{\mathrm{tr}}))^{-1}\nabla\ell(\widetilde{w}_{\mathcal{T}_i}^*, z^{\mathrm{ts}})\| \\
\leq& \mathbb{E}\|(I + \frac{1}{\lambda}\nabla^2\widehat{\mathcal{L}}(w_{\mathcal{T}_i}^*, S_i^{\mathrm{tr}}))^{-1}\|\|\nabla\ell(w_{\mathcal{T}_i}^*, z^{\mathrm{ts}}) - \nabla\ell(\widetilde{w}_{\mathcal{T}_i}^*, z^{\mathrm{ts}})\| \\
& + \mathbb{E}\|(I + \frac{1}{\lambda}\nabla^2\widehat{\mathcal{L}}(w_{\mathcal{T}_i}^*, S_i^{\mathrm{tr}}))^{-1} - (I + \frac{1}{\lambda}\nabla^2\widehat{\mathcal{L}}(w_{\mathcal{T}_i}^*, \widetilde{S}_i^{\mathrm{tr}}))^{-1}\|\|\nabla\ell(\widetilde{w}_{\mathcal{T}_i}^*, z^{\mathrm{ts}})\|
\end{aligned}
$$
(68)

For the first term in (68), since the inner-optimization problem is in $\lambda$ strongly-convex setting, then by using the standard stability result in [43], we have

$$
\mathbb{E}\|(I + \frac{1}{\lambda}\nabla^2\widehat{\mathcal{L}}(w_{\mathcal{T}_i}^*, S_i^{\mathrm{tr}}))^{-1}\|\|\nabla\ell(w_{\mathcal{T}_i}^*, z^{\mathrm{ts}}) - \nabla\ell(\widetilde{w}_{\mathcal{T}_i}^*, z^{\mathrm{ts}})\| \leq \frac{2LG^2}{\lambda n^{\mathrm{tr}}}
$$
(69)

For the second term in (68), since $S_i^{\mathrm{tr}}$ different with $\widetilde{S}_i^{\mathrm{tr}}$ at most one point, then we have

$$\mathbb{E}\|\big(I + \tfrac{1}{\lambda}\nabla^2\widehat{\mathcal{L}}(w_{\mathcal{T}_i}^*, S_i^{\mathrm{tr}})\big)^{-1} - \big(I + \tfrac{1}{\lambda}\nabla^2\widehat{\mathcal{L}}(w_{\mathcal{T}_i}^*, \widetilde{S}_i^{\mathrm{tr}})\big)^{-1}\|\|\nabla\ell(\widetilde{w}_{\mathcal{T}_i}^*, z^{\mathrm{ts}})\| \leq \frac{2LG}{\lambda n^{\mathrm{tr}}} \tag{70}$$

By plugging (69) and (70) into (68), then we have

$$\mathbb{E}\|\nabla\ell\big(w_{\mathcal{T}_i}(\widetilde{w}^t, S_i^{\mathrm{tr}}), z^{\mathrm{ts}}\big) - \nabla\ell\big(w_{\mathcal{T}_i}(\widetilde{w}^t, \widetilde{S}_i^{\mathrm{tr}}), z^{\mathrm{ts}}\big)\| \leq \frac{2L(G^2 + G)}{\lambda n^{\mathrm{tr}}} \tag{71}$$

Substituting (71) and (67) into (66), we have

$$\mathbb{E}\|w^{t+1} - \widetilde{w}^{t+1}\| \leq \mathbb{E}\|w^t - \widetilde{w}^t\| + \eta_t\frac{2L(G^2 + G)}{\lambda n^{\mathrm{tr}}} + 2\eta_t\mathbb{E}\|\nabla\widehat{F}_i(w^t)\|.$$

Combing the above two cases, we obtain

$$\begin{aligned}
\mathbb{E}\|w^{t+1} - \widetilde{w}^{t+1}\| \leq &(1 - \tfrac{1}{m})\mathbb{E}\|w^t - \widetilde{w}^t\| + \tfrac{1}{m}\mathbb{E}\|w^t - \widetilde{w}^t\| \\
&+ \tfrac{1}{m}\frac{2\eta_t L(G^2 + G)}{\lambda n^{\mathrm{tr}}} + \frac{2\eta_t}{m}\mathbb{E}\|\nabla\widehat{F}_i(w^t)\| \\
= &\mathbb{E}\|w^t - \widetilde{w}^t\| + \frac{2\eta_t L(G^2 + G)}{\lambda m n^{\mathrm{tr}}} + \frac{2\eta_t}{m}\mathbb{E}\|\nabla\widehat{F}_i(w^t)\|.
\end{aligned}$$

Unrolling it and noting that $\|w^0 - \widetilde{w}^0\| = 0$, we have

$$\begin{aligned}
\mathbb{E}[\|w^T - \widetilde{w}^T\|] \leq &\sum_{t=0}^{T-1}\frac{2\eta_t L(G^2 + G)}{\lambda m n^{\mathrm{tr}}} + \sum_{t=0}^{T-1}\frac{2\eta_t}{m}\mathbb{E}\|\nabla\widehat{F}(w^t, S_i)\| \\
\leq &\sum_{t=0}^{T-1}\frac{2\eta_t L(G^2 + G)}{\lambda m n^{\mathrm{tr}}} + \frac{1}{m}\sqrt{F(w^0) - \min_{\mathcal{W}} F + \frac{L_Q\sigma^2}{2}\sum_{t=0}^{T-1}\eta_t^2},
\end{aligned}$$

where we use Lemma 13 in the last inequality. which completes the proof.

### D.5  iMAML(non-convex)

This proof is similar to the proof of Theorem 3, we have

$$\begin{aligned}
\|w^{t+1} - \widetilde{w}^{t+1}\| = &\|\big(w^t - \eta_t\nabla\widehat{\mathcal{L}}\big(w_{\mathcal{T}_j}(w^t, S_j^{\mathrm{tr}}), S_j^{\mathrm{ts}}\big)\big) - \big(\widetilde{w}^t - \eta_t\nabla\widehat{\mathcal{L}}\big(w_{\mathcal{T}_j}(\widetilde{w}^t, S_j^{\mathrm{tr}}), S_j^{\mathrm{ts}}\big)\big)\| \\
\leq &(1 + \eta_t\phi_t)\|w^t - \widetilde{w}^t\|,
\end{aligned} \tag{72}$$

where $\phi_t = \min\{L_Q, \xi_t\}$ with $\xi_t = \|\nabla^2 F(w_0, S_t)\| + \frac{\rho_Q}{2}\left\|\sum_{l=1}^{t-1}\beta_l\nabla\widehat{F}(w_{\mathcal{S}}^l)\right\| + \frac{\rho_Q}{2}\left\|\sum_{l=1}^{t-1}\beta_l\nabla\widehat{F}(w_{\widetilde{\mathcal{S}}}^l)\right\|$.

Next, for the second case, similar to the proof of (66), we have

$$\begin{aligned}
\mathbb{E}\|w^{t+1} - \widetilde{w}^{t+1}\| \leq &\frac{1}{n^{\mathrm{ts}}}\sum_{z \in S_i^{\mathrm{ts}}}\mathbb{E}\|\big(w^t - \eta_t\nabla\ell\big(w_{\mathcal{T}_i}(w^t, S_i^{\mathrm{tr}}), z^{\mathrm{ts}}\big)\big) \\
&- \big(\widetilde{w}^t - \eta_t\nabla\ell\big(w_{\mathcal{T}_i}(\widetilde{w}^t, \widetilde{S}_i^{\mathrm{tr}}), z^{\mathrm{ts}}\big)\big)\| + 2\eta_t\mathbb{E}\|\nabla\widehat{F}_i(w)\|.
\end{aligned} \tag{73}$$

Note that

$$\begin{aligned}
&\mathbb{E}\|\big(w^t - \eta_t\nabla\ell\big(w_{\mathcal{T}_i}(w^t, S_i^{\mathrm{tr}}), z^{\mathrm{ts}}\big)\big) - \big(\widetilde{w}^t - \eta_t\nabla\ell\big(w_{\mathcal{T}_i}(\widetilde{w}^t, \widetilde{S}_i^{\mathrm{tr}}), z^{\mathrm{ts}}\big)\big)\| \\
\leq &\mathbb{E}\|\big(w^t - \eta_t\nabla\ell\big(w_{\mathcal{T}_i}(w^t, S_i^{\mathrm{tr}}), z^{\mathrm{ts}}\big)\big) - \big(\widetilde{w}^t - \eta_t\nabla\ell\big(w_{\mathcal{T}_i}(\widetilde{w}^t, S_i^{\mathrm{tr}}), z^{\mathrm{ts}}\big)\big)\| \\
&+ \eta_t\mathbb{E}\|\nabla\ell\big(w_{\mathcal{T}_i}(\widetilde{w}^t, S_i^{\mathrm{tr}}), z^{\mathrm{ts}}\big) - \nabla\ell\big(w_{\mathcal{T}_i}(\widetilde{w}^t, \widetilde{S}_i^{\mathrm{tr}}), z^{\mathrm{ts}}\big)\|.
\end{aligned} \tag{74}$$

Let us bound the two terms on the RHS of (74), separately. First, similar to how we bound (72), we could bound the first term by

$$\mathbb{E}\|\big(w^t - \eta_t\nabla\ell\big(w_{\mathcal{T}_i}(w^t, S_i^{\mathrm{tr}}), z^{\mathrm{ts}}\big)\big) - \big(\widetilde{w}^t - \eta_t\nabla\ell\big(w_{\mathcal{T}_i}(\widetilde{w}^t, S_i^{\mathrm{tr}}), z^{\mathrm{ts}}\big)\big)\| \leq (1 + \eta_t\phi_t)\mathbb{E}\|w^t - \widetilde{w}^t\|.$$

To bound the second term on the RHS of (74), similar to the proof of (68), under the $\lambda - L$ strongly-convex setting, we have

$$\mathbb{E}\|\nabla\ell\big(w_{\mathcal{T}_i}(\widetilde{w}^t, S_i^{\text{tr}}), z^{\text{ts}}\big) - \nabla\ell\big(w_{\mathcal{T}_i}(\widetilde{w}^t, \widetilde{S}_i^{\text{tr}}), z^{\text{ts}}\big)\| \leq \frac{2L(G^2 + G)}{(\lambda - L)n^{\text{tr}}} \tag{75}$$

Substituting (75) and (74) into (73), we obtain

$$\mathbb{E}\|w^{t+1} - \widetilde{w}^{t+1}\| \leq (1 + \eta_t\phi_t)\mathbb{E}\|w^t - \widetilde{w}^t\| + \eta_t\frac{2L(G^2 + G)}{(\lambda - L)n^{\text{tr}}} + 2\eta_t\mathbb{E}\|\nabla\widehat{F}_i(w^t)\|.$$

From Lemma 7, we can know $\mathbb{E}\|\nabla\widehat{F}_i(w^t)\| \leq eG$. Then combing the above two cases, we obtain

$$\mathbb{E}\|w^{t+1} - \widetilde{w}^{t+1}\| \leq (1 - \frac{1}{m})(1 + \eta_t\phi_t)\mathbb{E}\|w^t - \widetilde{w}^t\| + \frac{1}{m}(1 + \eta_t\phi_t)\mathbb{E}\|w^t - \widetilde{w}^t\|$$
$$+ \frac{2L(G^2 + G)}{(\lambda - L)mn^{\text{tr}}} + \frac{2\eta_t eG}{m}$$
$$\leq \exp(\eta_t\phi_t)\mathbb{E}\|w^t - \widetilde{w}^t\| + \frac{\eta_t\Phi}{m}.$$

where we use $1 + x \leq \exp(x)$ and $\alpha \leq \frac{1}{QL}$ in the second inequality, and we denote $\Phi = 2eG + \frac{2L(G^2 + G)}{(\lambda - L)n^{\text{tr}}}$. Using the same technique in Theorem 3. Then we complete the proof.

### D.6 Fo-MuML(convex)

Fo-MuML is more like a application of FoMAML in PDF. In particular, in the ourter-level, We no longer derive the derivative of w, we take the derivative of $w_{\mathcal{T}_i, Q}$. Then we have

$$\|w^{t+1} - \widetilde{w}^{t+1}\| = \|\big(w^t - \eta_t\nabla\widehat{\mathcal{L}}\big(w_{\mathcal{T}_j, Q}(w^t, S_j^{\text{tr}}), S_j^{\text{ts}}\big)\big) - \big(\widetilde{w}^t - \eta_t\nabla\widehat{\mathcal{L}}\big(w_{\mathcal{T}_j, Q}(\widetilde{w}^t, S_j^{\text{tr}}), S_j^{\text{ts}}\big)\big)\|$$
$$\leq \frac{1}{n^{\text{ts}}}\sum_{z^{\text{ts}} \in S_j^{\text{ts}}}\|\big(w^t - \eta_t\nabla\ell\big(w_{\mathcal{T}_j, Q}(w^t, S_j^{\text{tr}}), z^{\text{ts}}\big)\big) - \big(\widetilde{w}^t - \eta_t\nabla\ell\big(w_{\mathcal{T}_j, Q}(\widetilde{w}^t, S_j^{\text{tr}}), z^{\text{ts}}\big)\big)\|$$
$$\leq \|w^t - \widetilde{w}^t\| \tag{76}$$

Next, for a given time index $t$, with probability $\frac{1}{m}$, the task $\mathcal{T}_i$ is selected. In this case, we have

$$\mathbb{E}\|w^{t+1} - \widetilde{w}^{t+1}\| = \mathbb{E}\|\big(w^t - \eta_t\nabla\widehat{\mathcal{L}}\big(w_{\mathcal{T}_i, Q}(w^t, S_i^{\text{tr}}), S_i^{\text{ts}}\big)\big) - \big(\widetilde{w}^t - \eta_t\nabla\widehat{\mathcal{L}}\big(w_{\mathcal{T}_i, Q}(\widetilde{w}^t, \widetilde{S}_i^{\text{tr}}), \widetilde{S}_i^{\text{ts}}\big)\big)\|$$
$$\leq \mathbb{E}\|\big(w^t - \eta_t\nabla\widehat{\mathcal{L}}\big(w_{\mathcal{T}_i, Q}(w^t, S_i^{\text{tr}}), S_i^{\text{ts}}\big)\big) - \big(\widetilde{w}^t - \eta_t\nabla\widehat{\mathcal{L}}\big(w_{\mathcal{T}_i, Q}(\widetilde{w}^t, \widetilde{S}_i^{\text{tr}}), S_i^{\text{ts}}\big)\big)\|$$
$$+ \mathbb{E}\|\eta_t\nabla\widehat{\mathcal{L}}\big(w_{\mathcal{T}_i, Q}(\widetilde{w}^t, \widetilde{S}_i^{\text{tr}}), \widetilde{S}_i^{\text{ts}}\big) - \eta_t\nabla\widehat{\mathcal{L}}\big(w_{\mathcal{T}_i, Q}(\widetilde{w}^t, \widetilde{S}_i^{\text{tr}}), S_i^{\text{ts}}\big)\|$$
$$\leq \frac{1}{n^{\text{ts}}}\sum_{z^{\text{ts}} \in S_i^{\text{ts}}}\mathbb{E}\|\big(w^t - \eta_t\nabla\widehat{\mathcal{L}}\big(w_{\mathcal{T}_i, Q}(w^t, S_i^{\text{tr}}), z^{\text{ts}}\big)\big) - \big(\widetilde{w}^t - \eta_t\nabla\widehat{\mathcal{L}}\big(w_{\mathcal{T}_i, Q}(\widetilde{w}^t, \widetilde{S}_i^{\text{tr}}), z^{\text{ts}}\big)\big)\|$$
$$+ \eta_t\mathbb{E}\|\nabla\widehat{\mathcal{L}}\big(w_{\mathcal{T}_i, Q}(\widetilde{w}^t, \widetilde{S}_i^{\text{tr}}), \widetilde{S}_i^{\text{ts}}\big) - \nabla\widehat{\mathcal{L}}\big(w_{\mathcal{T}_i, Q}(\widetilde{w}^t, \widetilde{S}_i^{\text{tr}}), S_i^{\text{ts}}\big)\|$$
$$\leq \frac{1}{n^{\text{ts}}}\sum_{z \in S_i^{\text{ts}}}\mathbb{E}\|\big(w^t - \eta_t\nabla\ell\big(w_{\mathcal{T}_i, Q}(w^t, S_i^{\text{tr}}), z^{\text{ts}}\big)\big) - \big(\widetilde{w}^t - \eta_t\nabla\ell\big(w_{\mathcal{T}_i, Q}(\widetilde{w}^t, \widetilde{S}_i^{\text{tr}}), z^{\text{ts}}\big)\big)\|$$
$$+ 2\eta_t\mathbb{E}\|\nabla\widehat{F}_i(w)\|, \tag{77}$$

where the last inequality follows that $\widetilde{S}_i^{\text{ts}}$ and $S_i^{\text{ts}}$ are sampled from the same distribution, then $\mathbb{E}\|\nabla\widehat{\mathcal{L}}\big(w_{\mathcal{T}_i}(\widetilde{w}^t, \widetilde{S}_i^{\text{tr}}), \widetilde{S}_i^{\text{ts}}\big)\| = \mathbb{E}\|\nabla\widehat{\mathcal{L}}\big(w_{\mathcal{T}_i}(\widetilde{w}^t, \widetilde{S}_i^{\text{tr}}), S_i^{\text{ts}}\big) = \mathbb{E}\|\nabla\widehat{F}_i(w)\|$. Note that

$$\mathbb{E}\|\big(w^t - \eta_t\nabla\ell\big(w_{\mathcal{T}_i}(w^t, S_i^{\text{tr}}), z^{\text{ts}}\big)\big) - \big(\widetilde{w}^t - \eta_t\nabla\ell\big(w_{\mathcal{T}_i, Q}(\widetilde{w}^t, \widetilde{S}_i^{\text{tr}}), z^{\text{ts}}\big)\big)\|$$
$$\leq \mathbb{E}\|\big(w^t - \eta_t\nabla\ell\big(w_{\mathcal{T}_i, Q}(w^t, S_i^{\text{tr}}), z^{\text{ts}}\big)\big) - \big(\widetilde{w}^t - \eta_t\nabla\ell\big(w_{\mathcal{T}_i, Q}(\widetilde{w}^t, S_i^{\text{tr}}), z^{\text{ts}}\big)\big)\| \tag{78}$$
$$+ \eta_t\mathbb{E}\|\nabla\ell\big(w_{\mathcal{T}_i, Q}(\widetilde{w}^t, S_i^{\text{tr}}), z^{\text{ts}}\big) - \nabla\ell\big(w_{\mathcal{T}_i, Q}(\widetilde{w}^t, \widetilde{S}_i^{\text{tr}}), z^{\text{ts}}\big)\|.$$

Let us bound the two terms on the RHS of (78) separately. First, similar to how we derived (76), we could bound the first term by

$$\mathbb{E}\|\big(w^t - \eta_t\nabla\ell\big(w_{\mathcal{T}_i, Q}(w^t, S_i^{\text{tr}}), z^{\text{ts}}\big)\big) - \big(\widetilde{w}^t - \eta_t\nabla\ell\big(w_{\mathcal{T}_i, Q}(\widetilde{w}^t, S_i^{\text{tr}}), z^{\text{ts}}\big)\big)\| \leq \mathbb{E}\|w^t - \widetilde{w}^t\|.$$

To bound the second term on the RHS of (78), we consider two parallel processes of generating iterates $\{\widetilde{w}_{\mathcal{T}_i,q}^t\}$ and $\{\widetilde{w}_{\mathcal{T}_i,q}^{t,\prime}\}$ by using datasets $S_i^{\mathrm{tr}}$ and $\widetilde{S}_i^{\mathrm{tr}}$, respectively. Note that

$$
\begin{aligned}
\mathbb{E}\|\nabla\ell\big(w_{\mathcal{T}_i}(\widetilde{w}^t, S_i^{\mathrm{tr}}), z^{\mathrm{ts}}\big) - \nabla\ell\big(w_{\mathcal{T}_i}(\widetilde{w}^t, \widetilde{S}_i^{\mathrm{tr}}), z^{\mathrm{ts}}\big)\| &= \mathbb{E}\|\nabla\ell(\widetilde{w}_{\mathcal{T}_i,Q}^t, z^{\mathrm{ts}}) - \nabla\ell(\widetilde{w}_{i,Q}^{t,\prime}, z^{\mathrm{ts}})\| \\
&\leq L\mathbb{E}\|\widetilde{w}_{\mathcal{T}_i,Q}^t - \widetilde{w}_{i,Q}^{t,\prime}\| \\
&\leq \frac{2G^2}{\lambda n^{\mathrm{tr}}},
\end{aligned}
\tag{79}
$$

Substituting (79) and (78) into (77), we have

$$
\mathbb{E}\|w^{t+1} - \widetilde{w}^{t+1}\| \leq \mathbb{E}\|w^t - \widetilde{w}^t\| + \eta_t \frac{2G^2}{\lambda n^{\mathrm{tr}}} + 2\eta_t \mathbb{E}\|\nabla\widehat{F}_i(w^t)\|.
$$

Combing the above two cases, we obtain

$$
\begin{aligned}
\mathbb{E}\|w^{t+1} - \widetilde{w}^{t+1}\| &\leq (1 - \frac{1}{m})\mathbb{E}\|w^t - \widetilde{w}^t\| + \frac{1}{m}\mathbb{E}\|w^t - \widetilde{w}^t\| + \frac{1}{m}\eta_t\frac{2G^2}{\lambda n^{\mathrm{tr}}} + \frac{2}{m}\eta_t\mathbb{E}\|\nabla\widehat{F}_i(w^t)\| \\
&= \mathbb{E}\|w^t - \widetilde{w}^t\| + \eta_t\frac{2G^2}{\lambda m n^{\mathrm{tr}}} + \frac{2\eta_t}{m}\mathbb{E}\|\nabla\widehat{F}_i(w^t)\|.
\end{aligned}
$$

Unrolling it and noting that $\|w^0 - \widetilde{w}^0\| = 0$, we have

$$
\begin{aligned}
\mathbb{E}[\|w^T - \widetilde{w}^T\|] &\leq \sum_{t=0}^{T-1} \eta_t \frac{2G^2}{\lambda m n^{\mathrm{tr}}} + \sum_{t=0}^{T-1} \frac{2\eta_t}{m}\mathbb{E}\|\nabla\widehat{F}(w^t, S_i)\| \\
&\leq \sum_{t=0}^{T-1} \eta_t \frac{2G^2}{\lambda m n^{\mathrm{tr}}} + \frac{1}{m}\sqrt{F(w^0) - \min_{\mathcal{W}} F + \frac{L_Q\sigma^2}{2}\sum_{t=0}^{T-1}\eta_t^2},
\end{aligned}
$$

where we use Lemma 13 in the last inequality. which completes the proof.

### D.7 FoMuML(non-convex)

This proof is similar to the proof of Theorem 3, we have

$$
\begin{aligned}
\|w^{t+1} - \widetilde{w}^{t+1}\| &= \|\big(w^t - \eta_t\nabla\widehat{\mathcal{L}}(w_{\mathcal{T}_j}(w^t, S_j^{\mathrm{tr}}), S_j^{\mathrm{ts}})\big) - \big(\widetilde{w}^t - \eta_t\nabla\widehat{\mathcal{L}}(w_{\mathcal{T}_j}(\widetilde{w}^t, S_j^{\mathrm{tr}}), S_j^{\mathrm{ts}})\big)\| \\
&\leq (1 + \eta_t\phi_t)\|w^t - \widetilde{w}^t\|,
\end{aligned}
\tag{80}
$$

where $\phi_t = \min\{L_Q, \xi_t\}$ with $\xi_t = \|\nabla^2 F(w_0, S_t)\| + \frac{\rho_Q}{2}\left\|\sum_{l=1}^{t-1}\beta_l\nabla\widehat{F}(w_S^l)\right\| + \frac{\rho_Q}{2}\left\|\sum_{l=1}^{t-1}\beta_l\nabla\widehat{F}(w_{\widetilde{S}}^l)\right\|$.

Next, for the second case, similar to the proof of (77), we have

$$
\begin{aligned}
\mathbb{E}\|w^{t+1} - \widetilde{w}^{t+1}\| &\leq \frac{1}{n^{\mathrm{ts}}}\sum_{z\in S_i^{\mathrm{ts}}} \mathbb{E}\|\big(w^t - \eta_t\nabla\ell\big(w_{\mathcal{T}_i}(w^t, S_i^{\mathrm{tr}}), z^{\mathrm{ts}}\big)\big) \\
&\quad - \big(\widetilde{w}^t - \eta_t\nabla\ell\big(w_{\mathcal{T}_i}(\widetilde{w}^t, \widetilde{S}_i^{\mathrm{tr}}), z^{\mathrm{ts}}\big)\big)\| + 2\eta_t\mathbb{E}\|\nabla\widehat{F}_i(w)\|.
\end{aligned}
$$

Note that

$$
\begin{aligned}
&\mathbb{E}\|\big(w^t - \eta_t\nabla\ell\big(w_{\mathcal{T}_i}(w^t, S_i^{\mathrm{tr}}), z^{\mathrm{ts}}\big)\big) - \big(\widetilde{w}^t - \eta_t\nabla\ell\big(w_{\mathcal{T}_i}(\widetilde{w}^t, \widetilde{S}_i^{\mathrm{tr}}), z^{\mathrm{ts}}\big)\big)\| \\
&\leq \mathbb{E}\|\big(w^t - \eta_t\nabla\ell\big(w_{\mathcal{T}_i}(w^t, S_i^{\mathrm{tr}}), z^{\mathrm{ts}}\big)\big) - \big(\widetilde{w}^t - \eta_t\nabla\ell\big(w_{\mathcal{T}_i}(\widetilde{w}^t, S_i^{\mathrm{tr}}), z^{\mathrm{ts}}\big)\big)\| \\
&\quad + \eta_t\mathbb{E}\|\nabla\ell\big(w_{\mathcal{T}_i}(\widetilde{w}^t, S_i^{\mathrm{tr}}), z^{\mathrm{ts}}\big) - \nabla\ell\big(w_{\mathcal{T}_i}(\widetilde{w}^t, \widetilde{S}_i^{\mathrm{tr}}), z^{\mathrm{ts}}\big)\|.
\end{aligned}
\tag{81}
$$

Let us bound the two terms on the RHS of (81), separately. First, similar to how we bound (80), we could bound the first term by

$$
\mathbb{E}\|\big(w^t - \eta_t\nabla\ell\big(w_{\mathcal{T}_i}(w^t, S_i^{\mathrm{tr}}), z^{\mathrm{ts}}\big)\big) - \big(\widetilde{w}^t - \eta_t\nabla\ell\big(w_{\mathcal{T}_i}(\widetilde{w}^t, S_i^{\mathrm{tr}}), z^{\mathrm{ts}}\big)\big)\| \leq (1 + \eta_t\phi_t)\mathbb{E}\|w^t - \widetilde{w}^t\|.
$$

To bound the second term on the RHS of (81), similar to the proof of (79), we have

$$
\mathbb{E}\|\nabla\ell\big(w_{\mathcal{T}_i}(\widetilde{w}^t, S_i^{\mathrm{tr}}), z^{\mathrm{ts}}\big) - \nabla\ell\big(w_{\mathcal{T}_i}(\widetilde{w}^t, \widetilde{S}_i^{\mathrm{tr}}), z^{\mathrm{ts}}\big)\| \leq \frac{2G^2}{(\lambda - L)n^{\mathrm{tr}}}
\tag{82}
$$

Substituting (82) and (81) into (81), we obtain

$$\mathbb{E}\|w^{t+1} - \widetilde{w}^{t+1}\| \leq (1 + \eta_t \phi_t)\mathbb{E}\|w^t - \widetilde{w}^t\| + \eta_t \frac{2G^2}{(\lambda - L)n^{\mathrm{tr}}} + 2\eta_t \mathbb{E}\|\nabla \widehat{F}_i(w^t)\|.$$

From Lemma 7, we can know $\mathbb{E}\|\nabla \widehat{F}_i(w^t)\| \leq eG$. Then combing the above two cases, we obtain

$$\mathbb{E}\|w^{t+1} - \widetilde{w}^{t+1}\| \leq (1 - \frac{1}{m})(1 + \eta_t \phi_t)\mathbb{E}\|w^t - \widetilde{w}^t\| + \frac{1}{m}(1 + \eta_t \phi_t)\mathbb{E}\|w^t - \widetilde{w}^t\|$$

$$+ \frac{2G^2}{(\lambda - L)mn^{\mathrm{tr}}} + \frac{2\eta_t eG}{m}$$

$$\leq \exp(\eta_t \phi_t)\mathbb{E}\|\mathrm{w}^t - \widetilde{\mathrm{w}}^t\| + \frac{\eta_t \Phi}{\mathrm{m}}.$$

where we use $1 + x \leq \exp(\mathrm{x})$ and $\alpha \leq \frac{1}{QL}$ in the second inequality, and we denote $\Phi = 2eG + \frac{2G^2}{(\lambda - L)n^{\mathrm{tr}}}$. Using the same technique in Theorem 3. Then we complete the proof.

