# OpenReview forum: "On the Stability and Generalization of Meta-Learning: the Impact of Inner-Levels"
_NeurIPS.cc/2025/Conference — NeurIPS 2025 poster_

### Official Review · Reviewer_hYHQ · 2025-06-23

**Clarity:** 2
**Significance:** 3
**Originality:** 2
**Rating:** 4
**Confidence:** 3

**Summary:**

This paper provides new algorithmic-stability based generalisation bounds for meta-learning focussing on the impact of inner level optimization. The authors consider two broad classes of inner-optimisation frameworks: gradient descent framework (GDF), which includes classical MAML algorithm,  and proximal descent framework (PDF) that includes the iMAML and Meta-Minibatch Pros algorithms. Leveraging on-average algorithmic stability framework, the authors derive new meta-generalisation bounds that reveals the influence of the number $Q$ of inner-level iterations. For the GDF framework, authors report a trade-off relationship with respect to the choice of $Q$. The authors finally also propose a meta-objective with improved generalization performance.

**Questions:**

I think the following points need clarification:

1. Authors use Fig.1 to motivate the problem setting. However, their claim regarding MAML -- that generalization error first decreases with Q and then increases -- is not convincing from Fig.1(a). This will require more plots corresponding to increasing values of Q>20.

2. In Section 4, the authors present Theorem 2 and Theorem 3. The assumptions in both theorems are the same. Its hard to see the difference in the setting (strongly convex/non-convex) considered. Better statement of the theorems are required. Along this line, even the statement of Theorem 1 is incomplete with no reference to $\epsilon$.


3. Following Theorem 2 and Remark 2, it seems like the proposed bound in Theorem 2 yields looser bound  than the one in reference [9] in the paper. The authors claim that setting $\alpha \leq 1/(QL)$ simplifies the first term in the bound of [9] to scale as $O(1/n)$, mitigating the dependence on $Q$. However, under the same $\alpha$, the first term in Theorem 2 still has a linear dependence on $Q$. Note that for sufficiently large $Q$, $\alpha \leq 1/(QL)$ yields negligibly small learning rate, implying that the inner update will be close to the initialisation $w^t_{\mathcal{T}_i,0}$ for each task. Isn't it then reasonable to assume that no dependence on $Q$ in this regime is better?

4. Additionally, the term $F(w^0)-\min_wF(w)$ accounts for the difference between the meta-objective $F(\cdot)$ evaluated at initialisation $w_0$ and that evaluated with respect to the best initialisation $w^* =\arg \min_w F(w)$. It is not clear how increasing $Q$ can decrease this difference, as this difference depend on the proximity of $w^0$ to $w^*$.  Of course, this is true if the term was $F(w^T)-\min_w F(w)$, the difference between the meta-learned initialisation and the best initialisation possible.  I believe reference [9] has the latter term in their bound. The above issue also arises  Theorem 3, 4 and 5. This concern could also be attributed to the lack of clear definitions in the paper.  In this sense, I am also unsure of the relevance of Section 4.3.

5. More discussion on how the proposed bounds in Theorems 4 and 5 for PDF framework compare to other existing bounds are required.

**Ethical Concerns:**

["NO or VERY MINOR ethics concerns only"]

**Final Justification:**

Following the rebuttal, the authors have clarified my main concerns. The notational discrepancies were corrected, and additional experimental results are included. Given the strong theoretical contributions of the paper, I am happy to increase my score to 4: Borderline acceptance.

**Limitations:**

The work is purely theoretical in nature, and does not pose any negative societal impact.

**Paper Formatting Concerns:**

Already listed

**Quality:**

2

**Strengths And Weaknesses:**

The paper belongs to a recent series of works that have explored the generalization analysis of meta-learning algorithms. The general classification of the existing frameworks into GDF and PDF is interesting, and enables authors to simultaneously investigate multiple existing algorithms.

However, the paper is not easy to read and lack clarity in many sections. There is an overload of notation especially in Section 3, which makes it difficult to follow. I believe a thorough re-writing is required in Section 3. For instance, in Section 3.2, authors define $\hat{F_i}(w,S_i)$ and relate it to $\hat{\mathcal{L}}_{\lambda}(w,S_i)$, which is not used anywhere else . Later on, they introduce $\hat{\mathcal{K}}(w,S_i)$, whose relationship with $\hat{F}_i(w,S_i)$ requires clarification. In Algorithm 1, line 7, what is the loss function $\mathcal{L}$? Shouldnt this be $\mathcal{L}_i$?

Significance and Originality: The investigation of the impact of inner-level optimization on the generalization of meta-learning algorithms is not new. For instance, reference [1] has already studied this in the GDF framework and has reported a similar trade-off relationship with respect to the number Q of inner level iterations. The authors in this work have gone beyond the GDF and have analysed PDF under both convex and non-convex assumptions on the loss function.

Weaknesses: Please find the weaknesses under the Questions Tab.


Reference:
[1] Zhou, P., Zou, Y., Yuan, X.T., Feng, J., Xiong, C. and Hoi, S., 2021, December. Task similarity aware meta learning: Theory-inspired improvement on MAML. In Uncertainty in artificial intelligence (pp. 23-33). PMLR.

---

> ### Author Rebuttal · Authors · 2025-07-31
>
> We thank for all your valuable comments. According to the weaknesses and questions, we will reply these individually as the following.
>
> **Weakness 1:** Although we acknowledge certain typographical errors pointed out by the reviewer, we believe we have clarified the meaning of all symbols used in our paper. Regarding the terms $\hat{F}\_i(w, S\_i)$ and $\hat{L}\_{\lambda}(w, S\_i)$ and their relationship, we have explained them in lines 144-145. In particular, $\hat{F}\_i(w, S\_i)$ represents the empirical loss, i.e., the training loss, which has been explained in lines 132-133. We use $\hat{L}\_{\lambda}(w, S\_i)$ to reflect the effect of $\lambda$. If we did not do so, the definition of $F$ would be confusing. As for the term $\hat{\mathcal{K}}(w, S\_i)$, we have addressed it in lines 146-149: "For a practical PDF-based algorithm, it's difficult to obtain an exact solution of $\widehat{\mathcal{L}}\_{\lambda}(w, S\_i)$, so we typically resort to an inexact solution, $\widehat{\mathcal{K}}(w, S\_i) = \frac{1}{n} \sum\_{z \in S\_i} \ell(w\_{\mathcal{T}\_i}, z) + \frac{\lambda}{2} \| w\_{\mathcal{T}\_i} - w \|^2$."
>
> **Weakness 2 and Question 3:** We would like to emphasize that our work significantly extends the contributions of [1]. While [1] investigates the trade-off in MAML, their analysis is conducted within a relatively narrow scope and lacks a general or extensible framework. In contrast, we introduce two flexible and broadly applicable frameworks—GDF and PDF—which are designed to accommodate a wide range of meta-learning algorithms. To illustrate the generality and practicality of these frameworks, we derive generalization bounds for six different algorithms. Moreover, our GDF framework provides a more robust and interpretable characterization of the trade-off. Specifically, when comparing our Theorem 2 with the results of [1], let us consider the case where $T=m$. Under this, our theorem 2 yields a bound $\mathcal{O}(\frac{Q}{n}+\frac{\sqrt{F(w^0)-\min\_{\mathcal{W}}F}}{m})$ whereas the bound in [1] is $\mathcal{O}(\frac{1}{n}+F(w^T)-\min\_{\mathcal{W}}F))$. It is evident that our second term $\frac{\sqrt{F(w^0)-\min\_{\mathcal{W}}F}}{m})$ is significantly lower than the corresponding  $F(w^T)-\min\_{\mathcal{W}}F$ in [1]. Although our first term is looser than theirs $\frac{1}{n}$, we argue that it is intuitive outcome.. In particular, this statement, "For sufficiently large $Q$, $\alpha\leq \frac{1}{QL}$ yields negligibly small learning rate, implying the inner update will be close to initialisation $w\_{\tau\_i,0}^t$" reinforces the validity of our bound. Intuitively, if the inner-loop update is too close to the initialization, the model fails to learn task-specific information effectively, leading to larger generalization error—similar to the behavior observed when $Q=1$. However, the results in [1] ca not explain the trade-off relationship either is sufficiently small or large $Q$ when $\alpha\leq \frac{1}{QL}$, which means their results is not robust.
>
> **Question 1:** The core idea of MAML involves applying a few gradient descent steps on a small number of training samples to adapt the meta-model to test tasks. Therefore, We use Fig. 1 to illustrate an interesting phenomenon, i.e., a larger number of $Q$ is not suitable for MAML. To address the reviewers' concerns, we also provide further experiment for MAML when $Q>20$. The experiment is also conducted on the Omniglot dataset. We employ a $3\times 3$ CNN and use the Cross-Entropy Loss as the loss function. For each training task, the number of support samples and query samples to $n^{\rm{tr}}=1$ and $n^{\rm{ts}}=5$, respectively. For each test task, we use $n^{\rm{tr}}=1, n^{\rm{ts}}=15$. In addition, each task is formulated as a $5$-way classification problem.
>
> | Q=2    | Q=5    | Q=10   | Q=20   | Q=30   | Q=40   | Q=50   |
> | ------ | ------ | ------ | ------ | ------ | ------ | ------ |
> | 0.6429 | 0.2868 | 0.2300 | 0.2417 | 0.2423 | 0.2541 | 0.2622 |
>
> **Question 2:** In theoretical analysis, two identical assumptions do not necessarily imply that the difficulty and process of derivation under the two settings is the same, especially in stability analysis. In fact, the condition of some parameter is stricter in non-convex. For instance, under convex setting, the step size $\eta\leq\frac{1}{L\_Q}$ with $L\_Q = \alpha\rho Q(1+\alpha L)^{Q-1} + (1+\alpha L)^{Q}L$. In contrast, under non-convex setting, the step size $\eta\_t \leq \frac{c}{t}$ and $c\leq\min\{\frac{1}{L_Q},\frac{1}{4(2L\_Q\rm{ln}(T))^2}\}$ with $L\_Q= \frac{3\rho(1+\alpha L)^{2(Q-1)}}{L} + (1+\alpha L)^{Q}L$.
>
> **Question 4:** Referring to the Eq (1) and Eq(2) in this paper, we can get $F(w)=\frac{1}{m}\sum\_{i=1}^{m} F\_i(w)=\frac{1}{m}\sum\_{i=1}^{m} \mathbb{E}\_{\mathcal{D}\_i} \mathbb{E}\_{z\in\mathcal{P}\_i}[\ell(w\_{\mathcal{T}\_{i}}^Q(w, \mathcal{D}\_i),z]$. Then we can find that $F_i(w)$ is benefited by $Q$ due to $w\_{\tau\_i}^Q(w, \mathcal{D}\_i)=w-\alpha\sum\_{q=0}^{Q-1}\nabla \widehat{\mathcal{L}}\_i( w\_{\tau\_i}^q,\mathcal{D}\_i)$. Therefore, both $F(w^T)-\min\_{\mathcal{W}}F$ and $F(w^0)-\min\_{\mathcal{W}}F$ can be improved by increasing $Q$. The main difference is that $F(w^T)-\min\_{\mathcal{W}}F$ can also be improved by the number of outer-loop iterations $T$.
>
> **Question 5:** Since our work primarily focuses on generalization error, while other studies emphasize transfer error, it is difficult to make a fair comparison regarding the tightness of the bounds. Generalization error measures the discrepancy between training tasks and test tasks, making it a key metric for evaluating the practical capability of meta-learning algorithms.
>
> [1] Zhou, P., Zou, Y., Yuan, X.T., Feng, J., Xiong, C. and Hoi, S., 2021, December. Task similarity aware meta learning: Theory-inspired improvement on MAML. In Uncertainty in artificial intelligence (pp. 23-33). PMLR.

---

> > ### Comment · Reviewer_hYHQ · 2025-08-07
> > **Thanks**
> >
> > I thank the authors for clarifying my concerns and for providing extended experimental results. I would like to increase my score and lean towards borderline acceptance

---

> > > ### Author Response · Authors · 2025-08-07
> > >
> > > We sincerely appreciate the reviewers’ recognition of our work. If there are any additional questions you would like to discuss, please feel free to let us know.

---

### Official Review · Reviewer_wmex · 2025-06-28

**Clarity:** 3
**Significance:** 3
**Originality:** 3
**Rating:** 5
**Confidence:** 3

**Summary:**

This paper mainly focuses on exploring meta-learning algorithms. As previous works usually pay more attention to training strategies and data, the impact of inner-loop optimization on generalization is ignored. Thus, in this paper, the authors explore both gradient descent and proximal descent frameworks to obtain insights regarding meta-learning paradigm.

**Questions:**

- Could you provide some explanations regarding Eq. (2)? I do not really understand the formulation.
- What is the $\mathcal{K}$ in Algorithm 1? Although it is explained in the later part, it would be better to define it before.

**Ethical Concerns:**

["NO or VERY MINOR ethics concerns only"]

**Final Justification:**

I will maintain my score regarding this paper since this paper is good and my concerns have been well addressed.

**Quality:**

3

**Strengths And Weaknesses:**

Pros:
- Different from previous work, this work dives into the impact of inner-loop optimizations on the generalization in meta-learning algorithms.
- This work achieves some interesting insights from the perspective of theory.
- The paper starts from an interesting phenomenon of the generalization error and then develops many valuable insights regarding meta-learning, which is impressive.
- The results reported in this paper are good.

Cons:
From my perspective, I do not think this work has obvious drawbacks. I like this work.

---

> ### Author Rebuttal · Authors · 2025-07-31
>
> We thank for all your valuable comments. According to the questions, we will reply these individually as the following.
>
> **Question 1:** In the inner-level process of GDF, the learner $w$ needs to find task-specific parameters $w\_{\tau_i}$ after undergoing $Q$ times gradient updates. This process can be formulated as $w\_{\tau\_i}=w-\alpha\sum\_{q=0}^{Q-1}\nabla \widehat{\mathcal{L}}\_i(w\_{\tau\_i}^q,\mathcal{D}\_i)$. To emphasize its difference of inner-levels from PDF, we give the optimization objective of inner-process, i.e., Eq(2)=$\min\_{w\_{\mathcal{T}\_{i}}}\left\langle\sum\nolimits\_{q=0}^{Q-1}\nabla \widehat{\mathcal{L}}\_{i}(w\_{\mathcal{T}\_{i}}^{q},\mathcal{D}\_i),w\_{\mathcal{T}\_{i}}-w\right\rangle+\frac{1}{2\alpha}\|w\_{\mathcal{T}\_{i}}-w\|\_{2}^{2}$. By taking the derivative on $w\_{\tau_i}$ and making the gradient equal to $0$, Eq(2) can be written as $w\_{\tau_i}=w-\alpha\sum\_{q=0}^{Q-1}\nabla \widehat{\mathcal{L}}\_i (w\_{\tau_i}^q,\mathcal{D}\_i)$..
>
> **Question 2:** As we illustrated in section 3.2, PDF aims to obtain an exact solution of its empirical meta-loss $\widehat{L}\_{\lambda}(w, S\_i)=\min\_{w\_{\mathcal{T}\_i}}\{\frac{1}{n}\sum\_{z\in S\_i} \ell(w\_{\mathcal{T}\_i} , z)+\frac\lambda2\|w\_{\mathcal{T}\_i}-w\|^2 \}$ which is difficult in practical algorithms. Therefore, we often solve an approximation, i.e., $\widehat{\mathcal{K}}(w, S\_i)=\frac{1}{n}\sum\_{z\in S\_i} \ell(w\_{\mathcal{T}\_i}, z)+\frac\lambda2\|w\_{\mathcal{T}_i}-w\|^2$ in Algorithm 1.

---

### Official Review · Reviewer_3fQk · 2025-06-29

**Clarity:** 2
**Significance:** 3
**Originality:** 2
**Rating:** 4
**Confidence:** 2

**Summary:**

This paper considers two settings - the gradient descent framework (GDF) and proximal descent framework (PDF) - for a number of common meta-learning strategies. It discusses two types of common metrics, the generalisation error and optimisation error, whcih constitute the test error. Given convergence of the optimisation error, the paper produces theory about how the generalisation error changes for the GDF and PDF. It subsequently provides a new meta-objective function, informed by the theory, and a small number of experiments demonstrating that their method has better optimisation characteristics (i.e., better generalisation behaviour than more standard meta-objectives).

**Questions:**

- This work focuses on a subset of meta-learning. Does this theory generalise beyond these MAML-oriented meta-learning approaches?
- Can you explain for my sake what $Q$ is actually referring to?
- How does the tightness and derivation of your bound compare to the others derived in theory that are referenced here?
- Where does the intuition of the new meta-loss come from?

Below I provide some examples of typos:
Line 17 - well-generalisation is confusing
26 - Q inner-level gradient updating
45 - increases with the number of inner-levels Q grows
111 wells -> well
145 - forlicity
329 - 'We' shouldn't be capitalised. Neither should Objective on 333

**Ethical Concerns:**

["NO or VERY MINOR ethics concerns only"]

**Final Justification:**

I feel the theory is good, and htat the empirical side of the paper leaves some to be desired. I also find that there are a number of clarity issues. However, I think the contribution of the paper outweighs these issues, and thus maintain a score of 4 - borderline acceptance.

**Quality:**

3

**Strengths And Weaknesses:**

To preface, I am an empirical researcher and so while I did my best to follow the theory I admit that a times certain bounds and literature were unfamiliar to me. For this reason, I submit my review with slightly lower confidence.

Strengths:
- Throughout the paper, the work is constantly compared and contrasted with preexisting literature. As such, I felt it was very well framed with regards to the surrounding field.
- For a theoretically-focused paper, I felt their was still a reasonable empirical comparison and demonstration that their theoretical bounds transferred to practical tests.
- The paper had a natural narrative, following from introducing the problem setting -> introducing types of errors -> discussing how these errors affect generalisation -> providing theoretical bounds -> showing these bounds echo what happens in practice.
- From what I could follow, the paper is rigorous in its theorems and provides corresponding proofs in the supplementary material. For the sake of final publications, I think it would be good to provide these proofs in the appendix (so that they can all be found in the same PDF).

Weaknesses:
- There were a number of typos that I spotted (discussed below).
- I felt like the paper could have introduced a number of concepts in a background section before jumping in. For instance, what is $Q$ - initially I had assumed this was the number of inner-level gradient updates for e.g., MAML, but the way it is framed in the paper has me guessing if this is the case.
- I think this may be more common in theoretical papers, but it felt like a lot of important concepts were quoted by reference without actually saying what they were. It would be good for some of these to be directly discussed in the main test.
- Occasionally, I found it difficult to see where leaps in theory came from. This is partly a result of my naivete of theoretical meta-learning, as well as the fact that the proofs were held in a separate PDF.
- There is no access to the code, and while in the questionaire it states that significance in experiments was provided, this does not seem to be the case. It is not completely obvious to me that the new proposed objective (which basically just weights larger losses more) actually improves performance, given most lines are practically on top of each other and there are no error bars.
- I cna't quite follow the logic from theory to the new loss. To me the new loss seems to sort of come out of nowhere, and also doesn't seem to have the framing for GDF vs PDF and why this is useful.

---

> ### Author Rebuttal · Authors · 2025-07-31
>
> Thank you for your valuable suggestions, we will reply these as follows.
>
> **Weakness 1:** We appreciate the reviewer for pointing out the typographical errors, which we will promptly correct.
>
> **Weakness 2 and Question 2:** To further clarify the meaning of $Q$, we can use MAML as an example. MAML utilizes available training data from multiple tasks to derive a meta-initialization that performs well after a few updates during testing, tailored to a new task. Unlike traditional supervised learning, where the goal is to find a model that generalizes well to a new task without any adaptation, MAML focuses on finding an initial model that can efficiently learn a new task with only a small amount of labeled data by performing $Q$ gradient updates. Therefore, in this paper, our framework requires undergoing $Q$ gradient updates at the inner level to adapt to a small amount of data. We will add some relevant concepts explanation based on your advice.
>
> **Weakness 3:** We are sorry to make you confused and will explain them in details. Our work primarily focuses on the generalization error in meta-learning, which quantifies the performance gap between training and testing phases. The upper bound of this generalization error characterizes the worst-case performance at test time, and thus, optimizing this upper bound is critical for ensuring the real-world applicability of meta-learning algorithms. Moreover, stability is a powerful analytical tool in generalization theory. It helps us understand which parameters affect generalization and how they influence it. Crucially, the definition of stability plays a central role in this analysis, as it directly determines the tightness and relevance of the resulting bounds. Therefore, we introduce two novel stability definitions for our frameworks.
>
> **Weakness 4:** We further elaborate on our results and the proof approach. Our work focuses on the generalization error in meta-learning, and stability proves to be a powerful tool for deriving the generalization error of algorithms. The first step involves introducing two novel stability definitions for our frameworks, namely GDF and PDF. Based on these definitions, we derive Theorem 1, which states:  $\mathbb{E}\_{\mathcal{A}, \mathcal{S}}\left[F(w\_{\mathcal{S}})-\widehat{F}(w\_{\mathcal{S}},\mathcal{S}\_{i})\right] \leq \frac{1}{m} \sum\_{i=1}^m \frac1{n^{\rm{ts}}} \sum\_{j=1}^{n^{\rm{ts}}} \mathbb{E}\_{\mathcal{A}, \mathcal{S}, \widetilde{S}\_{i,k}^{\rm{tr}}, \tilde{z}\_{i,j}^{\rm{ts}}}[\ell(w\_{\mathcal{T}\_i}(w\_{\mathcal{S}}, \widetilde{S}\_{i,k}^{\rm{tr}}), \widetilde{z}\_{i,j}^{\rm{ts}})-  \ell\left(w\_{\mathcal{T}\_i}(w\_{\widetilde{\mathcal{S}}}, \widetilde{S}\_{i,k}^{\rm{tr}}), \widetilde{z}\_{i,j}^{\rm{ts}}\right)] \leq  \epsilon.$ This provides a bound for the generalization error, $\epsilon\_{\rm{gen}}$. It is important to note that $w\_{\mathcal{S}}$ and $w\_{\widetilde{{\mathcal{S}}}}$ are two different outputs of a randomized algorithm $\mathcal{A}$, which correspond to two neighboring datasets, $\mathcal{S}$ and $\widetilde{{\mathcal{S}}}$, differing by a single data point. Therefore, in our proof, the key idea is to bound the difference between $\ell(w\_{\mathcal{T}\_i}(w\_{\mathcal{S}}, \widetilde{S}\_{i,k}^{\rm{tr}}), \widetilde{z}\_{i,j}^{\rm{ts}})-  \ell\left(w\_{\mathcal{T}\_i}(w\_{\widetilde{\mathcal{S}}}, \widetilde{S}\_{i,k}^{\rm{tr}}), \widetilde{z}\_{i,j}^{\rm{ts}}\right)$. Under our assumption that the loss function is $G$-Lipschitz over $\mathbb{R}^d$, i.e., $| \ell(w,z) - \ell(u,z)|\leq G|w-u|$, we aim to bound the difference between $w\_{\mathcal{T}\_i}(w\_{\mathcal{S}}, \widetilde{S}\_{i,k}^{\rm{tr}})$ and $w\_{\mathcal{T}\_i}(w\_{\widetilde{\mathcal{S}}}, \widetilde{S}\_{i,k}^{\rm{tr}})$. This difference is inherently dependent on the updating rule used by different algorithms.
>
> **Weakness 5:** We have submitted our code in the supplementary materials. You can check them by downloading the supplementary materials.
>
> **Weakness 6 and Question 4:** Our results indicate that the objective function $F(w^0)$ plays a crucial role in reducing the generalization bound. For instance, in the non-convex setting, the generalization bound of GDF is $\mathcal{O}((\frac{1+\frac{1}{c\gamma}}{m}\Phi_Q)^{\frac{1}{c\gamma}}(F(w^0)T)^{\frac{c\gamma}{1+c\gamma}})$ and the generalization bound of PDF is $\mathcal{O}((\frac{1+\frac{1}{c\gamma}}{m}\Phi_Q)^{\frac{1}{c\gamma}}(F(w^0) T)^{\frac{c\gamma}{1+c\gamma}})$. We can observe that size of $F(w^0)$ is important when $T$ is sufficiently large.
>
> To reduce the term $F(w^0)$, we begin with the definition of $F(w)=\frac{1}{m}\sum_{i=1}^m F_i(w)$. Let us consider a simple relaxation technique: given a constant $\beta\in[0,1)$ and $\hat{m}\leq \frac{m}{2}$ with the losses sorted by $\{F_i(w^0)\}$ in  ascending order, we can obtain that $\sum_{i=1}^{\hat{m}}F_i(w^0)^{\psi{(i)}} \leq \sum_{i=\hat{m}+1}^{m}F_i(w^0)^{\psi{(i)}}$, where $\psi{(i)}$ is a mapping function associating the index with the original loss $F_i(w^0)$. Based on this, we can derive that $F(w^0) = \frac{1}{m}\sum_{i=1}^mF_i(w^0)^{\psi{(i)}} \geq \frac{1+\beta}{m}\sum_{i=1}^{\hat{m}}F_i(w^0)^{\psi{(i)}} + \frac{1-\beta}{m} \sum_{i=\hat{m}+1}^{m} F_i(w^0)^{\psi{(i)}}$. Motivated by this result, we propose the following optimization objective:
> $$F_{\rm{new}}(w)=\frac{1+\beta}{\tilde{m}}\sum\nolimits_{i=1}^{m}F_i(w)^{\psi{(i)}}
>     + \frac{1-\beta}{\tilde{m}}\sum\nolimits_{i=\hat{m}+1}^{m}F_i(w)^{\psi_{(i)}},
> $$ where $\tilde{m}=(1+\beta)\hat{m}+(1-\beta)(m-\hat{m})$ is to ensure the lower bound derivation.
>
> **Question 1:** Our works mainly focus on the optimization-oriented approaches, which is a main learning field for meta-learning. Therefore, we summarize the common characteristics of most optimization-oriented algorithms and classify them into GDF and PDF according to their different inner training methods. To validate the scalability of our theoretical frameworks, we derived six popular algorithms, five of which are MAML-oriented (except MetaProx). In fact, our frameworks can also be extended to other meta-learning approaches, e.g., MetaOptNet [1] and Hyper-representation[2].
>
> **Question 3:** Since our work primarily focuses on generalization error, while other studies emphasize transfer error, it is difficult to make a fair comparison regarding the tightness of the bounds. Generalization error measures the discrepancy between training tasks and test tasks, making it a key metric for evaluating the practical capability of meta-learning algorithms. Although some previous works [3][4] have analyzed the generalization error of MAML by using stability, their analyses are limited to the strongly convex setting. In contrast, our study extends the analysis to the more challenging and practically relevant convex and non-convex settings. The most close relevant results to our theorem 2 is theorem 1 in their work [5]. Under the convex-setting, if we let $T=m$, then our theorem 2 yields a bound $\mathcal{O}(\frac{Q}{n}+\frac{\sqrt{F(w^0)-\min_{\mathcal{W}}F}}{m})$. In contrast, the bound in [5] is $\mathcal{O}(\frac{1}{n}+F(w^T)-\min_{\mathcal{W}}F))$. We can observe that our second term $\frac{\sqrt{F(w^0)-\min_{\mathcal{W}}F}}{m})$ is significantly lower than the corresponding  $F(w^T)-\min_{\mathcal{W}}F$ in [5]. Although our first term $\frac{Q}{n}$ is looser than theirs $\frac{1}{n}$, we remain the trade-off relationship about $Q$.
>
> [1] Lee, Kwonjoon, et al. "Meta-learning with differentiable convex optimization." Proceedings of the IEEE/CVF conference on computer vision and pattern recognition. 2019.
>
> [2] Franceschi, Luca, et al. "Bilevel programming for hyperparameter optimization and meta-learning." International conference on machine learning. PMLR, 2018.
>
> [3] Fallah, Alireza, Aryan Mokhtari, and Asuman Ozdaglar. "Generalization of model-agnostic meta-learning algorithms: Recurring and unseen tasks." Advances in Neural Information Processing Systems 34 (2021): 5469-5480.
>
> [4] Wang, Lianzhe, et al. "Improving generalization of meta-learning with inverted regularization at inner-level." Proceedings of the IEEE/CVF Conference on Computer Vision and Pattern Recognition. 2023.
>
> [5] Zhou, Pan, et al. "Task similarity aware meta learning: Theory-inspired improvement on MAML." Uncertainty in artificial intelligence. PMLR, 2021.

---

> > ### Comment · Reviewer_3fQk · 2025-08-05
> >
> > Dear authors,
> >
> > Thank you for your response.
> >
> > Re: $\mathcal{Q}$, that's exactly what I thought. As a meta-learning researcher, I follow that quite clearly, but think the way the notation is introduced *in the paper* may cause clarity issues for those not already familiar with the field.
> >
> > Re: statistical significance, please correct me if I am wrong but the answer in the questionaire is wrong, no? There are no results reported with significance bars.
> >
> >
> > Your response for Q6/W4 unfortunately hasn't been formatted correctly.
> >
> > My feeling is to maintain my score, which recommends borderline acceptance, and I think this is broadly in line with other reviewers. My sense is that the theoretical side of the paper is solid, and that the empirical side leaves something to be desired. However, I choose to maintain my low confidence as I do not feel my knowledge of the theory is sufficient to contextualise this work in the surrounding literature.

---

> > > ### Author Response · Authors · 2025-08-05
> > >
> > > Thanks for the follow up discussion.
> > >
> > > Re: We appreciate and understand the reviewers’ concerns regarding the clarity of $Q$, and we will further clarify and improve it in future revisions.
> > >
> > > Re: Regarding the error bars, we acknowledge the limitations of our experiments and fully understand the reviewers’ concerns. We apologize for not being able to address this issue within the one-week rebuttal period, as training meta-learning models becomes extremely time-consuming especially when $Q$ is large. If our paper is accepted, we will make sure to add the error bars in the final version.
> > >
> > > Re(Q6/W4): Our results indicate that the objective function $F(w^0)$ plays a crucial role in reducing the generalization bound. For instance, in the non-convex setting, the generalization bound of GDF is $\mathcal{O}\big(\frac{1+\frac{1}{c\gamma}}{m}(1+\frac{Q}{n^{\rm{tr}}})\big)^{\frac{1}{c\gamma}}\big(F(w^0)T\big)^{\frac{c\gamma}{1+c\gamma}}\big)$ and the generalization bound of PDF is $\mathcal{O}\big(\frac{1+\frac{1}{c\gamma}}{m}(1+\frac{1}{C^Q})\big)^{\frac{1}{c\gamma}}\big(F(w^0)T\big)^{\frac{c\gamma}{1+c\gamma}}\big)$. We can observe that size of $F(w^0)$ is important when $T$ is sufficently large. To reduce the term $F(w^0)$, we begin with the definition of $F(w)=\frac{1}{m}\sum_{i=1}^m F_i(w)$. Let us consider a simple relaxation technique: given a constant $\beta\in[0,1)$ and $\hat{m}\leq \frac{m}{2}$ with the losses sorted by $\{F_i(w^0)\}$ in {\it ascending} order, we can obtain that $\sum_{i=1}^{\hat{m}}F_i(w^0)^{\psi{(i)}}\leq \sum_{i=\hat{m}+1}^{m}F_i(w^0)^{\psi{(i)}}$, where $\psi{(i)}$ is a mapping function associating the index with the original loss $F_i(w^0)$. Based on this, we can derive $F(w^0) = \frac{1}{m}\sum_{i=1}^mF_i(w^0)^{\psi{(i)}}\geq \frac{1+\beta}{m}\sum_{i=1}^{\hat{m}}F_i(w^0)^{\psi{(i)}} + \frac{1-\beta}{m}\sum_{i=\hat{m}+1}^{m}F_i(w^0)^{\psi{(i)}}$. Motivated by this result, we propose the following optimization objective:
> > > \begin{equation}
> > > \begin{split}
> > >  F_{\rm{new}}(w)=\frac{1+\beta}{\tilde{m}}\sum\nolimits_{i=1}^{m}F_i(w)^{\psi{(i)}}
> > >     + \frac{1-\beta}{\tilde{m}}\sum\nolimits_{i=\hat{m}+1}^{m}F_i(w)^{\psi_{(i)}},
> > > \end{split}
> > > \end{equation}
> > > where $\tilde{m}=(1+\beta)\hat{m}+(1-\beta)(m-\hat{m})$ is to ensure the lower bound derivation.

---

> > > > ### Comment · Reviewer_3fQk · 2025-08-05
> > > >
> > > > Thank you.
> > > >
> > > > Re: error bars, I just want to flag that my concern is two-fold. Obviously, not including any sort of error bars for statistical analysis is not great - but admittedly, this is a pretty classic and lazy reviewer criticism. I do think it’s important here though, especially when this work focuses on ‘stability’ which is near impossible to measure within a single seed when it comes to the complex task of meta-learning.
> > > >
> > > > It’s also that the questionnaire explicitly asks if statistic significance is shown, and your paper says it is (when it’s not).

---

> > > > > ### Author Response · Authors · 2025-08-05
> > > > >
> > > > > Thank you again for further explaining your concerns about error bars— we fully understand and sincerely appreciate it. We believe that our theoretical analysis is solid, and we are grateful for your recognition of this aspect. At the same time, we truly value your suggestion regarding the use of error bars, and we will incorporate them in the final version of the paper. Inspired by your feedback, we will also place greater emphasis on improving experimental details in our future work, rather than focusing solely on theoretical rigor.

---

### Official Review · Reviewer_9dZJ · 2025-06-30

**Clarity:** 3
**Significance:** 3
**Originality:** 3
**Rating:** 4
**Confidence:** 3

**Summary:**

This paper investigates the impact of inner-level updates on the stability and generalization of meta-learning algorithms. The study classifies prominent meta-learning algorithms into two frameworks: the Gradient Descent Framework (GDF) and the Proximal Descent Framework (PDF), based on their inner-processes. It introduces novel definitions of algorithmic stability for both frameworks and derives generalization bounds, revealing a trade-off between inner-levels and generalization in GDF, while PDF exhibits a beneficial relationship. The meta-objective function's critical role in minimizing generalization error is highlighted, leading to the proposal of a simplified meta-objective function to enhance performance. Experiments on datasets like Omniglot validate that GDF's generalization error first decreases then increases with inner-levels, whereas PDF's error consistently decreases, supporting the theoretical findings and the effectiveness of the new objective.

**Questions:**

See above.

**Ethical Concerns:**

["NO or VERY MINOR ethics concerns only"]

**Final Justification:**

After reading the authors' responses, I keep my score of 4 unchanged.
This score is mostly due to the fact that the authors consider a group of meta-learning methods (instead of a single method), and this is general.
Besides, the theoretical part seems correct to me.

I did not give a higher score, since the theoretical contributions still seem limited to me. But this is just a minor problem, and I still vote for an acceptance.

**Limitations:**

See above.

**Quality:**

3

**Strengths And Weaknesses:**

(+)
1. This paper considers an important topic in meta-learning, that is, the generalization performances. The authors use algorithmic stability to deal with the problem, which is algorithm-dependent bound.
2. This paper considers a group of meta-learning methods (gradient descent framework, and proximal descent framework) instead of a single method, which is general. This point is new to me.
3. The authors provide rigorous theoretical analysis on generalization. While I did not check all the details, I believe that the arguments here are mostly correct.
4. The authors further provide empirical evidence to validate their theoretical findings.

(-)
1. The technical novelty seems limited, and the comparison between the upper bounds (instead of the upper and lower bound) stops me from giving a higher score. We know that due to technical limits, we may have to compare between the upper bounds. But this still harms the scope of this paper.
2. The authors may need to add experiments acorss different datasets to validate the generalizability of findings.


Overall, I still lean to recommand accpetance, given these limitations.

---

> ### Author Rebuttal · Authors · 2025-07-31
>
> We thank for your valuable comments and will reply them as the following.
>
> **Question 1:** In generalization analysis, upper bounds are particularly valuable because they characterize the worst-case performance of an algorithm, thereby enhancing its reliability in real-world applications. Especially in stability-based analyses, upper bounds provide deeper insights into the intrinsic properties of learning algorithms. An algorithm $\mathcal{A}$ is said to be $\epsilon\_{\text{gen}}$-stable if the change in its output loss, when applied to two neighboring datasets that differ by a single sample, is bounded by $\epsilon\_{\text{gen}}$. By deriving an upper bound on this stability, we are able to systematically trace how differences between neighboring datasets propagate through both the inner and outer learning levels. This enables our bound to precisely reflect the influence of the inner-level optimization. Moreover, our work makes a significant technical contribution: to the best of our knowledge, this is the first study to establish generalization bounds for meta-learning under two scalable algorithmic frameworks. In addition, we introduce two novel definitions of stability, which serve as the theoretical foundation for our analysis. We also recognize that the lower bound is valuable for the next generation of deductions, and will consider exploring this area of work in the future.
>
> **Question 2:** Thanks for you suggestions and we additionally provide two experiments results in the convex setting and non-convex setting to verify our theoretical results again.
>
> **Few-shot regression on the new synthetic dataset:** Unlike the experiments conducted in our paper that the tasks are created randomly as random sinusoid function, we recreated tasks randomly as either random sinusoid functions, or random linear functions. The amplitude of the sinusoids varies within $[0.1,5.0]$ and the phase within $[0,\pi]$. The slope and intercept of the lines vary in $[-3.0, 3.0]$. The inputs are sampled uniformly in $[-5.0,5.0]$. We also use an MLP network with Means square error loss function. The generalization error is assessed as the difference between training and test errors. For each training task, we set the number of support samples and query samples to $n^{\text{tr}}=5$ and $n^{\text{ts}}=5$, respectively, while for each test task, we use $n^{\text{tr}}=5$,$n^{\text{ts}}=15$. Then we report our results as follows:
>
> | Sinusoid & lines | MAML   | MetaSGD | FOMAML | MetaProx | iMAML  | FOMuML |
> | ---------------- | ------ | ------- | ------ | -------- | ------ | ------ |
> | Q=2              | 0.7916 | 0.8682  | 0.6884 | 0.3330   | 0.3803 | 0.9063 |
> | Q=5              | 0.6393 | 0.6731  | 0.3196 | 0.1507   | 0.3569 | 0.6033 |
> | Q=10             | 0.4863 | 0.4296  | 0.2034 | 0.1040   | 0.3422 | 0.4674 |
> | Q=15             | 0.5356 | 0.5179  | 0.2209 | 0.0233   | 0.1222 | 0.2212 |
>
> | Sinusoid & lines | MAML(Ours)   | MetaSGD(Ours) | FOMAML(Ours) | MetaProx(Ours) | iMAML(Ours)  | FOMuML(Ours) |
> | ---------------- | ------ | ------- | ------ | -------- | ------ | ------ |
> | Q=2              | 0.7624 | 0.8484  | 0.6720 | 0.3190   | 0.3562 | 0.7381 |
> | Q=5              | 0.5964 | 0.6497  | 0.3071 | 0.1263   | 0.3247 | 0.5041 |
> | Q=10             | 0.4681 | 0.4118  | 0.1732 | 0.0594   | 0.2524 | 0.3338 |
> | Q=15             | 0.5144 | 0.4921  | 0.2018 | 0.0162   | 0.0864 | 0.1583 |
>
> **Few-shot classification on CIFARFS:** We conduct the experiment by using the real-world CIFARFS dataset, which is also a few-shot classification taks. It splits the 100 classes from CIFAR-100 into 64, 16 and 20 classes for training, validation, and test respectively. Each class contains 600 images of size 32×32×3. We employ a 3×3 CNN to align support samples and query samples to $n^{\text{tr}}=1$ and $n^{\text{ts}}=5$, respectively. For each test task, we use $n^{\text{tr}}=1$ and $n^{\text{ts}}=15$. Then we report our results as follows:
>
> | CIFARFS | MAML   | MetaSGD | FOMAML | MetaProx | iMAML  | FOMuML |
> | ------- | ------ | ------- | ------ | -------- | ------ | ------ |
> | Q=2     | 0.5989 | 0.5944  | 0.5865 | 0.6536   | 0.8020 | 0.5577 |
> | Q=5     | 0.4022 | 0.4423  | 0.4277 | 0.4636   | 0.7188 | 0.3894 |
> | Q=10    | 0.3677 | 0.2673  | 0.4709 | 0.3721   | 0.4105 | 0.2588 |
> | Q=15    | 0.3951 | 0.4798  | 0.4510 | 0.1360   | 0.2794 | 0.2478 |
>
> | CIFARFS | MAML(Ours)   | MetaSGD(Ours) | FOMAML(Ours) | MetaProx(Ours) | iMAML(Ours)  | FOMuML(Ours) |
> | ------- | ------ | ------- | ------ | -------- | ------ | ------ |
> | Q=2     | 0.5819 | 0.5855  | 0.5434 | 0.6196   | 0.5796 | 0.5430 |
> | Q=5     | 0.3884 | 0.4035  | 0.4137 | 0.3211   | 0.4458 | 0.3675 |
> | Q=10    | 0.3411 | 0.2556  | 0.4352 | 0.2748   | 0.3628 | 0.2473 |
> | Q=15    | 0.3861 | 0.4340  | 0.4493 | 0.0834   | 0.1342 | 0.2310  |
>
> As shown in the table above, a trade-off relationship exists among the GDF algorithms, i.e., MAML, MetaSGD, and FOMAML, while a beneficial relationship is observed among the PDF algorithms, including MetaProx, iMAML, and FOMuML. Furthermore, we present experiments with our new objective, demonstrating that it improves the generalization error across the six algorithms.

---

> > ### Comment · Reviewer_9dZJ · 2025-08-04
> >
> > I thank the authors for their efforts in providing the replies.
> > I keep my score unchanged and lean towards an acceptance.

---

> > > ### Author Response · Authors · 2025-08-05
> > >
> > > We sincerely appreciate the reviewers’ recognition of our work. If there are any additional questions you would like to discuss, please feel free to let us know.

---

### Note · Authors · 2025-08-12

We sincerely thank all reviewers for taking the time to evaluate and discuss our work. During the rebuttal period, we have addressed most of the reviewers’ concerns, and we are deeply grateful that all reviewers ultimately recommended acceptance of our work.

In response to suggestions regarding the explanation of certain symbols and concepts, we will provide clearer definitions in the final version and correct the identified typos. Furthermore, we will include experimental results with error bars and evaluations on more datasets. Looking ahead, we also plan to investigate the generalization lower bound of meta-learning in future work.

---

### Decision · Program_Chairs · 2025-09-17

**Decision:**

Accept (poster)

**Comment:**

This paper theoretically evaluates how the inner update in meta-learning affects generalization performance. This work centers on two representative approaches to the inner update, the gradient descent framework (GDF) and the proximal descent framework (PDF). From the derived theory, the authors show that the trade-off present in the GDF does not arise in PDF. As a consequence, they propose a new reweighting-style meta-objective and demonstrate its empirical effectiveness.

Multiple reviewers positively noted the theoretical significance of analyzing the impact of inner-loop optimization on generalization within two unified frameworks (GDF and PDF) and of providing stability-based generalization bounds. The analysis is sound, and by focusing on GDF and PDF the theory applies to a broad range of existing meta-learning algorithms, which is valuable. The consistency between the empirical results and the theory is also appreciated. Consequently, the proposed loss function is recognized as a contribution to the community.

Reviewers pointed out that the statistical significance of the experiments was not fully validated; in response, the authors committed to multi-seed runs and adding error bars. Several reviewers also found the logical development not always clear; the authors addressed this reasonably in the rebuttal and they promised the update of the manuscript. Although some questioned the novelty of studying the effect of inner-loop optimization on generalization, the authors argued that their analysis of GDF and PDF, which identifies the trade-offs over a wide range of existing methods, is valuable. A remaining concern is that the theoretical analysis provides only upper bounds, but this is a very common approach in generalization-error analysis and is not considered a critical flaw.

Taken together, the paper offers a novel theoretical analysis and a valuable examination of how inner updates influence generalization from the perspectives of GDF and PDF. It introduces a new loss function grounded in solid theory, and its performance is supported by experiments. Since the authors have responded reasonably to the main concerns, I recommend acceptance.